# Singular Bayesian Neural Networks

**Mame Diarra Toure** [1]  **David A. Stephens** [1]

## Abstract

Bayesian neural networks promise calibrated uncertainty but require $O(mn)$ parameters for standard mean-field Gaussian posteriors. We argue this cost is often unnecessary, particularly when weight matrices exhibit fast singular value decay. By parameterizing weights as $W = AB^\top$ with $A \in \mathbb{R}^{m \times r}$, $B \in \mathbb{R}^{n \times r}$, we induce a posterior that is *singular* with respect to the Lebesgue measure, concentrating on the rank-$r$ manifold. This singularity captures structured weight correlations through shared latent factors, geometrically distinct from mean-field's independence assumption. We derive PAC-Bayes generalization bounds whose complexity term scales as $\sqrt{r(m+n)}$ instead of $\sqrt{mn}$, and prove loss bounds that decompose the error into optimization and rank-induced bias using the Eckart-Young-Mirsky theorem. We further adapt recent Gaussian complexity bounds for low-rank deterministic networks to Bayesian predictive means. Empirically, across MLPs, LSTMs, and Transformers on standard benchmarks, our method achieves competitive predictive performance while using up to $33\times$ fewer parameters than 5-member Deep Ensembles. It substantially improves OOD detection and often improves calibration relative to mean-field and perturbation baselines, while Deep Ensembles can still be stronger on in-distribution likelihood-based metrics.

## 1. Introduction

Bayesian neural networks (BNNs) provide principled uncertainty quantification by maintaining distributions over weights rather than point estimates (MacKay, 1992; Neal, 1996). This capability is increasingly critical for high-stakes applications (Amodei et al., 2016) across healthcare (Ruhe

et al., 2019; Dusenberry et al., 2020a;b), autonomous systems (Kendall & Gal, 2017), and domains requiring robust generalization under distribution shift (Hendrycks & Gimpel, 2016; Louizos & Welling, 2017; Ovadia et al., 2019; Abdar et al., 2021). Recent calls for Bayesian approaches in large-scale AI (Papamarkou et al., 2024) underscore the growing recognition that uncertainty-aware models are essential for trustworthy deployment.

However, scaling BNNs to modern architectures remains challenging. Exact Bayesian inference is intractable for neural networks, necessitating approximate methods such as variational inference (Peterson & Anderson, 1987; Hinton & Van Camp, 1993; Jordan et al., 1999) or MCMC sampling (Neal, 1996). While MCMC methods and their modern stochastic variants like SGLD (Welling & Teh, 2011) and stochastic HMC (Chen et al., 2014) provide theoretical guarantees, they require costly iterative sampling that is prohibitive for large-scale models. On the other hand, traditional Mean-field variational inference (MFVI) implementations (Graves, 2011; Kingma & Welling, 2014; Blundell et al., 2015) parameterize each weight with a location-scale family (e.g., Gaussian $\mathcal{N}(\mu, \sigma^2)$), requiring two variational parameters per weight thus doubling the parameter count relative to deterministic networks. Additionally, they impose a fully factorized posterior discarding the structured correlations that may be important for expressiveness, performance and theoretical guarantees. Recent work has further revealed that weight-space inference in modern architectures such as transformers suffers from fundamental pathologies related to prior specification and the difficulty of mapping weight-space to function-space distributions (Cinquin et al., 2021). These limitations have largely confined BNNs to small-scale problems, while modern deep neural networks, from convolutional architectures (He et al., 2016) and LSTMs (Hochreiter & Schmidhuber, 1997) to transformers (Vaswani et al., 2017), routinely scale to billions of parameters. Recent work has begun addressing scalability through low-rank parameterizations. Indeed, neural networks have been observed to exhibit low intrinsic dimensionality (Hinton & Van Camp, 1993; Li et al., 2018; Izmailov et al., 2020; Aghajanyan et al., 2021), suggesting that full-rank parameterizations may be unnecessarily expensive. Current approaches to exploiting this structure fall into three camps: **(i)** post-hoc low-rank perturbations (Dusenberry et al., 2020a; Doan et al., 2025)

---

[1]Department of Mathematics and Statistics, McGill University, Montreal, Canada. Correspondence to: Mame Diarra Toure <mame.toure@mail.mcgill.ca>.

*Proceedings of the 43rd International Conference on Machine Learning*, Seoul, South Korea. PMLR 306, 2026. Copyright 2026 by the author(s).

that add stochastic noise to deterministic backbones, **(ii)** low-rank posterior covariance approximations (Tomczak et al., 2020; Swiatkowski et al., 2020) that enrich mean-field VI by replacing diagonal covariances with low-rank-plus-diagonal structures while maintaining full-rank weight parameterizations, and **(iii)** parameter-efficient low-rank fine-tuning methods like LoRA (Hu et al., 2022) for fine-tuning pre-trained models which inspired bayesian variants (Yang et al., 2024; Meo et al., 2024; Wang et al., 2024). However, all three approaches either require pretrained backbones, sacrificing the end-to-end learning of uncertainty, or, in the case of covariance methods, still parameterize $\mathcal{O}(mn)$ weight means rather than directly exploiting low-rank matrix structure.

At a broader level, our formulation is also distinct from several adjacent literatures. Classical reduced-rank regression (Izenman, 1975; Reinsel & Velu, 1998), probabilistic or Bayesian matrix factorization (Mnih & Salakhutdinov, 2007; Salakhutdinov & Mnih, 2008), and deterministic low-rank compression (Denton et al., 2014; Alvarez & Salzmann, 2017) all share the use of low-rank structure, but they study linear multivariate regression, directly observed data matrices, or deterministic compressed models rather than end-to-end variational uncertainty over neural-network weights. It is also complementary to function-space or post-training uncertainty methods such as linearized Laplace approximations, SNGP, and fixed-mean GP approaches, which attach uncertainty to a trained deterministic predictor or its output layer rather than learning a low-rank posterior over weights from initialization (Immer et al., 2021; Liu et al., 2020; Ortega et al., 2024). An in-depth comparison with these adjacent approaches is provided in **Appendix A**.

To the best of our knowledge, no prior work trains low-rank BNNs end-to-end across diverse architectures with rigorous theoretical guarantees.

We introduce a **fully end-to-end low-rank variational inference framework** for BNNs that jointly learns factorized weight uncertainties from initialization. By parameterizing each weight matrix $W \in \mathbb{R}^{m \times n}$ as $W = AB^\top$ where $A \in \mathbb{R}^{m \times r}$ and $B \in \mathbb{R}^{n \times r}$, we reduce variational parameters from $\mathcal{O}(mn)$ to $\mathcal{O}(r(m + n))$ while *capturing structured correlations* between weights. We provide both theoretical and empirical contributions:

**Theory.** We prove that the induced posterior $q(W)$ on weights is *singular*[1] with respect to Lebesgue measure, concentrating entirely on the measure-zero manifold of

rank-$r$ matrices (See Figure 2). This geometric constraint distinguishes our approach from full-rank mean-field methods and captures non-trivial weight correlations: $\text{Cov}(W_{ij}, W_{i'j'}) \neq 0$ for weights sharing latent factors. Building on this structure, we establish three theoretical guarantees: **(i)** deterministic loss approximation bounds via the Eckart-Young-Mirsky (EYM) theorem (Eckart & Young, 1936), with excess risk controlled by tail singular values $\sum_{i>r} \sigma_i^2$ of the target weight matrix, **(ii)** probabilistic risk decomposition for learned low-rank factors separating learning error from rank bias, and **(iii)** Tighter generalization bounds due to the complexity reduction. These results establish that low-rank structure provides not just efficiency but provably tighter theoretical guarantees when weight matrices exhibit fast singular value decay, a property we verify empirically holds for modern architectures (see Section 4 and Appendix H).

**Implementation.** We implement our method from scratch for three architecture families: multilayer perceptrons (MLPs) (Rumelhart et al., 1986), transformers (Vaswani et al., 2017), and LSTMs (Hochreiter & Schmidhuber, 1997), avoiding black-box libraries to ensure full control over the variational parameterization. Our implementation handles the unique challenges of each architecture: position-wise factorization for transformers, tied weight matrices for LSTMs (Fortunato et al., 2019), and standard feedforward layers for MLPs. **Our variational layers serve as drop-in replacements for standard Keras layers, enabling seamless integration into arbitrary architectures.**

**Experiments.** We evaluate our approach on tabular, text, time-series, and image benchmarks (See section 4 and Appendix I). On classification and regression tasks, rank-$r$ factorization often outperforms full-rank mean-field variational inference (Blundell et al., 2015) across Negative log-likelihood, accuracy, and AUROC. When compared to a 5-member Deep Ensemble (Lakshminarayanan et al., 2017), a single rank-$r$ model achieves competitive predictive performance with significantly reduced computational cost. We further demonstrate improved out-of-distribution detection and selective prediction capabilities than MFVI and deterministic-backbone with low-rank perturbations(Dusenberry et al., 2020a) (see Section 4).

Our work establishes that low-rank variational inference is not only a computational convenience but a principled approach with provable benefits, enabling practical Bayesian deep learning across modern architectures.

## 2. Methods

### 2.1. Low-Rank Variational Inference

We parameterize each weight matrix $W \in \mathbb{R}^{m \times n}$ via a low-rank factorization $W = AB^\top$ where $A \in \mathbb{R}^{m \times r}$ and

---

[1]Throughout this paper, "singular" refers to measure-theoretic singularity of the induced posterior on weight space with respect to Lebesgue measure, arising because it is supported on the rank-$r$ manifold, rather than to the broader asymptotic notion of singular models studied in Watanabe-style singular learning theory (Watanabe, 2009; Watanabe & Opper, 2010; Watanabe, 2013).

$B \in \mathbb{R}^{n \times r}$. We place independent priors on the factors, $p(A, B) = p_A(A) \, p_B(B)$, and employ a mean-field variational posterior over the factors, $q(A, B) = q_A(A) \, q_B(B)$. The weight distribution is induced through the transformation $(A, B) \mapsto AB^\top$. This concentrates the posterior support on the rank-$r$ matrices manifold, whose geometric and generalization properties we analyze in Section 3. We employ a scale mixture prior following Blundell et al. (2015):
$$p_A(A) = \prod_j \left[ \pi \mathcal{N}(a_j \mid 0, \sigma_1^2) + (1 - \pi) \mathcal{N}(a_j \mid 0, \sigma_2^2) \right],$$
and similarly for $p_B(B)$, where $\mathcal{N}(x \mid \mu, \sigma^2)$ denotes the Gaussian density with mean $\mu$ and variance $\sigma^2$, $j$ indexes the $mr$ elements of $A$ (and $nr$ elements of $B$), and $\sigma_1^2 \gg \sigma_2^2$ creates a heavy-tailed prior that encourages sparse structure while allowing occasional large weights. The variational posteriors are mean-field Gaussians: $q_A(A; \theta_A) = \prod_{j=1}^{mr} \mathcal{N}(a_j \mid \mu_{A,j}, \sigma_{A,j}^2)$ and $q_B(B; \theta_B) = \prod_{j=1}^{nr} \mathcal{N}(b_j \mid \mu_{B,j}, \sigma_{B,j}^2)$, where $\theta_A = \{\mu_A, \sigma_A\}$ and $\theta_B = \{\mu_B, \sigma_B\}$ are the variational parameters learned through backpropagation.

### 2.2. Evidence Lower Bound

We derive the ELBO for our factorized parameterization. Starting from the marginal log-likelihood and applying Jensen's inequality: $\log p(\mathcal{D}) \geq \mathbb{E}_{q(A,B)} \left[ \log \frac{p(\mathcal{D}, A, B)}{q(A, B)} \right] =: \mathcal{L}(q)$. Using $p(\mathcal{D}, A, B) = p(\mathcal{D} \mid AB^\top) p_A(A) p_B(B)$ and $q(A, B) = q_A(A) q_B(B)$:

$$\mathcal{L}(q) = \underbrace{\mathbb{E}_{q_A q_B}[\log p(\mathcal{D} \mid AB^\top)]}_{\text{data fit}} - \underbrace{\beta(\text{KL}(q_A \| p_A) + \text{KL}(q_B \| p_B))}_{\text{regularization}} \cdot \tag{1}$$

where $\beta$ is the KL scale (or temperature parameter) that controls the strength of the regularization penalty. The key insight is that the KL divergence decomposes into two independent terms enabling efficient parallel computation. All three terms in Eq. (1) are estimated via Monte Carlo. For the expected log-likelihood, we sample $A \sim q_A$ and $B \sim q_B$, form $W = AB^\top$, and evaluate $\log p(\mathcal{D} \mid W)$. The KL terms are computed similarly via sampling, as the scale mixture prior precludes closed-form computation.

### 2.3. Gradient Estimation and Optimization

We optimize the ELBO (1) using gradient-based methods with the reparameterization trick (Kingma & Welling, 2014). Following the Bayes by Back-propagation algorithm from Blundell et al. (2015), we parameterize the standard deviations as $\sigma = \log(1 + \exp(\rho))$ to ensure positivity. To sample from the posteriors, we draw $\epsilon_A, \epsilon_B \sim \mathcal{N}(0, I)$ and compute: $A = \mu_A + \log(1 + \exp(\rho_A)) \circ \epsilon_A$, $B = \mu_B + \log(1 + \exp(\rho_B)) \circ \epsilon_B$ where $\circ$ denotes elementwise multiplication. This makes sampling differentiable with respect to the variational parameters, allowing gradients to flow through Monte Carlo estimates of the ELBO. We use Adam optimizer (Kingma & Ba, 2015) for all models.

### 2.4. Architecture-Specific Implementations

We apply our factorization to MLPs, LSTMs, and Transformers.

**Multilayer Perceptrons.** For fully connected layers $W_\ell \in \mathbb{R}^{d_\ell \times d_{\ell+1}}$, we factorize as $W_\ell = A_\ell B_\ell^\top$ where $A_\ell \in \mathbb{R}^{d_\ell \times r_\ell}$ and $B_\ell \in \mathbb{R}^{d_{\ell+1} \times r_\ell}$. Each layer maintains its own rank $r_\ell$, tunable independently or uniformly. This requires no architecture-specific modifications.

**Transformers.** We factorize query, key, value projections ($\mathbf{W}_Q, \mathbf{W}_K, \mathbf{W}_V \in \mathbb{R}^{d_{\text{model}} \times d_{\text{model}}}$) and feed-forward weights ($\mathbf{W}_1 \in \mathbb{R}^{d_{\text{model}} \times d_{\text{ff}}}$, $\mathbf{W}_2 \in \mathbb{R}^{d_{\text{ff}} \times d_{\text{model}}}$). Since these are linear operations, we use the same variational layers as MLPs. For embeddings $W_{emb} \in \mathbb{R}^{V \times d}$, we exploit batch sparsity: only rows of $A$ for tokens in the current batch are sampled, reducing cost from $\mathcal{O}(Vd)$ to $\mathcal{O}(|U|r + dr)$ where $|U|$ is unique tokens. Independent factors per head preserve multi-head structure.

**LSTMs.** We factorize input-to-hidden ($\mathbf{W}_{ih} \in \mathbb{R}^{d_{\text{in}} \times 4h}$) and hidden-to-hidden ($\mathbf{W}_{hh} \in \mathbb{R}^{h \times 4h}$) matrices, where the factor of 4 accounts for input, forget, output, and cell gates. Following Fortunato et al. (2019), factors $A$ and $B$ are sampled once per batch, $W = AB^\top$ is cached across time steps, then resampled for the next batch. This ensures KL divergence is computed once per sequence rather than per time step, maintaining the correct variational objective.

## 3. Theoretical Analysis

We now establish the theoretical foundations of our approach. We show that the factorization $W = AB^\top$ induces a different posterior geometry than mean-field methods (Section 3.1), captures structured weight correlations (Section 3.2), provides bounded approximation error under optimal rank selection (Section 3.3), and yields provably tighter generalization bounds (Section 3.4).

### 3.1. The Induced Posterior: Geometry and Singularity

When we parameterize the posterior over factors $A$ and $B$ rather than directly over $W$, the resulting distribution $q_W$ on weight matrices has a geometric structure determined by the factorization constraint. We formalize this through the induced posterior and characterize its support.

**Definition 3.1** (Induced Posterior). Let $A$ and $B$ be drawn from the variational posterior $q(A, B)$. The **induced posterior** (pushforward measure (Bogachev, 2007)) $q_W$ is the distribution of the random variable $W = AB^\top$.

**Lemma 3.2** (Constrained Support ). *Let $\mathcal{R}_r \subset \mathbb{R}^{m \times n}$ be the set of matrices with rank at most $r$. $\mathcal{R}_r = \{W \in \mathbb{R}^{m \times n} \mid \text{rank}(W) \leq r\}$ and let $W = AB^\top$ where $A \in \mathbb{R}^{m \times r}$ and $B \in \mathbb{R}^{n \times r}$. Then: (i) $\text{rank}(W) \leq r$ and (ii) For any variational posterior $q(A, B)$, $q_W$, satisfies:*

$q_W(\mathcal{R}_r) = 1$ *(see Appendix B.4 for complete proof)*

**Lemma 3.3** (Measure Zero of $\mathcal{R}_r$). *The set $\mathcal{R}_r$ has Lebesgue measure zero in $\mathbb{R}^{m \times n}$ when $r < \min(m, n)$. (see Appendix B.5 for complete proof)*

**Theorem 3.4** (**Singularity of the Induced Posterior**). *Let $r < \min(m, n)$ and let $q(A, B)$ be any variational distribution over factors $A \in \mathbb{R}^{m \times r}$ and $B \in \mathbb{R}^{n \times r}$. Then the induced posterior $q_W$ is **singular** with respect to Lebesgue measure on $\mathbb{R}^{m \times n}$.*

*Proof.* We use the standard measure-theoretic criterion for singularity: a measure $\mu$ is singular with respect to a measure $\nu$, written $\mu \perp \nu$, if there exists a measurable set $A$ such that $\mu(A) = 1$ and $\nu(A) = 0$.

**(i)** From Lemma 3.2, we have already established that: $q_W(\mathcal{R}_r) = 1$, which means the induced posterior $q_W$ is entirely supported on the low-rank manifold $\mathcal{R}_r$.

**(ii)** From Lemma 3.3, we have that the Lebesgue measure of this manifold is zero: $\lambda(\mathcal{R}_r) = 0$.

Taking $A = \mathcal{R}_r$, conditions (i) and (ii) imply directly that $q_W \perp \lambda$. Therefore, the induced posterior $q_W$ is singular with respect to Lebesgue measure on $\mathbb{R}^{m \times n}$.

**Intuition.** The singularity statement immediately implies that $q_W$ cannot admit a density with respect to Lebesgue measure. Suppose for the sake of contradiction that $q_W$ admits a probability density function $f : \mathbb{R}^{m \times n} \to [0, \infty)$ with respect to the Lebesgue measure $\lambda$. By the definition of a density, the probability of the set $\mathcal{R}_r$ would be given by the integral of the density over that set: $q_W(\mathcal{R}_r) = \int_{\mathcal{R}_r} f(W) \, d\lambda(W)$. Since the domain of integration has measure zero ($\lambda(\mathcal{R}_r) = 0$), standard integration theory tells us this integral must be zero: $\int_{\mathcal{R}_r} f(W) \, d\lambda(W) = 0$. However, this contradicts our finding in step (i) that $q_W(\mathcal{R}_r) = 1$. Therefore, no such density function $f$ exists with respect to $\lambda$ on $\mathbb{R}^{m \times n}$. $\square$

**Interpretation.** This result reveals an important geometric distinction: standard mean-field posteriors $q(W) = \prod_{ij} \mathcal{N}(w_{ij} | \mu_{ij}, \sigma_{ij}^2)$ have positive density everywhere in $\mathbb{R}^{m \times n}$, while our induced posterior $q_W$ concentrates on the rank-$r$ manifold $\mathcal{R}_r$, which has zero volume in the full weight space (See Figure 2). This geometric constraint impacts inductive bias. As argued by Wilson & Izmailov (2020), generalization in probabilistic models depends on two properties: *the support of the posterior and the inductive biases* it encodes. Mean-field imposes a *bias of independence*, treating weights as freely adjustable parameters. Our factorization imposes a *bias of correlation* by restricting support to $\mathcal{R}_r$, acting as an implicit regularizer: updating $W_{ij} = \sum_{k=1}^{r} A_{ik} B_{jk}$ requires modifying shared factors

affecting entire rows and columns. This prevents *local memorization* and enables *coherent uncertainty propagation*.

### 3.2. Structured Weight Correlations

We now characterize the covariance structure of $W$ and show how the rank parameter $r$ controls the expressiveness of these correlations.

**Lemma 3.5** (Covariance of Weight Entries). *For weight entries $W_{ij} = \sum_{k=1}^{r} A_{ik} B_{jk}$ and $W_{i'j'} = \sum_{\ell=1}^{r} A_{i'\ell} B_{j'\ell}$, under mean-field factorization $q(A, B) = q_A(A) q_B(B)$ with independent entries, we have:*

$$\text{Cov}(W_{ij}, W_{i'j'}) = \sum_{k=1}^{r} \text{Cov}(A_{ik} B_{jk}, A_{i'k} B_{j'k}). \quad (2)$$

*In general, this is nonzero when weights share common latent factors. See Appendix B.7 for complete proof.*

**Implications.** While $A, B$ are under mean-field factorization, enabling us to leverage existing mean field algorithms, entries of W are not independent. Weights $W_{ij}$ and $W_{i'j'}$ that share latent factors (indexed by $k$) exhibit correlated uncertainties, capturing *global structure* in weight space (see Figure 1). This filters high-frequency noise that does not align with the dominant low-rank structure, providing generalization benefits that independent weight posteriors can not capture. The rank $r$ controls this expressiveness: higher rank enables richer correlation patterns while maintaining $O(r(m + n))$ parameters.

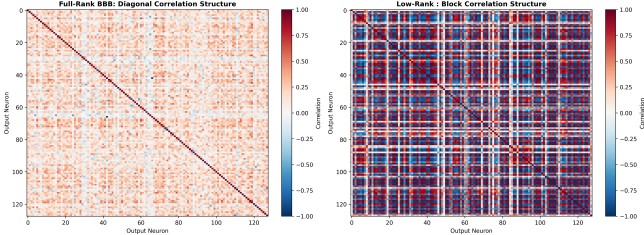

*Figure 1.* **Weight correlation structure.** Comparison of Full-Rank BBB (left) and Low-Rank Gaussian r=15 (right). Full-Rank exhibits diagonal correlations; Low-Rank captures block structure.

### 3.3. Loss Approximation Guarantees

We now establish conditions under which the low-rank model closely approximates the full-rank optimum.

**Assumptions.** We assume the loss function $\ell(\cdot, y)$ satisfies: **(i)** $L$-**Lipschitz**: $|\ell(z, y) - \ell(z', y)| \leq L \|z - z'\|_2$ and **(ii)** **Bounded inputs**: $\|x\|_2 \leq R$ almost surely

These conditions hold for common regression and classification losses (see Appendix C.5 for extended analysis)

Let $W^*$ be the optimal full-rank weight matrix (learned via back-propagation) and $W_r^*$ be the optimal rank-$r$ matrix of

$W^*$ obtained via truncated SVD (Eckart & Young, 1936))

**Theorem 3.6** (Bounded Loss Approximation ). *Under the assumptions above, the expected loss difference is bounded:*

$$|\mathbb{E}[\ell(W^*x, y)] - \mathbb{E}[\ell(W_r^*x, y)]| \leq LR\sqrt{\sum_{i=r+1}^{\rho} \sigma_i^2(W^*)}, \quad (3)$$

where $\sigma_i(W^*)$ are the singular values of $W^*$ and $\rho =$ rank$(W^*)$. **In practice, our learned matrix $W = AB^\top$ may not achieve the optimal rank-$r$ approximation** $W_r^*$ **of** $W^*$. Thus, we decompose the total approximation error into two components.

**Theorem 3.7** (Decomposition of Approximation Error). *Let $W = AB^\top$ be optimal learned rank-$r$ matrix and let $\sigma_{>r} := \sqrt{\sum_{i=r+1}^{\rho} \sigma_i^2(W^*)}$ . Then for any $(x, y)$,*

$$|\ell(Wx, y) - \ell(W^*x, y)| \leq LR(\|W - W_r^*\|_F + \sigma_{>r}). \quad (4)$$

See Appendix C.1 for complete proofs and **the Bayesian extension with $\beta$-smooth losses.**

**Implications.** Theorem 3.6 establishes that the optimal rank-$r$ approximation incurs minimal loss when tail singular values $\sigma_{>r}$ decay rapidly. Theorem 3.7 reveals that the total error for our learned factorization $W = AB^\top$ decomposes into two components: (i) **learning error** $\|W - W_r^*\|_F$, which measures how well training finds the optimal rank-$r$ solution and can be reduced through better optimization, and (ii) **rank bias** $\sigma_{>r}$, the unavoidable approximation error from rank restriction. In practice, rank $r$ of each layer can be tuned via ablation studies with reduced computational budget (fewer epochs, fewer MC samples). When pretrained deterministic weights are available, singular value decay analysis can optionally narrow the search range or validate ablation results (See Figures 7, 9, and 15). This connects our theoretical result directly to implementation decisions.

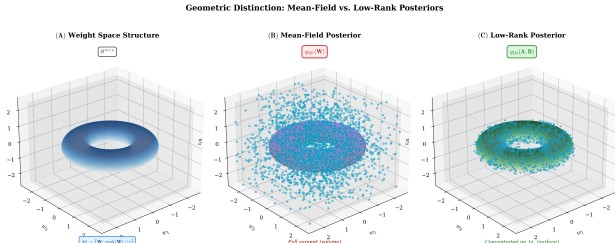

*Figure 2.* **Geometric distinction between mean-field and low-rank posteriors.** 3D projection of weight space $\mathbb{R}^{m \times n}$. Left: Rank-$r$ manifold $\mathcal{M}_r$ (blue surface, dimension $r(m + n - r)$). Middle: Mean-field posterior $q_{\text{MF}}(W)$ has full-dimensional support (volume). Right: Low-rank posterior $q_{\text{LR}}(A, B)$ concentrates on the manifold surface.

## 3.4. PAC-Bayes Generalization Bounds

**Theorem 3.8** (Tighter Bounds for Low-Rank Posteriors). *Consider full-rank mean-field with $D_{full} = mn$ and low-rank factorization with $D_{LR} = r(m+n)$ parameters. The ratio of complexity terms in the PAC-Bayes bound (McAllester, 1998) satisfies:*

$$\frac{Complexity(Q_{LR})}{Complexity(Q_{full})} \simeq \sqrt{\frac{r(m + n)}{mn}} \simeq \sqrt{r\left(\frac{1}{m} + \frac{1}{n}\right)} \ll 1$$
$$(5)$$

*when $r \ll \min(m, n)$. Figure 3 shows the empirical difference. See Appendix D for complete proof.*

## 3.5. Gaussian Complexity Generalization Bounds

While PAC-Bayes bounds (Section 3.4) capture generalization through KL divergence, we provide a complementary perspective using *Gaussian complexity*, a data-dependent measure exploiting the geometric structure of rank-constrained weight matrices. Leveraging recent work on vector-valued Gaussian complexity for deterministic low-rank networks (Pinto et al., 2025), we establish guarantees that explicitly depend on spectral norm bounds and rank constraints.

**Theorem 3.9** (Gaussian-complexity bound for low-rank BNN predictors). *Consider a depth-$D$ network with weights $W_i = A_iB_i^\top$ where $A_i \in \mathbb{R}^{h_i \times r_i}$, $B_i \in \mathbb{R}^{h_{i-1} \times r_i}$ and $r_i \leq \min(h_i, h_{i-1})$. Let the variational posterior factorize as $q(A, B \mid \theta) = \prod_{i=1}^{D} q_A(A_i \mid \theta_i^A)q_B(B_i \mid \theta_i^B)$ and assume $\|A_i\|_2 \leq C_i^A$, $\|B_i\|_2 \leq C_i^B$ almost surely. Define $C_i := C_i^A C_i^B$ and the posterior-mean predictor $f_{\text{BNN}}(x; \theta) = \mathbb{E}_{(A,B) \sim q}[f_{A,B}(x)]$. Let $\phi$ be 1-Lipschitz and let $\ell(\cdot, y) \in [0, 1]$ be $\mathcal{L}$-Lipschitz in its first argument. Then for $S \sim \mathcal{D}^N$, with probability at least $1 - \delta$,*

$$\mathbb{E}[\ell(f_{\text{BNN}}(x; \theta), y)] \leq \frac{1}{N}\sum_{j=1}^{N} \ell(f_{\text{BNN}}(x_j; \theta), y_j) +$$
$$\sqrt{\pi}\mathcal{L}\,\hat{\mathcal{G}}_S(\mathcal{F}_D^{\text{Pinto}}(C, r)) + 3\sqrt{\frac{\log(2/\delta)}{2N}}, \quad (6)$$

*where $\mathcal{F}_D^{\text{Pinto}}(C, r)$ is the deterministic class of depth-$D$ networks whose layers satisfy $\|W_i\|_2 \leq C_i$ and rank$(W_i) \leq r_i$. Moreover, $\hat{\mathcal{G}}_S(\mathcal{F}_D^{\text{Pinto}}(C, r))$ admits the explicit bound of (Pinto et al., 2025)(Thm.7). See Appendix E for complete proof .*

**Interpretation.** Although Gaussian complexity is not the canonical Bayesian tool (PAC-Bayes is), it offers insight into capacity control: low-rank layers impose geometric constraints that reduce complexity beyond parameter-count arguments. As shown in Figure 3, the Gaussian complexity bounds are vacuous for both methods and do not provide practical generalization certificates. However, they formalize how rank constraints and spectral norms jointly control model capacity, suggesting that the rank-induced restric-

tions contribute to the good generalization often observed in low-rank neural networks.

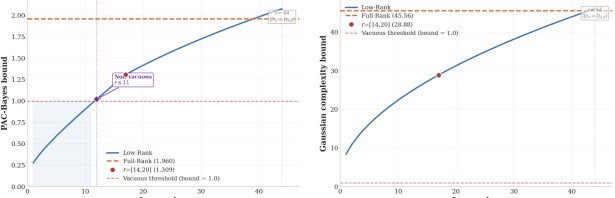

*Figure 3.* **Empirical generalization bounds for low-rank Bayesian neural networks.** PAC-Bayes **(left)** and Gaussian complexity **(right)** bounds use empirical values from trained LSTM model and training data. PAC-Bayes bound exhibits critical rank $r^* \approx 11$ transitioning from non-vacuous to vacuous ($> 1$, dashed red line). Gaussian complexity decreases from 45.56 (full-rank) to 18.97 with rank reduction. Full-Rank Bayesian models yield vacuous PAC-Bayes bounds regardless of configuration.

## 4. Experiments

We compare six approaches: **(i)** Deterministic baseline, **(ii)** Deep Ensemble (Lakshminarayanan et al., 2017) ($M = 5$), **(iii)** Full-Rank BBB (Blundell et al., 2015), **(iv)** Low-Rank (random init), **(v)** Low-Rank SVD init (see Appendix F.9), and **(vi)** Rank-1 multiplicative (Dusenberry et al., 2020a). Low-Rank methods (iv-v) use factorization $W = AB^\top$ with rank $r$. See Appendix G for all implementation details.

### 4.1. Evaluation Metrics

Predicted probabilities are Monte Carlo averages over posterior weight samples. For classification, we report Accuracy, AUROC (discrimination), NLL and ECE (calibration). For regression, we report MAE (accuracy), PICP (calibration). Uncertainty is quantified via mutual information (MI) (Houlsby et al., 2011) or predictive std: MI Ratio (OOD vs in-domain mean MI), AUPR-Err (error detection), and AUPR-Succ (selective prediction). OOD detection uses AUROC-OOD, AUPR-OOD (OOD as positive), and AUPR-In (in-domain as positive). Params denotes total trainable parameters. For Bayesian methods, this includes both posterior means and uncertainty parameters; for Deep Ensembles, counts are summed across members.

For each architecture, we select the rank $r$ via ablation studies with reduced computational budget (fewer epochs and Monte Carlo samples), and when deterministic baselines are available, we further validate the chosen ranks using singular value decay analysis of the corresponding weight matrices (See Appendix F.9 and Figures 7, 9, 15).

### 4.2. MIMIC-III: ICU Mortality Prediction

We first evaluate our method on the MIMIC-III clinical database (Johnson et al., 2016) to predict binary ICU mor-

tality from patient vitals and laboratory values. The dataset comprises 40,406 training and 4,490 test samples with 44 clinical features extracted from the first 48 hours of ICU stays (8.4% mortality rate). For out-of-distribution evaluation, we test on 5,357 newborn ICU records (1.1% mortality), representing significant distributional shift from the adult training population. All methods use a 2-layer MLP with hidden dimension 128. Low-Rank methods used $r = 15$. Complete experimental details are provided in Appendix G.3.

**Results.** Table 1 and Figure 4 show Low-Rank Gaussian (rank $r = 15$) achieves competitive in-domain discrimination (AUROC=0.895) while outperforming both Deep Ensemble and Full-Rank BBB on OOD detection: AUC-OOD=**0.802** versus 0.738 (Deep Ens.) and 0.770 (Full-Rank), AUPR-OOD=**0.788** versus 0.754 and 0.759, and AUPR-In=**0.824** versus 0.721 and 0.807 respectively. Figure 4 visualizes this tradeoff: Low-Rank Gaussian performs well on uncertainty quantification metrics which leads to better OOD detection, while Deep Ensemble maintains superior in-domain discrimination (AUROC=0.929) and calibration (NLL=0.300 versus 0.433 for Low-Rank). This suggests a calibration-sharpness tradeoff where the low-rank model prioritizes epistemic uncertainty estimation over likelihood-based calibration. Low-Rank Gaussian uses 70% fewer parameters than Full-Rank BBB and 88% fewer than Deep Ensemble while achieving best-in-class OOD detection. See Appendix H.2 for extended results and analysis.

*Table 1.* MIMIC-III evaluation metrics averaged over 5 independent runs. Best in **bold**, second underlined. Arrows indicate optimization direction: ↑ (higher better), ↓ (lower better).

| Model | AUROC↑ | AUPR-Err↑ | AUC-OOD↑ | AUPR-OOD↑ | AUPR-In↑ | NLL↓ | Params↓ |
|---|---|---|---|---|---|---|---|
| Deterministic | $.922_{\pm.003}$ | $.145_{\pm.035}$ | $.500_{\pm.000}$ | $.544_{\pm.000}$ | $.456_{\pm.000}$ | $.284_{\pm.017}$ | $\underline{22.4k}$ |
| Deep Ens. | $\mathbf{.929}_{\pm.002}$ | $.237_{\pm.035}$ | $.738_{\pm.061}$ | $.754_{\pm.048}$ | $.721_{\pm.053}$ | $\underline{.300}_{\pm.021}$ | 112k |
| Full-Rank | $.895_{\pm.004}$ | $.412_{\pm.029}$ | $.770_{\pm.051}$ | $.759_{\pm.055}$ | $\underline{.807}_{\pm.035}$ | $.401_{\pm.033}$ | 44.8k |
| Low-Rank (ours) | $.895_{\pm.001}$ | $\mathbf{.540}_{\pm.027}$ | $\mathbf{.802}_{\pm.018}$ | $\mathbf{.788}_{\pm.034}$ | $\mathbf{.824}_{\pm.013}$ | $.433_{\pm.020}$ | $\mathbf{13.6k}$ |
| LR-SVD (ours) | $\underline{.898}_{\pm.001}$ | $.486_{\pm.155}$ | $.713_{\pm.069}$ | $.719_{\pm.046}$ | $.738_{\pm.074}$ | $.452_{\pm.035}$ | $\mathbf{13.6k}$ |
| Rank-1 | $.901_{\pm.010}$ | $.322_{\pm.019}$ | $.705_{\pm.073}$ | $.742_{\pm.062}$ | $.663_{\pm.080}$ | $.317_{\pm.024}$ | 23.3k |

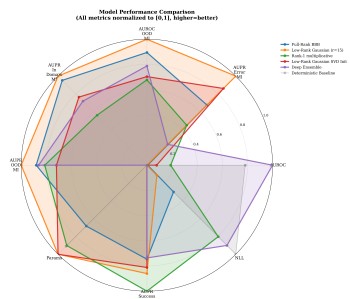

*Figure 4.* **Model comparison on MIMIC-III (averaged across 5 seeds).** Low-Rank Gaussian r=15 (orange) achieves superior OOD detection. Deep Ensemble maintains better calibration and in-domain discrimination. Rank-1 multiplicative (green) achieves better calibration but weaker OOD detection. Full-Rank BBB (blue) shows balanced but moderate performance across metrics.

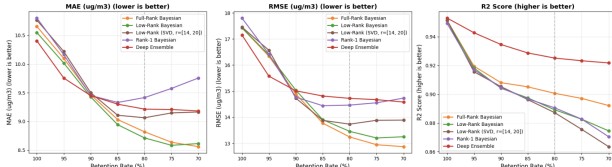

*Figure 5.* **Selective prediction on Beijing Air Quality LSTM.**
While Deep Ensemble (red) achieves best point predictions at
100% retention, Bayesian methods outperform when discarding
the most uncertain samples. Low-Rank (green) achieves largest
improvement (17.4% MAE reduction at 80% retention), demon-
strating superior uncertainty quality for selective prediction.

## 4.3. Beijing Air Quality: Time Series Forecasting

Next we evaluate our approach on the Beijing Multi-Site
Air Quality dataset (Chen, 2015) for $PM_{2.5}$ forecasting from
meteorological features. Using 24-hour sliding windows,
the dataset has 29,213 training, 6,260 validation, and 6,260
test samples with 15 features. For OOD evaluation, we
use 20,050 samples from Guangzhou (Chen, 2016), a city
with distinct subtropical climate representing distributional
shift from Beijing's continental climate. All methods use
2-layer LSTMs with 64 hidden units. Low-Rank meth-
ods decompose input-to-hidden ($\mathbf{W}_{ih} \in \mathbb{R}^{15 \times 256}$) and
hidden-to-hidden ($\mathbf{W}_{hh} \in \mathbb{R}^{64 \times 256}$) weight matrices as
$\mathbf{W} = \mathbf{A}\mathbf{B}^\top$ with ranks $r = 14$ for the first layer and 20 for
the second. Complete experimental details are provided in
Appendix G.4.

**Results.** Table 2 and Figure 6 show that Low-Rank Bayesian
LSTM achieves the best coverage (PICP=0.790) and near-
best ECE-based calibration (ECE=0.114, second only to
Full-Rank BBB's 0.111) while using 64% fewer parameters
than Full-Rank BBB. For OOD detection, Low-Rank attains
second-best performance (AUROC=0.710, AUPR=0.861)
after Deep Ensemble (0.730, 0.883) despite 86% fewer pa-
rameters. Figure 5 shows Bayesian methods outperform
Deep Ensemble when filtering uncertain samples, with Low-
Rank achieving 17.4% error reduction at 80% retention.
The calibration gap is notable: Low-Rank achieves the high-
est coverage among the compared methods (PICP=0.790)
while Deep Ensemble (0.310) and Rank-1 (0.449) severely
undercover. This confirms that structured weight correla-
tions from low-rank factorization (Section 3.2) propagate
uncertainty more reliably. See Appendix H.3 for extended
results and analysis.

## 4.4. Text Classification with Transformers

Finally, we evaluate our approach on binary sentiment clas-
sification using the Stanford Sentiment Treebank (SST-2)
dataset (Socher et al., 2013), which contains 67,349 train-
ing samples and 872 validation samples of movie reviews.
For out-of-distribution detection, we use the AG News
dataset (Zhang et al., 2015) with 7,600 samples from the

*Table 2.* Beijing Air Quality results averaged over 4 independent
runs. Results are reported as mean ± standard deviation. Best in
**bold**, second underlined. Arrows indicate optimization direction:
↑ (higher better), ↓ (lower better).

| Model | MAE↓ | ECE↓ | PICP↑ | AUROC-OOD↑ | AUPR-OOD↑ | Params↓ |
|---|---|---|---|---|---|---|
| Deterministic | $10.79_{\pm 0.16}$ | – | – | 0.500 | 0.500 | **33K** |
| Full-Rank BBB | $\underline{10.55}_{\pm 0.11}$ | $\mathbf{0.111}_{\pm 0.007}$ | $\underline{0.788}_{\pm 0.012}$ | $0.492_{\pm 0.039}$ | $0.743_{\pm 0.019}$ | 132K |
| Low-Rank (ours) | $10.63_{\pm 0.20}$ | $0.114_{\pm 0.005}$ | $\mathbf{0.790}_{\pm 0.006}$ | $\underline{0.710}_{\pm 0.021}$ | $\underline{0.861}_{\pm 0.022}$ | $\underline{47K}$ |
| LR-SVD (ours) | $10.70_{\pm 0.03}$ | $0.139_{\pm 0.022}$ | $0.773_{\pm 0.022}$ | $0.687_{\pm 0.068}$ | $0.812_{\pm 0.026}$ | $\underline{47K}$ |
| Rank-1 Mult. | $10.80_{\pm 0.15}$ | $0.307_{\pm 0.015}$ | $0.449_{\pm 0.025}$ | $0.580_{\pm 0.046}$ | $0.751_{\pm 0.028}$ | 66K |
| Deep Ensemble (5) | $\mathbf{10.45}_{\pm 0.03}$ | $0.317_{\pm 0.012}$ | $0.310_{\pm 0.021}$ | $\mathbf{0.730}_{\pm 0.029}$ | $\mathbf{0.883}_{\pm 0.012}$ | 330K |

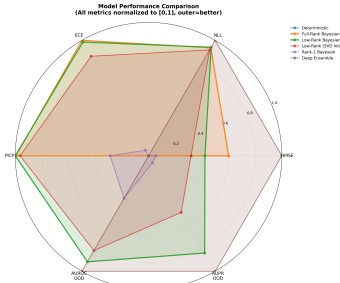

*Figure 6.* **Model comparison on Beijing Air Quality LSTM
(averaged across 4 seeds).** Deep Ensemble (brown) achieves best
OOD detection but poor coverage. Low-Rank BBB (green) shows
best coverage (PICP) and near-best calibration (ECE) with second-
best OOD performance. Low-Rank SVD (red) shows strong OOD
detection with good coverage. Full-Rank BBB (orange) achieves
best calibration but moderate coverage. Rank-1 BBB (purple)
shows poor uncertainty performance.

news domain, which provides a meaningful distributional
shift from the sentiment classification task. We implement a
4-layer Transformer encoder with $d_{\text{model}} = 256$, 4 attention
heads, and feed-forward dimension $d_{\text{ff}} = 512$, using the
BERT tokenizer with vocabulary size 30,522. The architec-
ture was inspired by BERT-mini (Turc et al., 2019). Singular
value decay analysis (Figure 7) shows that the embedding
layer, which represents 70% of the parameter counts for
all models, has particularly rapid decay. Applying rank
$r = 16$ for all layers yields 86% parameter reduction vs.
deterministic and 93% vs. full-rank BBB.

**Results.** Table 3 and Figure 8 show Low-Rank BBB
achieves 0.806 accuracy with best AUPR-In (0.302) and
second-best AUROC-OOD (0.640). Deep Ensemble
achieves best overall accuracy (0.825) and calibration
(NLL=0.434) with higher MI Ratio (1.55 vs 1.35), while
Full-Rank BBB underperforms across metrics (0.752 ac-
curacy, 0.552 NLL), consistent with (Cinquin et al., 2021)
findings. Low-Rank BBB uses 13× fewer parameters than
Full-Rank (1.5M vs 19.8M) and 33× fewer parameters than
Deep Ensemble (49.6M). Training times reflect this effi-
ciency: Low-Rank trains in 8.2 minutes versus 23.1 for
Full-Rank BBB and 64.7 for Deep Ensemble. See Appendix
H.4 for extended results and analysis.

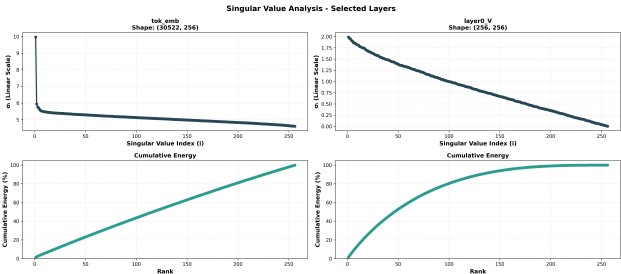

*Figure 7.* Singular value decay for embedding and value projection matrices showing rapid decay.

*Table 3.* SST-2 evaluation metrics averaged over 4 independent runs. Best in **bold**, second underlined. Arrows indicate optimization direction: ↑ (higher better), ↓ (lower better).

| Model | Acc↑ | NLL↓ | AUROC-OOD↑ | MI Ratio↑ | AUPR-In↑ | AUPR-Succ↑ | Params↓ |
|---|---|---|---|---|---|---|---|
| Deterministic | $.812_{\pm.006}$ | $.490_{\pm.003}$ | $.500_{\pm.000}$ | $.000_{\pm.000}$ | $.102_{\pm.000}$ | $.812_{\pm.006}$ | 9.9M |
| Deep Ens. | $\mathbf{.825}_{\pm.006}$ | $\mathbf{.434}_{\pm.023}$ | $\mathbf{.657}_{\pm.009}$ | $\mathbf{1.55}_{\pm.05}$ | $.267_{\pm.016}$ | $\mathbf{.933}_{\pm.003}$ | 49.6M |
| Full-Rank | $.752_{\pm.011}$ | $.552_{\pm.010}$ | $.622_{\pm.019}$ | $1.31_{\pm.07}$ | $.222_{\pm.028}$ | $.888_{\pm.006}$ | 19.8M |
| Low-Rank (ours) | $.806_{\pm.003}$ | $.527_{\pm.006}$ | $.640_{\pm.028}$ | $1.35_{\pm.10}$ | $\mathbf{.302}_{\pm.027}$ | $.917_{\pm.005}$ | 1.5M |
| LR-SVD (ours) | $.800_{\pm.005}$ | $.516_{\pm.002}$ | $.616_{\pm.011}$ | $1.22_{\pm.10}$ | $.272_{\pm.012}$ | $.923_{\pm.003}$ | 1.5M |
| Rank-1 | $.795_{\pm.013}$ | $.507_{\pm.005}$ | $.613_{\pm.015}$ | $1.11_{\pm.09}$ | $.271_{\pm.027}$ | $.931_{\pm.004}$ | 10.0M |

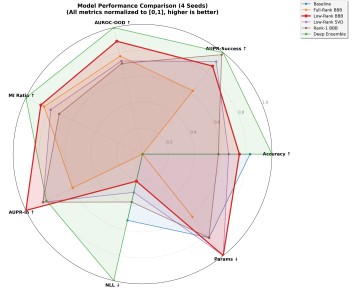

*Figure 8.* **Model comparison on SST-2 Transformer (averaged across 4 seeds).** Deep Ensemble (green) achieves best overall performance. Low-Rank BBB (red) shows good OOD detection and highest epistemic uncertainty discrimination (MI Ratio) among bayesian models. Full-Rank BBB (orange) demonstrates moderate balanced performance. Rank-1 BBB (brown) maintains good calibration but weaker OOD metrics.

### 4.5. Comparison to SWAG

We additionally compare against SWAG (Maddox et al., 2019), a strong scalable posterior baseline, using multi-seed runs on SST-2, MIMIC-III, and Beijing PM$_{2.5}$; full implementation details and dataset-specific results are reported in Appendix H.6 (Tables 18, 19, and 20). Across these settings, SWAG is a relevant baseline, but it does not overturn the central quality-efficiency advantage of our low-rank approach.

On SST-2, SWAG achieves 0.808 accuracy, 0.556 NLL, 0.111 ECE, and 0.613 AUROC-OOD-MI (Table 18). Compared with Low-Rank BBB in Table 3, it is essentially tied on accuracy (0.808 vs. 0.806) but is worse on NLL (0.556 vs. 0.527) and AUROC-OOD-MI (0.613 vs. 0.641), while storing 208.37M parameters compared with 1.47M for the low-rank posterior.

On MIMIC-III, SWAG reaches 0.922 AUROC, 0.359 NLL, 0.190 ECE, and 0.634 AUROC-OOD-MI (Table 19). Relative to the low-rank model in Table 1, SWAG is stronger on in-domain AUROC (0.922 vs. 0.895) and NLL (0.359 vs. 0.433), but remains weaker on the main MI-based OOD-separation metrics, with AUROC-OOD-MI 0.634 and AUPR-OOD-MI 0.680 versus 0.802 and 0.788 for the low-rank model. Thus, SWAG is a credible MIMIC comparator, but it does not overturn the stronger OOD uncertainty separation of the low-rank approach.

On Beijing PM$_{2.5}$, SWAG attains 10.542 MAE, 17.313 RMSE, 4.323 NLL, 0.216 ECE, and 0.973 PICP (Table 20). Compared with the low-rank model in Table 2, it is competitive on point prediction and achieves substantially higher interval coverage (0.973 vs. 0.790), but this comes with much wider intervals (MPIW 87.0) and weaker OOD detection (AUROC-OOD 0.585 vs. 0.710; AUPR-OOD 0.783 vs. 0.861), while using 1.18M parameters compared with 47K for the low-rank model.

### 4.6. Experimental Insights

**Calibration–OOD Detection Tradeoff.** We observe a more nuanced tradeoff than a single uniform pattern: on MIMIC-III, low-rank improves OOD detection despite weaker NLL than Deep Ensembles; on SST-2, Deep Ensembles remain stronger on both NLL and AUROC-OOD, while low-rank stays competitive at much lower parameter and training cost; and on Beijing, the main low-rank advantage is better calibration, coverage, and selective prediction rather than better OOD AUROC. We hypothesize that the rank constraint maintains broader uncertainty distributions that can improve some reliability metrics at the cost of predictive sharpness, but the empirical manifestation of this tradeoff depends on the task. This pattern favors low-rank in settings where useful uncertainty for abstention, coverage, or OOD awareness matters more than marginal likelihood gains.

**Computational Efficiency.** Parameter reduction on small or medium scales architecture provides substantial memory savings, enabling deployment in resource-constrained settings, but minimal training speedups (Table 10). However, efficiency gains emerge at scale on the larger Transformer architecture: our low-rank approaches train in 7.95 minutes (SVD init) and 8.22 minutes (random init) versus 23.1 for Full-Rank BBB and 64.7 for Deep Ensembles (Table 15). A controlled matched profiling benchmark on SST-2 confirms that these gains are not only artifacts of full training pipelines (Table 4; Appendix G.5.4, Appendix H.4.1). Under this benchmark, Low-Rank BBB retains large advantages in parameter count and peak GPU memory, is substantially faster per epoch than a 5-member Deep Ensemble, and remains slightly faster than Full-Rank BBB. A recovered low-rank rank-sweep further shows that runtime

and memory improve more gradually than parameter count, but still move consistently in the favorable direction as rank decreases (Appendix H.4.2). Additional appendix comparisons with multi-seed SWAG (Maddox et al., 2019) support the same conclusion: while SWAG is a relevant baseline, across SST-2, MIMIC-III, and Beijing it does not overturn the central quality-efficiency advantage of our low-rank approach (Appendix H.6).

*Table 4.* Controlled matched profiling benchmark on SST-2. Peak memory and epoch time are measured under a fixed-step `train_on_batch` profiling loop; full protocol is given in Appendix G.5.4.

| Model | Params | Peak Mem. | Epoch Time |
|---|---|---|---|
| Low-Rank BBB | 1.47M | 357.5MB | 5.88s |
| Full-Rank BBB | 19.84M | 721.1MB | 6.45s |
| Deep Ensemble | 49.61M | 670.1MB | 18.99s |

**Hyperparameter Sensitivity.** The KL weight $\beta$ and rank $r$ require tuning. We recommend starting with $\beta = 1/N_{\text{train}}$ or $\beta = 1/N_{\text{batches}}$ depending on layer implementation, then adjusting based on validation performance. Higher $\beta$ may improve OOD detection but degrade in-distribution metrics; we use KL annealing for stability. Rank $r$ impacts the calibration-OOD tradeoff (see Figures 10, 13, 14) and can be tuned via ablation and singular value analysis when pretrained weights are available.

**SVD Initialization.** SVD warm-starting yields modest gains on specific metrics (MIMIC-III AUROC: 0.898 vs 0.895; SST-2 AUPR-Succ: 0.923) but shows no consistent benefit across tasks. On Beijing LSTMs, random initialization performs better overall. SVD initialization is only worthwhile when pretrained weights already exist, as the improvements do not warrant training an additional deterministic model. See Appendix H.1 for detailed analysis.

## 5. Discussion and Conclusion

We presented a low-rank variational inference framework for Bayesian neural networks that parameterizes weights as $W = AB^\top$ with learned posteriors on the factors. This induces a posterior singular with respect to Lebesgue measure, concentrating on the rank-$r$ manifold, a geometric property distinguishing our approach from mean-field VI, low-rank covariance approximations, and perturbation-based methods. The factorization reduces parameters from $O(mn)$ to $O(r(m+n))$ while capturing structured weight correlations through shared latent factors.

Our theoretical analysis established tighter PAC-Bayes bounds scaling with rank, loss decomposition via Eckart-Young-Mirsky, and complexity transfer from deterministic to Bayesian predictors. Empirically, the method achieved competitive predictive performance across MLPs, LSTMs, and Transformers.

On MIMIC-III, low-rank attained superior OOD detection; on SST-2, it achieved second-best OOD performance after Deep Ensembles while substantially outperforming Full-Rank BBB and Rank-1 baselines. On Beijing Air Quality (LSTM), low-rank achieved best prediction interval coverage and second-best OOD detection after Deep Ensembles. We observed a recurring but task-dependent calibration–OOD tradeoff: the rank constraint can maintain broader epistemic uncertainty at the cost of predictive sharpness, favoring applications where reliable uncertainty matters more than marginal likelihood gains.

Several limitations exist. The calibration gap relative to ensembles may limit applicability in likelihood-focused settings. However, a supplementary SST-2 low-rank ensembling study (Appendix H.5) shows that a 5-member low-rank Bayesian ensemble substantially improves calibration and OOD separation relative to a single low-rank model, supporting ensembling as a practical mitigation path at transformer scale.

Architecture choices reflect our computational constraints, not method limitations. In fact, computational benefits emerge primarily at larger scales where GPU parallelism can be exploited.

Rank and KL weight require tuning, and while improvements are directionally consistent, some confidence intervals overlap across seeds. Appendix H.6 adds multi-seed SWAG comparisons across our three main benchmarks; broader comparisons with low-rank covariance methods would further strengthen the positioning of this work.

Our method is agnostic to the distributional family of the posterior. While we employ Gaussian posteriors for tractability, the factorization $W = AB^\top$ naturally extends to other location-scale distributions such as the Laplace distribution, which provides heavier tails that may improve uncertainty quantification. Additional experiments using a Laplace posterior are presented in Appendix I.

Future work includes adaptive rank selection and scaling to larger language models. We are exploring spike-and-slab priors that enforce sparsity while permitting large weights where needed; however, an in-depth analysis of prior choice and its effect on performance across different data regimes and tasks would provide valuable guidance for practitioners. The framework also extends naturally to computer vision, including medical imaging and mineral prospectivity mapping. The singular posterior geometry provides both theoretical foundations and practical benefits for uncertainty quantification, establishing low-rank factorization as a principled path toward scalable Bayesian deep learning.

## Impact Statement

This paper presents work whose goal is to advance the field of Machine Learning. There are many potential societal consequences of our work, none of which we feel must be specifically highlighted here.

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

## Guide to Appendix Proofs

This appendix provides complete proofs for all theoretical results. The table below maps main paper statements to their appendix locations.

| Main Paper | Statement | Appendix Location |
|---|---|---|
| *Related Work and background* | | |
| – | Comparison with related low-rank approaches | Section A |
| *Section 3.1: Geometry and Singularity* | | |
| Definition 3.1 | Induced posterior | Defined in main text |
| Lemma 3.2 | Constrained support | Section B.4 |
| Lemma 3.3 | Measure zero of $\mathcal{R}_r$ | Section B.5 |
| Theorem 3.4 | Posterior singularity | Proved in main text |
| *Section 3.2: Structured Correlations* | | |
| Lemma 3.5 | Covariance structure | Section B.7 |
| *Section 3.3: Loss Approximation* | | |
| Theorem 3.6 | Optimal rank-$r$ approximation | Section C.2 |
| Theorem 3.7 | Learned approximation | Section C.3 |
| – | Eckart-Young-Mirsky theorem | Section C.1.2 |
| – | Bayesian extension ($\beta$-smooth) | Section C.4 |
| – | Loss function verification | Section C.5 |
| *Section 3.4: PAC-Bayes Bounds* | | |
| Theorem 3.8 | Low-rank PAC-Bayes bounds | Section D |
| *Section 3.5: Gaussian Complexity* | | |
| Theorem 3.9 | Gaussian complexity transfer | Section E |
| – | High-probability alternatives | Section E.10 |
| *Initialization Theory* | | |
| – | Variance-preserving initialization | Section F |
| – | SVD initialization from pretrained weights | Section F.9 |
| *Section 4: Experimental Details* | | |
| – | Hardware, software, and metrics | Section G |
| – | MIMIC-III mortality prediction | Section G.3 |
| – | Beijing PM2.5 time-series forecasting | Section G.4 |
| – | SST-2 sentiment classification | Section G.5 |
| – | SST-2 controlled profiling protocol | Section G.5.4 |
| *Extended Results from Main Experiments* | | |
| – | Experimental insights and practical considerations | Section H.1 |
| – | MIMIC-III additional results | Section H.2 |
| – | Beijing PM2.5 additional results | Section H.3 |
| – | SST-2 additional results | Section H.4 |
| – | SST-2 controlled profiling benchmark | Section H.4.1 |
| – | SST-2 low-rank rank-sweep profiling | Section H.4.2 |
| – | Supplementary SST-2 low-rank ensembling study | Section H.5 |
| – | Supplementary SWAG comparisons | Section H.6 |
| *Additional Experiments* | | |
| – | MNIST classification | Section I.1 |
| – | Fashion-MNIST classification | Section I.2 |
| – | Toy regression | Section I.3 |

**Organization.** Sections are ordered to minimize forward references: comparison with related low-rank approaches (Section A), background and proofs for geometric properties (Section B), loss approximation guarantees (Section C), PAC-Bayes generalization bounds (Section D), Gaussian complexity transfer (Section E), initialization theory (Section F), experimental details (Section G), extended results from main experiments (Section H), and additional experiments including MNIST, Fashion-MNIST, and toy regression (Section I).

# A. Comparison with Related Low-Rank Approaches

*Table 5.* Comparison with related low-rank approaches for BNNs. Our method is the only approach that (i) learns both backbone and uncertainty end-to-end without pretrained models, (ii) reduces both mean and variance parameterization, and (iii) induces a geometrically singular posterior.

| Method | Backbone training | Parameter count (mean + var) | Posterior support | Weight correlations |
|---|---|---|---|---|
| Mean-field VI (Blundell et al., 2015) | Learned Bayes. | $2mn$ | Full $\mathbb{R}^{m \times n}$ | Independent |
| Low-rank cov. (Swiatkowski et al., 2020; Tomczak et al., 2020) | Learned Bayes. | $mn + r(m + n)$ | Full $\mathbb{R}^{m \times n}$ | Via covariance |
| Rank-1 mult. (Dusenberry et al., 2020a) | Learned det. + pert. | $mn + 2(m + n)$ | Full $\mathbb{R}^{m \times n}$ | Rank-1 structure |
| Bella (Doan et al., 2025) | Frozen + adapt | $n_{particles} \times r(m + n)$ | Affine subspace | Via $\Delta\theta_i$ |
| LoRA + Bayes (Yang et al., 2024) | Frozen + adapt | $r(m + n)$ | Affine subspace | Via $\Delta W$ |
| **Ours** | Learned Bayes. | $2r(m + n)$ | Rank-$r$ manifold $\mathcal{R}_r$ | Via shared factors |

## A.1. Mean-Field Variational Inference (Blundell et al., 2015)

Mean-field variational inference places independent Gaussian posteriors over each weight: $q(W) = \prod_{ij} \mathcal{N}(w_{ij}|\mu_{ij}, \sigma_{ij}^2)$. This requires $\mathcal{O}(2mn)$ variational parameters per layer (mean and variance for each of $mn$ weights) and imposes an independence assumption: weight uncertainties are decoupled.

Our method differs in several respects. We place distributions directly over the factor matrices $A$ and $B$, optimizing $\mathcal{L}(q) = \mathbb{E}_{q_A q_B}[\log p(\mathcal{D}|AB^\top)] - \mathrm{KL}(q_A\|p_A) - \mathrm{KL}(q_B\|p_B)$ with only $\mathcal{O}(r(m + n))$ variational parameters. The induced posterior $q_W$ concentrates on the rank-$r$ manifold $\mathcal{R}_r$ (Theorem 3.4, Lemma 3.2), which has measure zero in $\mathbb{R}^{m \times n}$. In contrast, mean-field has full-dimensional support.

Additionally, our factorization induces non-trivial covariances between weights sharing latent factors (Lemma 3.5), capturing structured dependencies. Mean-field assumes complete independence. The rank constraint prevents overfitting to individual examples by forcing shared factor modifications rather than allowing arbitrary independent weight adjustments (Section 3.1).

## A.2. Low-Rank Covariance Approximations (Swiatkowski et al., 2020), (Tomczak et al., 2020)

Low-rank covariance methods maintain full mean parameters $\mu_W \in \mathbb{R}^{m \times n}$ but approximate the posterior covariance via low-rank factors: $\mathrm{Cov}(W) \approx LL^\top$ where $L \in \mathbb{R}^{mn \times r}$. This requires $\mathcal{O}(mn + r(m + n))$ parameters.

Our approach differs in parameterization and support. These methods learn the full mean $\mu_W$ independently and use low-rank factors solely to approximate off-diagonal covariances. We jointly parameterize both mean and uncertainty through the same factorization $W = AB^\top$. Their posterior retains full-dimensional support in $\mathbb{R}^{m \times n}$ with only the covariance structure constrained, whereas our support is restricted to $\mathcal{R}_r$, a lower-dimensional manifold.

The parameter count difference is substantial: they scale as $mn + r(m + n)$ due to storing the full mean vector, while we achieve $r(m + n)$ by encoding both mean and covariance structure in the factorization. This translates to reduced memory requirements and more efficient gradient computation.

## A.3. Rank-1 Multiplicative Perturbations (Dusenberry et al., 2020a)

Rank-1 Bayesian neural networks learn a deterministic weight matrix $W_0$ and scale it by multiplicative rank-1 factors: $W = (1 + rs^\top) \odot W_0$, where $r \in \mathbb{R}^m$ and $s \in \mathbb{R}^n$ are learned stochastically. The posterior is placed over these perturbation vectors, requiring $\mathcal{O}(2(m + n))$ additional variational parameters for means and variances on $r, s$.

Several distinctions emerge. First, rank-1 methods learn end-to-end on a deterministic backbone, whereas we place uncertainty on both factors from initialization without a fixed mean weight matrix. Second, they parameterize $q(r, s)$ with $W$ emerging through element-wise multiplicative perturbations, while we directly parameterize $q_A(A)q_B(B)$ with $W = AB^\top$ emerging through matrix multiplication.

The induced posteriors exhibit different uncertainty structures: theirs has multiplicative structure (perturbations scale deviations from $W_0$), while ours has additive low-rank structure (sums of rank-1 components). This affects uncertainty propagation. Furthermore, rank-1 methods are restricted to scaling-based perturbations, whereas our factorization permits arbitrary rank-$r$ matrices when $r > 1$, enabling richer expressiveness (Lemma 3.5).

## A.4. Bayesian LoRA via SVGD ([Doan et al., 2025](#))

Bella adapts a frozen pretrained backbone via low-rank LoRA factors and applies Stein Variational Gradient Descent (SVGD) over an ensemble of rank-$r$ particles. Each particle $i$ has weights $\theta_i = \theta_0 + B_i A_i^\top$, where $\theta_0$ is frozen and only $(A_i, B_i)$ are updated. This requires $\mathcal{O}(r(m + n))$ parameters per particle, totaling $n_{\text{particles}} \times r(m + n)$.

This approach differs from ours in training scheme and posterior representation. Bella is post-hoc, requiring a pretrained backbone, whereas ours learns end-to-end from random initialization. Bella uses SVGD (kernel-based repulsion over discrete particles) to approximate the posterior over adapters; we employ mean-field variational inference over factors. SVGD maintains $n_{\text{particles}}$ distinct solutions; variational inference maintains a continuous distribution.

Regarding parameterization, Bella freezes $\theta_0$ and optimizes $(A_i, B_i)$ for each particle independently. We learn variational distributions $q_A(A), q_B(B)$ over shared factors, with individual samples drawn implicitly. Bella's posterior is an empirical distribution over $n_{\text{particles}}$ affine subspaces around $\theta_0$, while ours concentrates on a single rank-$r$ manifold $\mathcal{R}_r$ in the full weight space.

Computationally, Bella requires $n_{\text{particles}}$ forward and backward passes; we require one per minibatch, leveraging reparameterization for efficient gradient estimation. Additionally, Bella depends on a high-quality pretrained model, whereas we work from random initialization, enabling application to novel architectures without pretrained checkpoints.

## A.5. Post-hoc Laplace Approximation on LoRA ([Yang et al., 2024](#))

This approach freezes a pretrained backbone, finetunes LoRA adapters to a maximum a posteriori (MAP) estimate, then applies the Laplace approximation (K-FAC-based Hessian curvature) around that point in the LoRA subspace. Computing and storing the Hessian approximation requires $\mathcal{O}(r(m + n))$ parameters with additional structural overhead.

The training schemes differ substantially: Laplace-LoRA is post-hoc (requiring pretrained backbone, finetuning, then Hessian computation), whereas ours is end-to-end from initialization. The posterior types also differ: Laplace assumes a local quadratic approximation around the MAP point, while our mean-field variational posterior does not rely on local curvature assumptions.

In terms of expressiveness, the Laplace approximation is accurate near the MAP but can be poor in regions of high curvature. Our factorization captures global structure via the low-rank constraint (Lemma 3.5). Regarding scalability, Hessian computation in high dimensions is computationally expensive even with K-FAC approximations; we avoid this entirely through reparameterization.

## A.6. Adjacent low-rank literatures

Our framework is also related to several adjacent low-rank literatures, but the connection is only partial.

**Reduced-rank regression.**   The closest link is the shared use of low-rank coefficient structure ([Izenman, 1975](#); [Reinsel & Velu, 1998](#)). However, reduced-rank regression studies low-rank coefficient matrices in multivariate linear models, whereas our setting is a Bayesian neural predictor with nonlinear feature learning and uncertainty-aware prediction and generalization.

**Probabilistic and Bayesian matrix factorization.**   These methods also use bilinear latent-factor parameterizations ([Mnih & Salakhutdinov, 2007](#); [Salakhutdinov & Mnih, 2008](#)). The key difference is that they model the entries of an observed data matrix directly, whereas our factorization parameterizes neural-network weight matrices inside a predictive model. Thus the induced posterior in our setting is over model parameters and predictors rather than over observed matrix entries.

**Deterministic low-rank compression.**   Low-rank compression methods exploit redundancy in neural networks to reduce memory or computation, either by approximating trained models or by encouraging low-rank structure during deterministic training ([Denton et al., 2014](#); [Alvarez & Salzmann, 2017](#)). By contrast, our method learns a stochastic low-rank parameterization end-to-end from initialization, so the low-rank structure shapes both training and inference rather than only producing a compressed deterministic model.

**Post-training and function-space uncertainty methods.**   Our approach is also distinct from post-training or function-space uncertainty methods such as linearized Laplace approximations ([Immer et al., 2021](#)), SNGP ([Liu et al., 2020](#)), and

fixed-mean GP approaches (Ortega et al., 2024). These methods start from a trained deterministic or pretrained predictor and then add uncertainty through local linearization, GP-style output layers, or a GP whose mean is fixed to the original network output. They are therefore complementary rather than direct substitutes: they do not learn a posterior over low-rank weight factors from initialization, and they typically preserve the underlying predictive backbone. By contrast, our method places a variational posterior directly on matrix-factorized network weights, so the uncertainty model shapes both optimization and prediction end-to-end.

### A.7. Summary

Our method is distinguished by combining **(i) end-to-end variational learning** with **(ii) parameter-efficient factorization** and **(iii) geometrically-constrained posterior support** on a rank-$r$ manifold. The rank-$r$ manifold constraint (Theorem 3.4) provides theoretical advantages in generalization bounds (Section 3.4) and empirical benefits in out-of-distribution detection and calibration, as demonstrated in our experiments. Table 5 summarizes the key difference across the approaches we discussed.

## B. Background and Proofs

### B.1. Variational Inference for Bayesian Neural Networks

Given a dataset $\mathcal{D} = \{(x_n, y_n)\}_{n=1}^N$ and a neural network with weights $W$, Bayesian inference seeks the posterior distribution $p(W|\mathcal{D}) \propto p(\mathcal{D}|W)p(W)$, which is intractable for modern architectures. Variational inference approximates the posterior with a tractable family $q(W; \theta)$ by maximizing the evidence lower bound (ELBO): $\mathcal{L}(q) = \mathbb{E}_{q(W)}[\log p(\mathcal{D}|W)] - \mathrm{KL}(q(W)\|p(W))$. The first term encourages data fit while the second regularizes by penalizing deviation from the prior. The standard approach, mean-field variational inference (MFVI), assumes fully factorized Gaussians $q(W) = \prod_{ij} \mathcal{N}(w_{ij}|\mu_{ij}, \sigma_{ij}^2)$. While this enables tractable optimization via the reparameterization trick (Kingma & Welling, 2014), it suffers from two key limitations: **(i)** the doubling of the number of parameters relative to a deterministic network, requiring storage of both $\mu_{ij}$ and $\sigma_{ij}$ for each weight, and **(ii)** the independence assumption prevents capturing correlations between weights. These limitations motivate our low-rank approach.

### B.2. Low-Rank Matrix Factorization

**Theorem B.1** (Existence of Low-Rank Factorization). *Let $W \in \mathbb{R}^{m \times n}$ be a matrix of rank $r \leq \min(m, n)$ Then there exist matrices $A \in \mathbb{R}^{m \times r}$ and $B \in \mathbb{R}^{n \times r}$ such that $W = AB^T$ (Ford, 2014). The EYM theorem further establishes that for any full-rank matrix with singular value decomposition $W_{full} = \sum_{i=1}^{\ell} \sigma_i u_i v_i^\top$, where $\sigma_i$ are the singular values (in decreasing order) $u_i \in \mathbb{R}^m$ are the left singular vectors, and $v_i \in \mathbb{R}^n$ are the right singular vectors, the truncated SVD $W_r = \sum_{i=1}^{r} \sigma_i u_i v_i^\top$ provides the optimal rank-$r$ approximation in Frobenius norm. (See B.3 for complete proof.) with approximation error $\|W_{full} - W_r\|_F^2 = \sum_{i=r+1}^{\ell} \sigma_i^2$.*

### B.3. Proof of Theorem B.1: Existence of Low-Rank Factorization

*Proof.* We prove that for any matrix $W \in \mathbb{R}^{m \times n}$ with $\mathrm{rank}(W) \leq r$, there exist matrices $A \in \mathbb{R}^{m \times r}$ and $B \in \mathbb{R}^{n \times r}$ such that $W = AB^\top$. The proof proceeds by considering two cases: (i) when $\mathrm{rank}(W) = r$ exactly, and (ii) when $\mathrm{rank}(W) = k < r$ is strictly less than $r$.

**Case 1:** $\mathrm{rank}(W) = r$**.**

Assume $\mathrm{rank}(W) = r$. Then the columns of $W$ span an $r$-dimensional subspace of $\mathbb{R}^m$. Let $A = [a_1, \ldots, a_r]$ be the matrix whose columns form a basis for this column space, so $A \in \mathbb{R}^{m \times r}$.

For every column $w_j$ of $W$ (for $j = 1, \ldots, n$), there exists a set of coefficients $(b_{j,i})_{i=1}^r$ such that $w_j$ is a linear combination of the basis vectors $a_1, \ldots, a_r$:

$$w_j = \sum_{i=1}^r b_{j,i} a_i, \quad \text{where} \quad a_i = \begin{bmatrix} a_{1i} \\ a_{2i} \\ \vdots \\ a_{mi} \end{bmatrix}.$$

Now we define $B \in \mathbb{R}^{n \times r}$ with rows $B_j = [b_{j,1}, \ldots, b_{j,r}]$:

$$
B = \begin{bmatrix} b_{1,1} & \cdots & b_{1,r} \\ \vdots & \ddots & \vdots \\ b_{n,1} & \cdots & b_{n,r} \end{bmatrix}.
$$

For each column $w_j$, we have $w_j = AB_j^\top$:

$$
w_j = \begin{bmatrix} a_{1,1} & \cdots & a_{1,r} \\ \vdots & \ddots & \vdots \\ a_{m,1} & \cdots & a_{m,r} \end{bmatrix} \begin{bmatrix} b_{j,1} \\ \vdots \\ b_{j,r} \end{bmatrix} = \begin{bmatrix} \sum_{i=1}^{r} a_{1,i} b_{j,i} \\ \vdots \\ \sum_{i=1}^{r} a_{m,i} b_{j,i} \end{bmatrix}.
$$

Thus $w_j = AB_j^\top$ for each column $j = 1, \ldots, n$, which yields $W = AB^\top$.

**Case 2:** $\mathrm{rank}(W) = k < r$**.**

Now assume $\mathrm{rank}(W) = k < r$. Then by Case 1, there exist matrices $A_k \in \mathbb{R}^{m \times k}$ and $B_k \in \mathbb{R}^{n \times k}$ such that $W = A_k B_k^\top$.

Define $A_r = [A_k \mid O_{m \times (r-k)}]$ by augmenting $A_k$ with $r - k$ zero columns, and similarly define $B_r = [B_k \mid O_{n \times (r-k)}]$ by augmenting $B_k$ with $r - k$ zero columns. Then:

$$
\begin{aligned}
A_r B_r^\top &= \begin{bmatrix} A_k \mid O_{m \times (r-k)} \end{bmatrix} \begin{bmatrix} B_k^\top \\ O_{(r-k) \times n} \end{bmatrix} \\
&= A_k B_k^\top + O_{m \times n} \\
&= A_k B_k^\top \\
&= W.
\end{aligned}
$$

Therefore $W = A_r B_r^\top$ where $A_r \in \mathbb{R}^{m \times r}$ and $B_r \in \mathbb{R}^{n \times r}$.

**Conclusion.**

We have shown that for any $W$ with $\mathrm{rank}(W) \leq r$, there exist $A \in \mathbb{R}^{m \times r}$ and $B \in \mathbb{R}^{n \times r}$ such that $W = AB^\top$, completing the proof. $\square$

### B.4. Proof of Lemma 3.2: Constrained Support

*Proof.* Let $q(A, B)$ be any variational distribution over $A \in \mathbb{R}^{m \times r}$ and $B \in \mathbb{R}^{n \times r}$. We show that the induced posterior $q_W$ places all its probability mass on the set of rank-at-most-$r$ matrices.

By Theorem B.1, any matrix of the form $W = AB^\top$ with $A \in \mathbb{R}^{m \times r}$ and $B \in \mathbb{R}^{n \times r}$ satisfies $\mathrm{rank}(W) \leq r$.

The distribution of the induced posterior, $q_W$, is the image measure (or pushforward measure) of $q(A, B)$ under the deterministic function:

$$
f(A, B) = AB^\top
$$

defined as:

$$
f : \mathbb{R}^{m \times r} \times \mathbb{R}^{n \times r} \longrightarrow \mathbb{R}^{m \times n}, \qquad (A, B) \longmapsto AB^\top.
$$

Thus every sample $(A, B) \sim q(A, B)$ is mapped to $W = AB^\top$, which by construction satisfies $\mathrm{rank}(W) \leq r$.

Therefore, $q_W$ places all its probability mass on the set

$$
\mathcal{R}_r = \{ W \in \mathbb{R}^{m \times n} : \mathrm{rank}(W) \leq r \},
$$

that is, $q_W(\mathcal{R}_r) = 1$. $\square$

## B.5. Proof of Lemma 3.3: Measure Zero of Low-Rank Manifold

*Proof.* We prove that the set of matrices with rank at most $r$ has Lebesgue measure zero in $\mathbb{R}^{m \times n}$.

Recall from the characterization of matrix rank (see, e.g., (Ford, 2014)) that for any $W \in \mathcal{R}_r$ (i.e., $\text{rank}(W) \leq r$), we have:

$$\det(W_{I,J}) = 0 \quad \forall\, I \subset \{1, \ldots, m\} \text{ and } J \subset \{1, \ldots, n\}$$

such that $|I| = |J| = r + 1$. That is, all $(r+1) \times (r+1)$ minors must vanish.

Let us take a given pair of indices $(I_0, J_0)$ of size $r + 1$. Define the function $g : \mathbb{R}^{m \times n} \to \mathbb{R}$ as

$$g(W) = \det(W_{I_0, J_0}).$$

Since the determinant is a sum of products of entries, $g(W)$ is a **polynomial** in the entries of $W$.

**Claim:** The polynomial $g(W)$ is *not identically zero*.

*Proof of claim:* There exists at least one matrix $W \in \mathbb{R}^{m \times n}$ such that $g(W) \neq 0$. To see this, construct $W^*$ such that the submatrix $W^*_{I_0, J_0} = I_{r+1}$ (the identity matrix of size $r + 1$) and all other entries are zero. Then:

$$g(W^*) = \det(I_{r+1}) = 1 \neq 0.$$

Thus $g$ is not identically zero.

We now apply the following classical result from real analysis (see (Folland, 1999)):

**Theorem B.2** (Zero Sets of Polynomials)**.** *If $p : \mathbb{R}^d \to \mathbb{R}$ is a polynomial that is not identically zero, then the set of its zeros $\{x \in \mathbb{R}^d : p(x) = 0\}$ has Lebesgue measure zero.*

Applying this theorem to our function $g$, the set

$$Z_{I_0, J_0} = \{W \in \mathbb{R}^{m \times n} \mid g(W) = 0\}$$

has Lebesgue measure zero:

$$\lambda(Z_{I_0, J_0}) = 0.$$

Now, since a matrix has rank at most $r$ if and only if *all* its $(r+1) \times (r+1)$ minors vanish, we can write:

$$\mathcal{R}_r = \bigcap_{\substack{I, J: \\ |I| = |J| = r+1}} Z_{I,J}.$$

There are finitely many such index pairs (at most $\binom{m}{r+1} \cdot \binom{n}{r+1}$). Since each $Z_{I,J}$ has measure zero and a finite (hence countable) intersection of measure-zero sets has measure zero, we conclude:

$$\lambda(\mathcal{R}_r) = 0.$$

This completes the proof. $\qquad\square$

## B.6. Additional Remarks on Theorem 3.4: Singularity

The singularity of $q_W$ with respect to Lebesgue measure has several important implications beyond what we discussed in the main text.

**Relationship to Measure-Theoretic Concepts:** The induced posterior $q_W$ is mutually singular with respect to Lebesgue measure, meaning it places all its mass on a set $(\mathcal{R}_r)$ that has zero Lebesgue measure. When $q(A, B)$ is chosen to be continuous (e.g., factorized Gaussians), $q_W$ is an example of a *singular continuous measure* i.e it has no atoms (point masses) yet cannot be represented by a density with respect to Lebesgue measure. This is distinct from absolutely continuous measures (which have densities) and discrete measures (which have atoms).

**Comparison to Mean-Field.** Mean-field Gaussian posteriors are absolutely continuous with respect to Lebesgue measure, meaning they admit a density function $f : \mathbb{R}^{m \times n} \to [0, \infty)$. This difference in measure-theoretic character explains why the two approaches have such different inductive biases: mean-field spreads mass across the entire space with positive density everywhere, while our induced posterior concentrates entirely on the low-rank manifold.

**Practical Implications.** Despite being singular with respect to Lebesgue measure, $q_W$ is perfectly well-defined and tractable. Samples can be drawn efficiently by sampling $(A, B) \sim q(A, B)$ and computing $W = AB^\top$. Expectations with respect to $q_W$ can be computed via Monte Carlo sampling. The singularity is a geometric property that constrains the support to $\mathcal{R}_r$, not a computational obstacle. In fact, this concentration of support is precisely what provides the beneficial inductive bias discussed in Section 3.1.

### B.7. Proof of Lemma 3.5: Covariance Structure

We derive the covariance structure between weight entries induced by the low-rank factorization $W = AB^\top$.

#### B.7.1. SETUP

Consider two weight entries $W_{ij}$ and $W_{i'j'}$ where

$$W_{ij} = \sum_{k=1}^{r} A_{ik} B_{jk}, \qquad W_{i'j'} = \sum_{\ell=1}^{r} A_{i'\ell} B_{j'\ell}.$$

We seek to compute $\mathrm{Cov}(W_{ij}, W_{i'j'})$ under a factorized variational posterior:

$$q(A, B) = \left( \prod_{i,k} q(A_{ik}) \right) \left( \prod_{j,k} q(B_{jk}) \right),$$

which implies $A \perp B$ and independence between distinct entries within each factor.

#### B.7.2. AUXILIARY LEMMA: COVARIANCE OF PRODUCTS

**Lemma B.3** (Covariance of Independent Products). *Let $A$ and $B$ be independent random variables, and let $X, U$ be measurable functions of $A$ and $Y, V$ be measurable functions of $B$. Then*

$$\mathrm{Cov}(XY, UV) = \mathbb{E}[XU]\mathbb{E}[YV] - \mathbb{E}[X]\mathbb{E}[U]\mathbb{E}[Y]\mathbb{E}[V].$$

*Proof.* By definition of covariance:

$$\mathrm{Cov}(XY, UV) = \mathbb{E}[XYUV] - \mathbb{E}[XY]\mathbb{E}[UV].$$

Since $X, U$ depend only on $A$ and $Y, V$ depend only on $B$, and $A \perp B$, the expectations factorize:

$$\mathbb{E}[XYUV] = \mathbb{E}[XU]\mathbb{E}[YV].$$

Similarly:

$$\mathbb{E}[XY] = \mathbb{E}[X]\mathbb{E}[Y], \qquad \mathbb{E}[UV] = \mathbb{E}[U]\mathbb{E}[V].$$

Substituting:

$$\mathrm{Cov}(XY, UV) = \mathbb{E}[XU]\mathbb{E}[YV] - \mathbb{E}[X]\mathbb{E}[Y]\mathbb{E}[U]\mathbb{E}[V].$$

$\square$

B.7.3. DERIVATION OF WEIGHT COVARIANCE

Define $Z_k = A_{ik}B_{jk}$ and $Z'_\ell = A_{i'\ell}B_{j'\ell}$. Then:

$$\mathrm{Cov}(W_{ij}, W_{i'j'}) = \mathrm{Cov}\left(\sum_{k=1}^r Z_k, \sum_{\ell=1}^r Z'_\ell\right).$$

By bilinearity of covariance:

$$\mathrm{Cov}(W_{ij}, W_{i'j'}) = \sum_{k=1}^r \sum_{\ell=1}^r \mathrm{Cov}(Z_k, Z'_\ell).$$

**Case 1:** $k \neq \ell$.

By independence of distinct entries and Lemma B.3:

$$\begin{aligned}
\mathrm{Cov}(A_{ik}B_{jk}, A_{i'\ell}B_{j'\ell}) &= \mathbb{E}[A_{ik}A_{i'\ell}]\mathbb{E}[B_{jk}B_{j'\ell}] - \mathbb{E}[A_{ik}]\mathbb{E}[A_{i'\ell}]\mathbb{E}[B_{jk}]\mathbb{E}[B_{j'\ell}] \\
&= \mathbb{E}[A_{ik}]\mathbb{E}[A_{i'\ell}]\mathbb{E}[B_{jk}]\mathbb{E}[B_{j'\ell}] - \mathbb{E}[A_{ik}]\mathbb{E}[A_{i'\ell}]\mathbb{E}[B_{jk}]\mathbb{E}[B_{j'\ell}] \\
&= 0.
\end{aligned}$$

**Case 2:** $k = \ell$.

For the diagonal terms:

$$\mathrm{Cov}(A_{ik}B_{jk}, A_{i'k}B_{j'k}) = \mathbb{E}[A_{ik}A_{i'k}]\mathbb{E}[B_{jk}B_{j'k}] - \mathbb{E}[A_{ik}]\mathbb{E}[A_{i'k}]\mathbb{E}[B_{jk}]\mathbb{E}[B_{j'k}].$$

Therefore:

$$\mathrm{Cov}(W_{ij}, W_{i'j'}) = \sum_{k=1}^r \left(\mathbb{E}[A_{ik}A_{i'k}]\mathbb{E}[B_{jk}B_{j'k}] - \mathbb{E}[A_{ik}]\mathbb{E}[A_{i'k}]\mathbb{E}[B_{jk}]\mathbb{E}[B_{j'k}]\right).$$

B.7.4. INTERPRETATION

This result establishes that weight entries $W_{ij}$ and $W_{i'j'}$ are **correlated** whenever the corresponding factor entries share statistical dependencies. Specifically:

- **Row correlation**: If $i = i'$ (same row), weights in that row are correlated through shared $A$ factors.

- **Column correlation**: If $j = j'$ (same column), weights in that column are correlated through shared $B$ factors.

- **Rank-modulated coupling**: The correlation strength depends on all $r$ latent dimensions, allowing the model to capture complex interdependencies.

**Contrast with Mean-Field.** A full-rank mean-field posterior imposes $\mathrm{Cov}(W_{ij}, W_{i'j'}) = 0$ for $(i, j) \neq (i', j')$, forcing all weights to be independent. The low-rank factorization naturally induces structured correlations, enabling the posterior to capture coherent variations across many weights simultaneously. This structured dependency is crucial when the true posterior exhibits correlated directions, as it provides a richer variational family than fully factorized approximations (see Figure 1).

# C. Proofs from Theoretical Analysis

## C.1. Loss Approximation Preliminaries

C.1.1. PRELIMINARIES

**Definition C.1** (Expected Loss)**.** The expected loss (risk) over the data distribution $\mathcal{D}$ is given by

$$\mathcal{L}(W) = \mathbb{E}_{(x,y)\sim\mathcal{D}}\big[\ell(Wx, y)\big].$$

**Definition C.2** ($\beta$-Smoothness). A function $\ell(z, y)$ is $\beta$-smooth (with respect to $z$) if, for all $z, z' \in \mathbb{R}^d$,

$$\left\| \nabla_z \ell(z, y) - \nabla_z \ell(z', y) \right\|_2 \leq \beta \|z - z'\|_2.$$

This implies the standard quadratic upper bound (descent lemma): for all $z, z' \in \mathbb{R}^d$,

$$\ell(z', y) \leq \ell(z, y) + \langle \nabla_z \ell(z, y), \, z' - z \rangle + \frac{\beta}{2} \|z' - z\|_2^2.$$

### C.1.2. ECKART-YOUNG-MIRSKY THEOREM

**Theorem C.3** (Eckart-Young-Mirsky Theorem (Eckart & Young, 1936)). *Let $W \in \mathbb{R}^{m \times n}$ have singular value decomposition*

$$W = \sum_{i=1}^{\rho} \sigma_i u_i v_i^\top,$$

*where $\rho = \mathrm{rank}(W)$, $\sigma_1 \geq \sigma_2 \geq \cdots \geq \sigma_\rho > 0$ are the singular values, and $\{u_i\}$, $\{v_i\}$ are the corresponding left and right singular vectors. For any $r < \rho$, define the rank-$r$ truncation*

$$W_r = \sum_{i=1}^{r} \sigma_i u_i v_i^\top.$$

*Then $W_r$ is the best rank-$r$ approximation to $W$ in both the Frobenius norm and the spectral (operator) norm:*

$$W_r = \arg \min_{\substack{M \in \mathbb{R}^{m \times n} \\ \mathrm{rank}(M) \leq r}} \|W - M\|_F, \tag{7}$$

$$W_r = \arg \min_{\substack{M \in \mathbb{R}^{m \times n} \\ \mathrm{rank}(M) \leq r}} \|W - M\|_2. \tag{8}$$

*Moreover, the approximation errors are given exactly by the tail singular values:*

$$\|W - W_r\|_F = \sqrt{\sum_{i=r+1}^{\rho} \sigma_i^2}, \tag{9}$$

$$\|W - W_r\|_2 = \sigma_{r+1}. \tag{10}$$

*Proof sketch.* The proof relies on the orthogonal invariance of both norms and the fact that singular vectors form orthonormal bases.

For the Frobenius norm: Since $\{u_i v_i^\top\}$ are orthonormal under the Frobenius inner product $\langle A, B \rangle_F = \mathrm{tr}(A^\top B)$, we have

$$\|W - W_r\|_F^2 = \left\| \sum_{i=r+1}^{\rho} \sigma_i u_i v_i^\top \right\|_F^2 = \sum_{i=r+1}^{\rho} \sigma_i^2,$$

by orthonormality. To show optimality, suppose $M$ has rank at most $r$. The dimension of the space orthogonal to $M$'s range has dimension at least $m - r$, which must intersect the $(n - r)$-dimensional space spanned by $\{u_{r+1}, \ldots, u_\rho\}$. This intersection implies $\|W - M\|_F \geq \sigma_{r+1}$, with equality achieved by $W_r$.

For the spectral norm: Since $\|W - W_r\|_2 = \max_{\|x\|=1} \|(W - W_r)x\|$, and $W - W_r = \sum_{i=r+1}^{\rho} \sigma_i u_i v_i^\top$, the maximum is achieved when $x = v_{r+1}$, giving $\|W - W_r\|_2 = \sigma_{r+1}$.

For complete proofs, see (Eckart & Young, 1936; Stewart & Sun, 1990). $\qquad\square$

### C.1.3. FROBENIUS-EUCLIDEAN CONSISTENCY

**Lemma C.4** (Consistency of Frobenius and Euclidean Norms). *For any matrix $A \in \mathbb{R}^{m \times n}$ and vector $x \in \mathbb{R}^n$,*

$$\|Ax\|_2 \leq \|A\|_F \|x\|_2.$$

*Proof.* Let $A = [a_1 \mid a_2 \mid \cdots \mid a_n]$ where $a_j \in \mathbb{R}^m$ are the columns of $A$. Then

$$Ax = \sum_{j=1}^n x_j a_j.$$

By the triangle inequality and Cauchy-Schwarz inequality:

$$
\begin{aligned}
\|Ax\|_2 &= \left\| \sum_{j=1}^n x_j a_j \right\|_2 \\
&\leq \sum_{j=1}^n |x_j| \|a_j\|_2 \quad \text{(triangle inequality)} \\
&\leq \left( \sum_{j=1}^n |x_j|^2 \right)^{1/2} \left( \sum_{j=1}^n \|a_j\|_2^2 \right)^{1/2} \quad \text{(Cauchy-Schwarz)} \\
&= \|x\|_2 \sqrt{\sum_{j=1}^n \sum_{i=1}^m a_{ij}^2} \\
&= \|x\|_2 \|A\|_F.
\end{aligned}
$$

Here we used the definition of the Frobenius norm:

$$\|A\|_F = \sqrt{\sum_{i=1}^m \sum_{j=1}^n a_{ij}^2} = \sqrt{\sum_{j=1}^n \|a_j\|_2^2}.$$

This establishes the desired inequality. See Chapters 4 and 5 of (Ford, 2014) for further discussion of matrix norm relationships. □

### C.2. Proof of Theorem 3.6: Optimal Rank-$r$ Approximation

**Theorem C.5** (Restating Theorem 3.6). *Let $W^*$ be the optimal full-rank weight matrix and $W_r^*$ be the optimal rank-$r$ matrix (via truncated SVD). Under $L$-Lipschitz loss and bounded inputs $\|x\|_2 \leq R$ a.s.,*

$$|\mathbb{E}[\ell(W^* x, y)] - \mathbb{E}[\ell(W_r^* x, y)]| \leq LR \sqrt{\sum_{i=r+1}^\rho \sigma_i^2(W^*)}.$$

*Proof.* First, since $W^*$ is the global minimizer of the loss $\mathcal{L}$, we know that for any other matrix, including $W_r^*$, the excess risk is non-negative:

$$\mathcal{L}(W_r^*) - \mathcal{L}(W^*) \geq 0.$$

The same holds for the pointwise loss:

$$\ell(W_r^* x, y) - \ell(W^* x, y) \geq 0.$$

By definition:

$$W^* = \operatorname*{argmin}_{W \in \mathbb{R}^{m \times n}} \mathcal{L}(W), \quad W_r^* = \sum_{i=1}^r \sigma_i u_i^* v_i^{*\top}.$$

From the Lipschitz property of $\ell$:

$$\ell(W_r^* x, y) - \ell(W^* x, y) \le L\|(W_r^* - W^*)x\|_2.$$

Using the consistency of the Frobenius norm with the Euclidean norm (Lemma C.4):

$$\|(W_r^* - W^*)x\|_2 \le \|x\|_2\|W_r^* - W^*\|_F.$$

Thus:

$$\ell(W_r^* x, y) - \ell(W^* x, y) \le L\|x\|_2 \underbrace{\|W_r^* - W^*\|_F}_{\sqrt{\sum_{i=r+1}^\rho \sigma_i^2}}.$$

By the Eckart-Young-Mirsky theorem (Theorem C.3), since $W_r^*$ is the SVD truncation of $W^*$, the Frobenius norm of the difference is exactly determined by the tail singular values:

$$\|W_r^* - W^*\|_F = \sqrt{\sum_{i=r+1}^\rho \sigma_i^2(W^*)}.$$

Taking the expectation over the data distribution $\mathcal{D}$ to move from pointwise loss $\ell$ to risk $\mathcal{L}$:

$$\mathbb{E}_{(x,y)\sim\mathcal{D}}\left[\ell(W_r^* x, y) - \ell(W^* x, y)\right] \le \mathbb{E}_{(x,y)\sim\mathcal{D}}\left[L\|x\|_2 \sqrt{\sum_{i=r+1}^\rho \sigma_i^2(W^*)}\right].$$

Since $\|x\|_2 \le R$ almost surely by assumption, and the singular values are constants with respect to the expectation:

$$\mathbb{E}_{(x,y)\sim\mathcal{D}}\left[\ell(W_r^* x, y) - \ell(W^* x, y)\right] \le LR\sqrt{\sum_{i=r+1}^\rho \sigma_i^2(W^*)}.$$

By definition of the risk $\mathcal{L}(W) = \mathbb{E}_{(x,y)\sim\mathcal{D}}[\ell(Wx, y)]$:

$$\mathcal{L}(W_r^*) - \mathcal{L}(W^*) \le LR\sqrt{\sum_{i=r+1}^\rho \sigma_i^2(W^*)}.$$

Since $\mathcal{L}(W_r^*) - \mathcal{L}(W^*) \ge 0$ (as $W^*$ is the minimizer), we have established the bound. This completes the proof. $\square$

### C.3. Proof of Theorem 3.7: Learned Approximation

**Theorem C.6** (Restating Theorem 3.7). *Let $W^*$ be the optimal full-rank weight matrix and $W_r^*$ be the optimal rank-$r$ matrix (via truncated SVD) and let $W = AB^\top$ be the optimal learned rank-$r$ matrix. Then for any $(x, y)$,*

$$\left|\ell(Wx, y) - \ell(W^* x, y)\right| \le LR\left(\|W - W_r^*\|_F + \sqrt{\sum_{i=r+1}^\rho \sigma_i^2(W^*)}\right).$$

*Proof.* Add and subtract $\ell(W_r^* x, y)$ and apply the triangle inequality:

$$\left|\ell(Wx, y) - \ell(W^* x, y)\right| = \left|\ell(Wx, y) - \ell(W_r^* x, y) + \ell(W_r^* x, y) - \ell(W^* x, y)\right|$$
$$\le \left|\ell(Wx, y) - \ell(W_r^* x, y)\right| + \left|\ell(W_r^* x, y) - \ell(W^* x, y)\right|.$$

**First term (learning error):**

By $L$-Lipschitz continuity:

$$\left|\ell(Wx, y) - \ell(W_r^* x, y)\right| \leq L\|(W - W_r^*)x\|_2 \leq L\|x\|_2\|W - W_r^*\|_F \leq LR\|W - W_r^*\|_F,$$

where we used Lemma C.4 and $\|x\|_2 \leq R$ a.s.

**Second term (rank bias):**

From Theorem 3.6 (or equivalently, by Lipschitz continuity and the Eckart-Young-Mirsky theorem):

$$\left|\ell(W_r^* x, y) - \ell(W^* x, y)\right| \leq L\|x\|_2\|W_r^* - W^*\|_F = L\|x\|_2\sqrt{\sum_{i=r+1}^{\rho} \sigma_i^2(W^*)} \leq LR\sqrt{\sum_{i=r+1}^{\rho} \sigma_i^2(W^*)}.$$

Combining both terms:

$$\left|\ell(Wx, y) - \ell(W^* x, y)\right| \leq LR\left(\|W - W_r^*\|_F + \sqrt{\sum_{i=r+1}^{\rho} \sigma_i^2(W^*)}\right).$$

This completes the proof. $\qquad\square$

**Interpretation.** This decomposition reveals two sources of error:

1. **Learning error**: $\|W - W_r^*\|_F$ measures how far the learned $W = AB^\top$ is from the optimal rank-$r$ approximation. This depends on optimization quality and can be reduced with better training procedures.

2. **Rank bias (approximation error)**: $\sqrt{\sum_{i=r+1}^{\rho} \sigma_i^2(W^*)}$ is the unavoidable error from restricting to rank $r$. If $W^*$ has rapidly decaying singular values, this term is small.

### C.4. Extension to Stochastic Weights (Bayesian Setting)

**Lemma C.7** (Variance and Loss, Lemma 1.2.3 in (Nesterov, 2018)). *Assume $\ell(\cdot, y)$ is differentiable and $\beta$-smooth in its first argument. Let $Z$ be an $\mathbb{R}^d$-valued random variable with $\mathbb{E}\|Z\|_2^2 < \infty$ and set $\mu := \mathbb{E}[Z]$. Then*

$$\left|\mathbb{E}\big[\ell(Z, y)\big] - \ell(\mathbb{E}[Z], y)\right| \leq \frac{\beta}{2}\,\mathbb{E}\big[\|Z - \mathbb{E}[Z]\|_2^2\big].$$

*Proof.* Since $\ell$ is $\beta$-smooth, by the descent lemma (Definition C.2):

$$\left|\ell(Z, y) - \ell(\mathbb{E}[Z], y) - \big\langle\nabla_z\ell(\mathbb{E}[Z], y),\, Z - \mathbb{E}[Z]\big\rangle\right| \leq \frac{\beta}{2}\|Z - \mathbb{E}[Z]\|_2^2.$$

This implies:

$$-\frac{\beta}{2}\|Z - \mathbb{E}[Z]\|_2^2 \leq \ell(Z, y) - \ell(\mathbb{E}[Z], y) - \big\langle\nabla_z\ell(\mathbb{E}[Z], y),\, Z - \mathbb{E}[Z]\big\rangle \leq \frac{\beta}{2}\|Z - \mathbb{E}[Z]\|_2^2.$$

Taking expectations (using linearity) and noting that $\mathbb{E}[Z - \mathbb{E}[Z]] = 0$:

$$-\frac{\beta}{2}\mathbb{E}[\|Z - \mathbb{E}[Z]\|_2^2] \leq \mathbb{E}\big[\ell(Z, y)\big] - \ell(\mathbb{E}[Z], y) \leq \frac{\beta}{2}\mathbb{E}[\|Z - \mathbb{E}[Z]\|_2^2].$$

Therefore:

$$\left|\mathbb{E}\big[\ell(Z, y)\big] - \ell(\mathbb{E}[Z], y)\right| \leq \frac{\beta}{2}\,\mathbb{E}[\|Z - \mathbb{E}[Z]\|_2^2].$$

$\qquad\square$

**Theorem C.8** (Expected Loss Difference with Random Weights). *Let $W_{\text{full}} \sim q_{\text{full}}$ and $AB^\top \sim q_{\text{lr}}$ be random weight matrices with means $\mu_f = \mathbb{E}[W_{\text{full}}]$ and $\mu_r = \mathbb{E}[AB^\top]$. Assume $\ell$ is L-Lipschitz and $\beta$-smooth. Define*

$$\Delta := \left| \mathbb{E}\big[\ell(W_{\text{full}}x, y)\big] - \mathbb{E}\big[\ell(AB^\top x, y)\big] \right|.$$

*Then*

$$\Delta \leq L\|x\|_2 \|\mu_f - \mu_r\|_F + \frac{\beta}{2}\left(\text{Var}(W_{\text{full}}x) + \text{Var}(AB^\top x)\right).$$

*Proof.* Let $z_f = W_{\text{full}}x$ and $z_r = AB^\top x$. We decompose:

$$
\begin{aligned}
\Delta &= \left| \mathbb{E}\big[\ell(z_f, y)\big] - \mathbb{E}\big[\ell(z_r, y)\big] \right| \\
&= \left| \mathbb{E}\big[\ell(z_f, y)\big] - \ell(\mathbb{E}[z_f], y) + \ell(\mathbb{E}[z_f], y) - \ell(\mathbb{E}[z_r], y) + \ell(\mathbb{E}[z_r], y) - \mathbb{E}\big[\ell(z_r, y)\big] \right| \\
&\leq \left| \mathbb{E}\big[\ell(z_f, y)\big] - \ell(\mathbb{E}[z_f], y) \right| + \left| \ell(\mathbb{E}[z_f], y) - \ell(\mathbb{E}[z_r], y) \right| + \left| \ell(\mathbb{E}[z_r], y) - \mathbb{E}\big[\ell(z_r, y)\big] \right|.
\end{aligned}
$$

**Term 1 and Term 3:** By Lemma C.7:

$$
\begin{aligned}
\left| \mathbb{E}\big[\ell(z_f, y)\big] - \ell(\mathbb{E}[z_f], y) \right| &\leq \frac{\beta}{2}\,\mathbb{E}[\|z_f - \mathbb{E}[z_f]\|_2^2] = \frac{\beta}{2}\,\text{Var}(W_{\text{full}}x), \\
\left| \ell(\mathbb{E}[z_r], y) - \mathbb{E}\big[\ell(z_r, y)\big] \right| &\leq \frac{\beta}{2}\,\text{Var}(AB^\top x).
\end{aligned}
$$

**Term 2:** By $L$-Lipschitz continuity and submultiplicativity:

$$\left| \ell(\mathbb{E}[z_f], y) - \ell(\mathbb{E}[z_r], y) \right| \leq L\,\|\mathbb{E}[z_f] - \mathbb{E}[z_r]\|_2 \leq L\,\|x\|_2\,\|\mu_f - \mu_r\|_F.$$

Combining all three terms:

$$\Delta \leq L\|x\|_2\,\|\mu_f - \mu_r\|_F + \frac{\beta}{2}\Big(\text{Var}(W_{\text{full}}x) + \text{Var}(AB^\top x)\Big).$$

$\square$

**Remark.** This variance-based bound shows that posterior uncertainty contributes an additional $O(\beta \cdot \text{Var})$ term beyond the deterministic approximation error. However, for practical rank selection, Theorems 3.6 and 3.7 provide more directly actionable guidance.

## C.5. Analysis of Common Loss Functions

We verify that standard loss functions used in practice satisfy the assumptions of our loss approximation theorems (Theorems 3.6 and 3.7). The key requirement is that $\ell(\cdot, y)$ is $L$-Lipschitz in its first argument.

### C.5.1. CLASSIFICATION LOSSES

The following standard classification losses are globally Lipschitz:

**Cross-Entropy with Softmax.** For multi-class classification with $C$ classes, let $z \in \mathbb{R}^C$ be the logits and $y \in \{1, \ldots, C\}$ be the true class. The cross-entropy loss is

$$\ell(z, y) = -z_y + \log\left(\sum_{k=1}^{C} e^{z_k}\right).$$

**Proposition C.9.** *The cross-entropy loss with softmax is $\sqrt{2}$-Lipschitz.*

*Proof.* The gradient is $\nabla_z \ell(z, y) = \sigma(z) - e_y$, where $\sigma(z)$ is the softmax and $e_y$ is the one-hot encoding. Since $\|\sigma(z) - e_y\|_2 \leq \sqrt{2}$ (both are probability vectors, so the maximum distance is $\sqrt{2}$), we have $\|\nabla_z \ell\|_2 \leq \sqrt{2}$, establishing the Lipschitz constant. $\square$

**Binary Cross-Entropy.** For binary classification with sigmoid activation $\sigma(z) = 1/(1 + e^{-z})$:

$$\ell(z, y) = -y \log(\sigma(z)) - (1 - y) \log(1 - \sigma(z)).$$

The gradient is $\nabla_z \ell = \sigma(z) - y \in [-1, 1]$, so the loss is 1-Lipschitz.

**Hinge Loss.** For binary classification with $y \in \{-1, +1\}$:

$$\ell(z, y) = \max(0, 1 - yz).$$

The gradient is $\nabla_z \ell \in \{0, -y\}$, giving $L = 1$.

### C.5.2. REGRESSION LOSSES

**Mean Absolute Error (MAE).** For scalar-output regression,

$$\ell(z, y) = |z - y|.$$

This loss is 1-Lipschitz with respect to the Euclidean norm:

$$|\ell(z, y) - \ell(z', y)| \le |z - z'| = \|z - z'\|_2.$$

**Mean Squared Error (MSE).**

$$\ell(z, y) = \frac{1}{2} \|z - y\|_2^2.$$

**Important:** MSE is *not* globally Lipschitz. The gradient is $\nabla_z \ell = z - y$, which is unbounded as $\|z - y\| \to \infty$. However, under the assumption that outputs are bounded (i.e., $\|z\|_2, \|y\|_2 \le B$ for some constant $B$), MSE becomes $(2B)$-Lipschitz on the restricted domain.

*Remark* C.10 (Bounded Outputs for MSE). In practice, neural network outputs are often implicitly bounded:

- Activation functions like tanh, sigmoid impose explicit bounds

- Numerical stability requires finite outputs

- Training with gradient clipping effectively bounds the domain

Under such conditions, MSE satisfies our Lipschitz assumption with $L = 2B$ where $B$ is the output bound. For unbounded regression, MAE or Huber loss should be used instead.

### C.5.3. SMOOTHNESS PROPERTIES

For the Bayesian extension (Theorem C.8), we also require $\beta$-smoothness.

**MSE is 1-smooth.** For

$$\ell(z, y) = \frac{1}{2} \|z - y\|_2^2,$$

the Hessian is

$$\nabla_z^2 \ell(z, y) = I,$$

so

$$\|\nabla_z^2 \ell(z, y)\|_2 = 1.$$

Therefore, under Definition C.2, MSE is 1-smooth.

**Cross-Entropy is smooth.** The Hessian is the covariance matrix of the softmax distribution, which has eigenvalues in $[0, 1/4]$, giving $\beta = O(1)$.

### C.5.4. INTERPRETATION OF LIPSCHITZ AND SMOOTHNESS

*Remark* C.11 (Lipschitz Continuity). A loss $\ell(\hat{y}, y)$ being $L$-Lipschitz (with respect to $\hat{y}$) means that small changes in predictions lead to small changes in loss:

$$|\ell(\hat{y}, y) - \ell(\hat{y}', y)| \leq L\|\hat{y} - \hat{y}'\|.$$

When differentiable, this is equivalent to bounded gradients:

$$\|\nabla_{\hat{y}}\ell(\hat{y}, y)\| \leq L \quad \text{for all } \hat{y}.$$

**Intuition:** Lipschitz continuity prevents the loss from having arbitrarily steep slopes. Non-Lipschitz losses (like unbounded MSE) can produce very large gradients when predictions are far from targets, potentially causing optimization instability.

*Remark* C.12 ($\beta$-Smoothness). A loss being $\beta$-smooth means the gradient itself cannot change too rapidly:

$$\|\nabla_{\hat{y}}\ell(\hat{y}, y) - \nabla_{\hat{y}}\ell(\hat{y}', y)\| \leq \beta\|\hat{y} - \hat{y}'\|.$$

This is equivalent to bounded curvature: $\|\nabla_{\hat{y}}^2 \ell\|_2 \leq \beta$.

**Consequence:** Smoothness ensures the loss can be upper-bounded by a quadratic approximation:

$$\ell(\hat{y}', y) \leq \ell(\hat{y}, y) + \langle \nabla_{\hat{y}}\ell(\hat{y}, y), \hat{y}' - \hat{y}\rangle + \frac{\beta}{2}\|\hat{y}' - \hat{y}\|^2.$$

This is the *descent lemma*, which is essential for optimization: taking gradient steps with learning rate $\eta \leq 1/\beta$ guarantees sufficient descent without overshooting.

*Remark* C.13 (Lipschitz vs. Smoothness). These are complementary properties:

- **Lipschitz**: Gradient magnitude is bounded $\|\nabla\ell\| \leq L$

- **Smooth**: Gradient variation is bounded $\|\nabla\ell - \nabla\ell'\| \leq \beta\|x - x'\|$

A function can be Lipschitz but not smooth (e.g., $|x|$ has bounded gradient but discontinuous derivative), or smooth but not Lipschitz (e.g., $x^2$ has unbounded gradient but bounded curvature on compact sets).

### C.5.5. SUMMARY TABLE

*Table 6.* Lipschitz and smoothness constants for common loss functions.

| Loss Function | $L$ (Lipschitz) | $\beta$ (Smooth) | Notes |
|---|---|---|---|
| MAE | 1 | N/A | Not differentiable |
| MSE | $2B$ | 1 | Requires bounded outputs $\|z\|, \|y\| \leq B$ |
| Cross-Entropy | $\sqrt{2}$ | $O(1)$ | Global for softmax |
| Binary Cross-Entropy | 1 | $O(1)$ | Global for sigmoid |
| Hinge Loss | 1 | N/A | Not differentiable everywhere |

## D. PAC-Bayes Generalization Bounds

### D.1. Background: PAC-Bayes Theory

The PAC-Bayes framework provides generalization bounds that depend on the complexity of the posterior distribution, measured by KL divergence from the prior.

**Definition D.1** (Risk and Empirical Risk). Let $\mathcal{D}$ be a distribution over $\mathcal{X} \times \mathcal{Y}$, and let $\ell(\theta, x, y) \in [0, 1]$ be a bounded loss function. For a posterior $Q$ over parameters $\Theta$:

**True risk (generalization error):**

$$L(Q) = \mathbb{E}_{\theta \sim Q}\left[\mathbb{E}_{(x,y) \sim \mathcal{D}}[\ell(\theta, x, y)]\right].$$

**Empirical risk (training error):**

$$\hat{L}(Q) = \mathbb{E}_{\theta \sim Q} \left[ \frac{1}{N} \sum_{i=1}^{N} \ell(\theta, x_i, y_i) \right].$$

The PAC-Bayes bound relates true risk to empirical risk plus a complexity penalty.

**Theorem D.2** (McAllester's Inequality (McAllester, 1998))**.** *Let $P$ be a fixed prior distribution over $\Theta$. For any $\delta \in (0, 1)$, with probability at least $1 - \delta$ over the random draw of training set $S \sim \mathcal{D}^N$,*

$$L(Q) \leq \hat{L}(Q) + \sqrt{\frac{\mathrm{KL}(Q\|P) + \ln\left(\frac{2\sqrt{N}}{\delta}\right)}{2N}}.$$

This bound decomposes generalization error into:

- **Data fit**: $\hat{L}(Q)$ measures training performance

- **Complexity penalty**: $\mathrm{KL}(Q\|P)$ measures deviation from prior

### D.2. KL Divergence for Factorized Distributions

**Lemma D.3** (KL for Factorized Distributions)**.** *Assume fully factorized posterior and prior:*

$$Q(\theta) = \prod_{i=1}^{D} q_i(\theta_i), \qquad P(\theta) = \prod_{i=1}^{D} p_i(\theta_i).$$

*Define $C_{\max} = \max_i \mathrm{KL}(q_i\|p_i)$. Then*

$$\mathrm{KL}(Q\|P) \leq C_{\max} \cdot D.$$

*Proof.* By factorization:

$$\begin{aligned}
\mathrm{KL}(Q\|P) &= \mathbb{E}_{\theta \sim Q} \left[ \ln \frac{Q(\theta)}{P(\theta)} \right] \\
&= \mathbb{E}_{\theta \sim Q} \left[ \ln \frac{\prod_{i=1}^{D} q_i(\theta_i)}{\prod_{i=1}^{D} p_i(\theta_i)} \right] \\
&= \mathbb{E}_{\theta \sim Q} \left[ \sum_{i=1}^{D} \ln \frac{q_i(\theta_i)}{p_i(\theta_i)} \right] \\
&= \sum_{i=1}^{D} \mathbb{E}_{\theta \sim Q} \left[ \ln \frac{q_i(\theta_i)}{p_i(\theta_i)} \right].
\end{aligned}$$

For the $i$-th term, by independence:

$$\begin{aligned}
\mathbb{E}_{\theta \sim Q} \left[ \ln \frac{q_i(\theta_i)}{p_i(\theta_i)} \right] &= \int \cdots \int \ln \frac{q_i(\theta_i)}{p_i(\theta_i)} \prod_{j=1}^{D} q_j(\theta_j) \, d\theta_1 \cdots d\theta_D \\
&= \int \ln \frac{q_i(\theta_i)}{p_i(\theta_i)} q_i(\theta_i) \, d\theta_i \cdot \prod_{j \neq i} \underbrace{\int q_j(\theta_j) \, d\theta_j}_{=1} \\
&= \mathrm{KL}(q_i\|p_i).
\end{aligned}$$

Therefore:

$$\mathrm{KL}(Q\|P) = \sum_{i=1}^{D} \mathrm{KL}(q_i\|p_i) \leq \sum_{i=1}^{D} C_{\max} = C_{\max} \cdot D.$$

$\square$

## D.3. Proof of Theorem 3.8

**Theorem D.4** (Restating Theorem 3.8). *Assume bounded loss $\ell(\theta, x, y) \in [0, 1]$ and factorized priors/posteriors.*

***Full-rank:*** *$W \in \mathbb{R}^{d_{in} \times d_{out}}$, $D_{full} = d_{in}d_{out}$, with $C_{\max}^{full} = \max_{i,j} \mathrm{KL}(q_{W_{ij}} \| p_{W_{ij}})$.*

***Low-rank:*** *$W = AB^\top$ with $A \in \mathbb{R}^{d_{in} \times r}$, $B \in \mathbb{R}^{d_{out} \times r}$, $D_{LR} = r(d_{in} + d_{out})$, with $C_{\max}^{LR} = \max\{C_{\max}^A, C_{\max}^B\}$ where $C_{\max}^A = \max_{i,k} \mathrm{KL}(q_{A_{ik}} \| p_{A_{ik}})$ and $C_{\max}^B = \max_{j,k} \mathrm{KL}(q_{B_{jk}} \| p_{B_{jk}})$.*

*Then with probability at least $1 - \delta$ over $S \sim \mathcal{D}^N$:*

$$L(Q_{full}) \leq \hat{L}(Q_{full}) + \sqrt{\frac{C_{\max}^{full} \cdot d_{in}d_{out} + \ln\left(\frac{2\sqrt{N}}{\delta}\right)}{2N}},$$

$$L(Q_{LR}) \leq \hat{L}(Q_{LR}) + \sqrt{\frac{C_{\max}^{LR} \cdot r(d_{in} + d_{out}) + \ln\left(\frac{2\sqrt{N}}{\delta}\right)}{2N}}.$$

*When $C_{\max}^{full} = C_{\max}^{LR} = C_{\max}$ (e.g., when using identical prior and posterior families for all parameters), the complexity ratio satisfies:*

$$\frac{Complexity(Q_{LR})}{Complexity(Q_{full})} = \sqrt{r\left(\frac{1}{d_{in}} + \frac{1}{d_{out}}\right)} < 1 \quad \text{when } r \ll \min(d_{in}, d_{out}).$$

*Proof.* **Step 1: Apply McAllester's bound.**

For the full-rank posterior $Q_{\text{full}}$, Theorem D.2 gives:

$$L(Q_{\text{full}}) \leq \hat{L}(Q_{\text{full}}) + \sqrt{\frac{\mathrm{KL}(Q_{\text{full}} \| P_{\text{full}}) + \ln\left(\frac{2\sqrt{N}}{\delta}\right)}{2N}}.$$

Similarly for low-rank:

$$L(Q_{\text{LR}}) \leq \hat{L}(Q_{\text{LR}}) + \sqrt{\frac{\mathrm{KL}(Q_{\text{LR}} \| P_{\text{LR}}) + \ln\left(\frac{2\sqrt{N}}{\delta}\right)}{2N}}.$$

**Step 2: Bound KL divergences using Lemma D.3.**

For full-rank with $D_{\text{full}} = d_{\text{in}}d_{\text{out}}$ parameters:

$$\mathrm{KL}(Q_{\text{full}} \| P_{\text{full}}) \leq C_{\max}^{\text{full}} \cdot d_{\text{in}}d_{\text{out}}.$$

For low-rank, we have:

$$\begin{aligned}
\mathrm{KL}(Q_{\text{LR}} \| P_{\text{LR}}) &= \mathrm{KL}(q_A(A) \| p_A(A)) + \mathrm{KL}(q_B(B) \| p_B(B)) \\
&\leq C_{\max}^A \cdot d_{\text{in}} \cdot r + C_{\max}^B \cdot d_{\text{out}} \cdot r \\
&\leq C_{\max}^{\text{LR}} \cdot (d_{\text{in}} \cdot r + d_{\text{out}} \cdot r) \\
&= C_{\max}^{\text{LR}} \cdot r(d_{\text{in}} + d_{\text{out}}).
\end{aligned}$$

**Step 3: Compute complexity ratio (under equal $C_{\max}$ assumption).**

When $C_{\max}^{\text{full}} = C_{\max}^{\text{LR}} = C_{\max}$, ignoring the common $\ln\left(\frac{2\sqrt{N}}{\delta}\right)$ term:

$$\frac{\text{Complexity}(Q_{\text{LR}})}{\text{Complexity}(Q_{\text{full}})} = \frac{\sqrt{C_{\max} \cdot r(d_{\text{in}} + d_{\text{out}})}}{\sqrt{C_{\max} \cdot d_{\text{in}} d_{\text{out}}}}$$

$$= \sqrt{\frac{r(d_{\text{in}} + d_{\text{out}})}{d_{\text{in}} d_{\text{out}}}}$$

$$= \sqrt{r\left(\frac{1}{d_{\text{in}}} + \frac{1}{d_{\text{out}}}\right)}.$$

When $r \ll \min(d_{\text{in}}, d_{\text{out}})$, this ratio is much less than 1. $\qquad\square$

*Remark* D.5 (General Case with Different $C_{\max}$ Values). In general, when $C_{\max}^{\text{full}} \neq C_{\max}^{\text{LR}}$, the low-rank approach provides a tighter bound if:

$$C_{\max}^{\text{LR}} \cdot r(d_{\text{in}} + d_{\text{out}}) < C_{\max}^{\text{full}} \cdot d_{\text{in}} d_{\text{out}}.$$

This condition is typically satisfied when using the same variational family (e.g., Gaussians with comparable variances) for both parameterizations, since the per-parameter KL divergences are then of similar magnitude.

*Remark* D.6 (Precise KL for Low-Rank Factors). For the sake of complete rigor, with $A \in \mathbb{R}^{m \times r}$ and $B \in \mathbb{R}^{n \times r}$:

$$\text{KL}(q_A(A)\|p_A(A)) \leq C_{\max}^A \cdot m \cdot r, \tag{11}$$

$$\text{KL}(q_B(B)\|p_B(B)) \leq C_{\max}^B \cdot n \cdot r. \tag{12}$$

Therefore:

$$\text{KL}(Q_{\text{LR}}\|P_{\text{LR}}) \leq C_{\max}^A \cdot m \cdot r + C_{\max}^B \cdot n \cdot r. \tag{13}$$

When $C_{\max}^A = C_{\max}^B = C_{\max}^{\text{LR}}$:

$$\text{KL}(Q_{\text{LR}}\|P_{\text{LR}}) \leq C_{\max}^{\text{LR}} \cdot r(n + m). \tag{14}$$

### D.4. Interpretation

*Remark* D.7 (Generalization Benefits of Low-Rank). Even if the empirical risk $\hat{L}(Q_{\text{LR}})$ is slightly higher than $\hat{L}(Q_{\text{full}})$ (due to rank restriction), the low-rank posterior may still yield a **tighter generalization bound** because its complexity penalty scales as $r(d_{\text{in}} + d_{\text{out}})$ rather than $d_{\text{in}} d_{\text{out}}$.

This advantage is most pronounced in **data-limited regimes** where $N \ll d_{\text{in}} d_{\text{out}}$, making the complexity term a dominant contributor to the upper bound on true risk. The parameter reduction from $O(d_{\text{in}} d_{\text{out}})$ to $O(r(d_{\text{in}} + d_{\text{out}}))$ directly translates to better generalization guarantees.

## E. Gaussian Complexity Transfer: Deterministic to Bayesian

### E.1. Motivation and Positioning

This section addresses a specific conceptual gap: *Can deterministic rank-sensitive Gaussian complexity bounds be extended to Bayesian predictive means?*

**The problem.** Recent work by Pinto et al. (2025) has established sharp generalization bounds for deep networks with low-rank weight constraints using vector-valued Gaussian complexity. These bounds scale with the rank $r$ rather than the ambient dimension, providing dramatically tighter guarantees when $r$ is small. However, these results apply to *deterministic* predictors with fixed rank-$r$ weights.

In Bayesian neural networks, the deployed predictor is the posterior predictive mean $f_{\text{BNN}}(x) = \mathbb{E}_{W \sim q}[f_W(x)]$. Even when every sample $W \sim q$ has rank at most $r$, the posterior mean does not preserve this rank structure at the parameter level. Therefore, deterministic rank-based bounds do not immediately apply.

**The solution.** We show that the posterior predictive mean belongs to the *closed convex hull* of the class of rank-$r$ networks, and that vector-valued Gaussian complexity is invariant under convex hull and closure operations. This allows deterministic rank-sensitive bounds to transfer without degradation to Bayesian predictors.

**Relationship to PAC-Bayes.** This result complements PAC-Bayes analysis. PAC-Bayes provides posterior-dependent, information-theoretic bounds through KL divergence and, as we saw in Section D, the PAC-Bayes bound is effectively rank-aware with the complexity scaling as $r(d_{\text{in}} + d_{\text{out}})$ rather than $d_{\text{in}}d_{\text{out}}$. We believe our contribution is architectural: it enables rank-sensitive deterministic bounds to apply to Bayesian predictive averaging, despite rank destruction under expectation.

### E.2. Preliminaries

#### E.2.1. LOW-RANK PARAMETERIZATION

For $i = 1, \ldots, D$, each layer weight is parameterized as

$$W_i = A_i B_i^\top, \qquad A_i \in \mathbb{R}^{h_i \times r_i}, \;\; B_i \in \mathbb{R}^{h_{i-1} \times r_i}, \qquad r_i \leq \min(h_i, h_{i-1}).$$

Hence $\text{rank}(W_i) \leq r_i$ by construction.

#### E.2.2. VARIATIONAL POSTERIOR

We consider a factorized variational posterior over factors:

$$q(A, B \mid \theta) = \prod_{i=1}^{D} q_A(A_i \mid \theta_i^A)\, q_B(B_i \mid \theta_i^B), \qquad A = (A_1, \ldots, A_D), \;\; B = (B_1, \ldots, B_D).$$

#### E.2.3. BNN PREDICTOR

Given $(A, B)$ and induced weights $W_i = A_i B_i^\top$, let $f_{A,B} : \mathbb{R}^d \to \mathbb{R}^k$ denote the corresponding network (with fixed activation $\phi$ applied coordinate-wise). The posterior-mean predictor is

$$f_{\text{BNN}}(x; \theta) = \mathbb{E}_{(A,B) \sim q(\cdot \mid \theta)}\big[f_{A,B}(x)\big].$$

#### E.2.4. HYPOTHESIS CLASSES

We define three function classes that will be related through inclusions.

**Definition E.1** (Deterministic low-rank class of Pinto et al. (2025)). Fix rank bounds $r = (r_1, \ldots, r_D)$ and spectral norm bounds $C = (C_1, \ldots, C_D)$. Let $\mathcal{F}_D^{\text{Pinto}}(C, r)$ denote the set of depth-$D$ networks

$$\mathcal{F}_D^{\text{Pinto}}(C, r) = \{f_W : \; \|W_i\|_2 \leq C_i, \; \text{rank}(W_i) \leq r_i, \; i = 1, \ldots, D\}.$$

**Definition E.2** (Posterior support class). Define the *support class* induced by $q(\cdot \mid \theta)$ as

$$\mathcal{F}_D^{\text{supp}}(\theta) = \big\{f_{A,B} : \; (A_i, B_i) \in \text{supp}(q_A(\cdot \mid \theta_i^A)) \times \text{supp}(q_B(\cdot \mid \theta_i^B)) \;\; \forall i\big\}.$$

Note that this class is deterministic: $A$ and $B$ are not sampled from $q$, they are deterministic matrices chosen to belong to the support of $q$.

**Definition E.3** (Posterior-mean hypothesis class). Define the class containing the posterior-mean predictor as the *closed convex hull* of the support class:

$$\mathcal{F}_D^{\text{BNN}}(\theta) := \overline{\text{conv}\big(\mathcal{F}_D^{\text{supp}}(\theta)\big)}.$$

The closure is taken in the pointwise topology on $\mathbb{R}^d$. Equivalently, it is sufficient to take closure on the finite set $\{x_1, \ldots, x_m\}$ for generalization analysis (see Lemma E.10).

Our goal is to control the generalization of the specific predictor $f_{\text{BNN}}(\cdot; \theta)$, but it will be convenient to bound the complexity of $\mathcal{F}_D^{\text{BNN}}(\theta)$ via $\mathcal{F}_D^{\text{Pinto}}(C, r)$.

### E.3. Bounded Factor Support and Induced Weight Constraints

To connect $\mathcal{F}_D^{\text{supp}}(\theta)$ with $\mathcal{F}_D^{\text{Pinto}}(C, r)$, we require that the variational posterior has support contained within spectral norm balls.

**Assumption E.4** (Bounded factor support). For each layer $i = 1, \ldots, D$, there exist constants $C_i^A, C_i^B > 0$ such that

$$\text{supp}(q_A(\cdot \mid \theta_i^A)) \subseteq \{A_i : \|A_i\|_2 \leq C_i^A\}, \qquad \text{supp}(q_B(\cdot \mid \theta_i^B)) \subseteq \{B_i : \|B_i\|_2 \leq C_i^B\}.$$

Equivalently, $\|A_i\|_2 \leq C_i^A$ and $\|B_i\|_2 \leq C_i^B$ hold with probability 1 under $q(A, B \mid \theta)$.

This is an almost-sure requirement. We will discuss practical enforcement and high-probability alternatives in Sections E.11 and E.10.

**Lemma E.5** (Induced rank and spectral norm bounds for $W_i = A_i B_i^\top$). *Under Assumption E.4, define $C_i := C_i^A C_i^B$. Then for each layer $i$, every realization $W_i = A_i B_i^\top$ satisfies*

$$\text{rank}(W_i) \leq r_i, \qquad \|W_i\|_2 \leq C_i.$$

*Proof.* The rank bound is immediate from $\text{rank}(A_i B_i^\top) \leq \min(\text{rank}(A_i), \text{rank}(B_i)) \leq r_i$.

For the spectral norm, we use the submultiplicativity property:

$$\|W_i\|_2 = \|A_i B_i^\top\|_2 \leq \|A_i\|_2 \|B_i^\top\|_2 = \|A_i\|_2 \|B_i\|_2 \leq C_i^A C_i^B = C_i.$$

$\square$

**Lemma E.6** (Support class inclusion into Pinto's class). *Under Assumption E.4,*

$$\mathcal{F}_D^{\text{supp}}(\theta) \subseteq \mathcal{F}_D^{\text{Pinto}}(C, r) \qquad \text{with } C_i = C_i^A C_i^B.$$

*Proof.* Take any $f_{A,B} \in \mathcal{F}_D^{\text{supp}}(\theta)$. By Definition E.2, each factor realization lies in the corresponding support set, hence satisfies $\|A_i\|_2 \leq C_i^A$ and $\|B_i\|_2 \leq C_i^B$ by Assumption E.4.

Lemma E.5 yields $\|W_i\|_2 \leq C_i$ and $\text{rank}(W_i) \leq r_i$ for all $i = 1, \ldots, D$.

Therefore the corresponding network belongs to $\mathcal{F}_D^{\text{Pinto}}(C, r)$ by Definition E.1. $\square$

### E.4. Vector-Valued Gaussian Complexity and Basic Properties

We now define the empirical vector-valued Gaussian complexity and establish the convex-hull and closure invariances that are essential for our main result.

**Definition E.7** (Empirical vector-valued Gaussian complexity). Let $\mathcal{F}$ be a class of functions $f : \mathbb{R}^d \to \mathbb{R}^k$. Given sample inputs $S_x = \{x_1, \ldots, x_m\}$, define

$$\hat{\mathcal{G}}_{S_x}(\mathcal{F}) := \mathbb{E}_\Gamma \left[ \sup_{f \in \mathcal{F}} \frac{1}{m} \sum_{j=1}^m \langle \gamma_j, f(x_j) \rangle \right],$$

where $\gamma_1, \ldots, \gamma_m \overset{\text{iid}}{\sim} \mathcal{N}(0, I_k)$ and $\Gamma = (\gamma_1, \ldots, \gamma_m)$.

**Lemma E.8** (Monotonicity). *If $\mathcal{F} \subseteq \mathcal{G}$ then $\hat{\mathcal{G}}_{S_x}(\mathcal{F}) \leq \hat{\mathcal{G}}_{S_x}(\mathcal{G})$.*

*Proof.* Fix a realization of $\Gamma = (\gamma_1, \ldots, \gamma_m)$. Define the functional

$$\Phi(f) := \frac{1}{m} \sum_{j=1}^m \langle \gamma_j, f(x_j) \rangle.$$

Since $\mathcal{F} \subseteq \mathcal{G}$, we have

$$\sup_{f \in \mathcal{F}} \Phi(f) \leq \sup_{g \in \mathcal{G}} \Phi(g)$$

for every realization of $\Gamma$.

Taking expectations over $\Gamma$ on both sides:

$$\mathbb{E}_\Gamma \left[ \sup_{f \in \mathcal{F}} \Phi(f) \right] \leq \mathbb{E}_\Gamma \left[ \sup_{g \in \mathcal{G}} \Phi(g) \right],$$

which is precisely $\hat{\mathcal{G}}_{S_x}(\mathcal{F}) \leq \hat{\mathcal{G}}_{S_x}(\mathcal{G})$. $\qquad\square$

**Lemma E.9** (Convex hull invariance). *For any class $\mathcal{F}$,*

$$\hat{\mathcal{G}}_{S_x}(\mathrm{conv}(\mathcal{F})) = \hat{\mathcal{G}}_{S_x}(\mathcal{F}).$$

*Proof.* The inequality $\hat{\mathcal{G}}_{S_x}(\mathcal{F}) \leq \hat{\mathcal{G}}_{S_x}(\mathrm{conv}(\mathcal{F}))$ follows immediately from $\mathcal{F} \subseteq \mathrm{conv}(\mathcal{F})$ and Lemma E.8.

For the reverse inequality, fix $\Gamma$ and let $g \in \mathrm{conv}(\mathcal{F})$. Then by definition of the convex hull, there exist finitely many functions $f^{(1)}, \ldots, f^{(T)} \in \mathcal{F}$ and coefficients $\alpha_1, \ldots, \alpha_T \geq 0$ with $\sum_{t=1}^T \alpha_t = 1$ such that

$$g = \sum_{t=1}^T \alpha_t f^{(t)}.$$

By linearity of the inner product and summation:

$$\Phi(g) = \frac{1}{m} \sum_{j=1}^m \left\langle \gamma_j, \sum_{t=1}^T \alpha_t f^{(t)}(x_j) \right\rangle = \sum_{t=1}^T \alpha_t \left( \frac{1}{m} \sum_{j=1}^m \langle \gamma_j, f^{(t)}(x_j) \rangle \right) = \sum_{t=1}^T \alpha_t \Phi(f^{(t)}).$$

Since $\alpha_t \geq 0$ and $\sum_{t=1}^T \alpha_t = 1$, this is a convex combination:

$$\Phi(g) = \sum_{t=1}^T \alpha_t \Phi(f^{(t)}) \leq \sum_{t=1}^T \alpha_t \sup_{f \in \mathcal{F}} \Phi(f) = \sup_{f \in \mathcal{F}} \Phi(f).$$

Taking the supremum over all $g \in \mathrm{conv}(\mathcal{F})$:

$$\sup_{g \in \mathrm{conv}(\mathcal{F})} \Phi(g) \leq \sup_{f \in \mathcal{F}} \Phi(f).$$

This holds for every realization of $\Gamma$, so taking expectations:

$$\hat{\mathcal{G}}_{S_x}(\mathrm{conv}(\mathcal{F})) \leq \hat{\mathcal{G}}_{S_x}(\mathcal{F}).$$

Combined with the forward inequality, we obtain equality. $\qquad\square$

**Lemma E.10** (Closure invariance on the sample). *Let $\overline{\mathcal{F}}^{S_x}$ be the closure of $\mathcal{F}$ under pointwise convergence on $S_x$ (i.e., closure of evaluation vectors in $\mathbb{R}^{k \times m}$). Then*

$$\hat{\mathcal{G}}_{S_x}(\overline{\mathcal{F}}^{S_x}) = \hat{\mathcal{G}}_{S_x}(\mathcal{F}).$$

*Consequently,*

$$\hat{\mathcal{G}}_{S_x}(\overline{\mathrm{conv}(\mathcal{F})}^{S_x}) = \hat{\mathcal{G}}_{S_x}(\mathrm{conv}(\mathcal{F})) = \hat{\mathcal{G}}_{S_x}(\mathcal{F}).$$

*Proof.* Fix a realization $\Gamma = (\gamma_1, \ldots, \gamma_m)$. The functional

$$\Phi(f) = \frac{1}{m} \sum_{j=1}^m \langle \gamma_j, f(x_j) \rangle$$

depends only on the evaluation vector $(f(x_1), \ldots, f(x_m)) \in \mathbb{R}^{k \times m}$.

We can rewrite $\Phi$ as:

$$\Phi(f) = \left\langle \frac{1}{m} \sum_{j=1}^{m} \gamma_j \otimes e_j, (f(x_1), \ldots, f(x_m)) \right\rangle_{\mathbb{R}^{k \times m}},$$

where $e_j$ is the $j$-th standard basis vector in $\mathbb{R}^m$ and $\otimes$ denotes the outer product. This is a linear functional on $\mathbb{R}^{k \times m}$, hence continuous with respect to the Euclidean topology.

Since $\Phi$ is continuous in the evaluation vector, and $\overline{\mathcal{F}}^{S_x}$ is precisely the closure of $\mathcal{F}$ with respect to pointwise convergence on $S_x$, we have:

$$\sup_{f \in \overline{\mathcal{F}}^{S_x}} \Phi(f) = \sup_{f \in \mathcal{F}} \Phi(f).$$

This holds for every realization of $\Gamma$, so taking expectations:

$$\hat{\mathcal{G}}_{S_x}(\overline{\mathcal{F}}^{S_x}) = \hat{\mathcal{G}}_{S_x}(\mathcal{F}).$$

For the second statement, apply Lemma E.9 to obtain $\hat{\mathcal{G}}_{S_x}(\text{conv}(\mathcal{F})) = \hat{\mathcal{G}}_{S_x}(\mathcal{F})$, then apply the first part with $\mathcal{F}$ replaced by $\text{conv}(\mathcal{F})$ to get:

$$\hat{\mathcal{G}}_{S_x}(\overline{\text{conv}(\mathcal{F})}^{S_x}) = \hat{\mathcal{G}}_{S_x}(\text{conv}(\mathcal{F})) = \hat{\mathcal{G}}_{S_x}(\mathcal{F}).$$

$\square$

### E.5. Posterior Mean Lies in Closed Convex Hull

**Lemma E.11** (Posterior mean predictor is in the closed convex hull). *Assume that for each fixed $x \in \mathbb{R}^d$, the random vector $f_{A,B}(x)$ is integrable: $\mathbb{E}\|f_{A,B}(x)\|_2 < \infty$. Then*

$$f_{\text{BNN}}(\cdot; \theta) \in \overline{\text{conv}(\mathcal{F}_D^{\text{supp}}(\theta))}.$$

*Proof.* Let $(A^{(t)}, B^{(t)})_{t \geq 1}$ be i.i.d. samples from $q(\cdot \mid \theta)$ and define the Monte Carlo predictor

$$\hat{f}_M(x) := \frac{1}{M} \sum_{t=1}^{M} f_{A^{(t)}, B^{(t)}}(x).$$

For each $M$, the predictor $\hat{f}_M$ is a convex combination of elements of $\mathcal{F}_D^{\text{supp}}(\theta)$. To see this, note that each $(A^{(t)}, B^{(t)})$ is drawn from the support of $q(\cdot \mid \theta)$, hence by Definition E.2, $f_{A^{(t)}, B^{(t)}} \in \mathcal{F}_D^{\text{supp}}(\theta)$ for all $t = 1, \ldots, M$.

Therefore,

$$\hat{f}_M = \frac{1}{M} \sum_{t=1}^{M} f_{A^{(t)}, B^{(t)}} \in \text{conv}(\mathcal{F}_D^{\text{supp}}(\theta)).$$

By the strong law of large numbers, applied coordinate-wise in $\mathbb{R}^k$, and using the integrability assumption $\mathbb{E}\|f_{A,B}(x)\|_2 < \infty$, we have for each fixed $x \in \mathbb{R}^d$:

$$\hat{f}_M(x) \to \mathbb{E}_{(A,B) \sim q(\cdot|\theta)}[f_{A,B}(x)] = f_{\text{BNN}}(x; \theta) \quad \text{almost surely as } M \to \infty.$$

Since $\hat{f}_M \in \text{conv}(\mathcal{F}_D^{\text{supp}}(\theta))$ for all $M$, and $\hat{f}_M$ converges pointwise to $f_{\text{BNN}}$, we conclude that

$$f_{\text{BNN}}(\cdot; \theta) \in \overline{\text{conv}(\mathcal{F}_D^{\text{supp}}(\theta))},$$

where the closure is taken in the pointwise topology (or equivalently, in the topology of pointwise convergence on any countable dense subset of $\mathbb{R}^d$, which includes the finite sample set $\{x_1, \ldots, x_m\}$). $\square$

*Remark* E.12 (Integrability assumption). The integrability condition $\mathbb{E}\|f_{A,B}(x)\|_2 < \infty$ is satisfied in practice when:

- The factors $A_i, B_i$ have finite second moments (e.g., Gaussian variational posteriors), and

- The network depth $D$ and activation functions are such that the composition does not produce exponential growth.

Under Assumption E.4, which requires almost-sure spectral norm bounds on the factors, this integrability is automatically satisfied since the network outputs are then almost-surely bounded.

### E.6. Complexity Chain: From BNN Class to Pinto's Class

We now establish the key inclusion and complexity inequality that connects the Bayesian predictive mean to Pinto's deterministic class.

**Proposition E.13** (Gaussian complexity chain for the BNN class). *Under Assumption E.4, with $C_i = C_i^A C_i^B$,*

$$\hat{\mathcal{G}}_{S_x}\big(\mathcal{F}_D^{\mathrm{BNN}}(\theta)\big) \leq \hat{\mathcal{G}}_{S_x}\big(\mathcal{F}_D^{\mathrm{Pinto}}(C,r)\big).$$

*Moreover,* $\mathcal{F}_D^{\mathrm{BNN}}(\theta) \subseteq \overline{\mathrm{conv}}\big(\mathcal{F}_D^{\mathrm{Pinto}}(C,r)\big).$

*Proof.* We establish the result through a chain of inclusions and applications of the invariance lemmas.

**Step 1: Support class is contained in Pinto's class.**

By Lemma E.6, under Assumption E.4,
$$\mathcal{F}_D^{\mathrm{supp}}(\theta) \subseteq \mathcal{F}_D^{\mathrm{Pinto}}(C,r).$$

**Step 2: Convex hull and closure preserve inclusion.**

Taking the convex hull of both sides:
$$\mathrm{conv}\big(\mathcal{F}_D^{\mathrm{supp}}(\theta)\big) \subseteq \mathrm{conv}\big(\mathcal{F}_D^{\mathrm{Pinto}}(C,r)\big).$$

Taking the closure (on the sample $S_x$) of both sides:
$$\overline{\mathrm{conv}\big(\mathcal{F}_D^{\mathrm{supp}}(\theta)\big)}^{S_x} \subseteq \overline{\mathrm{conv}\big(\mathcal{F}_D^{\mathrm{Pinto}}(C,r)\big)}^{S_x}.$$

By Definition E.3, the left-hand side is $\mathcal{F}_D^{\mathrm{BNN}}(\theta)$, so:
$$\mathcal{F}_D^{\mathrm{BNN}}(\theta) \subseteq \overline{\mathrm{conv}}\big(\mathcal{F}_D^{\mathrm{Pinto}}(C,r)\big).$$

This establishes the second claim.

**Step 3: Apply closure and convex hull invariance for complexity.**

By Lemma E.10, the Gaussian complexity is invariant under closure on the sample:
$$\hat{\mathcal{G}}_{S_x}\big(\mathcal{F}_D^{\mathrm{BNN}}(\theta)\big) = \hat{\mathcal{G}}_{S_x}\big(\overline{\mathrm{conv}\big(\mathcal{F}_D^{\mathrm{supp}}(\theta)\big)}^{S_x}\big) = \hat{\mathcal{G}}_{S_x}\big(\mathrm{conv}\big(\mathcal{F}_D^{\mathrm{supp}}(\theta)\big)\big).$$

By Lemma E.9, the Gaussian complexity is invariant under convex hull:
$$\hat{\mathcal{G}}_{S_x}\big(\mathrm{conv}\big(\mathcal{F}_D^{\mathrm{supp}}(\theta)\big)\big) = \hat{\mathcal{G}}_{S_x}\big(\mathcal{F}_D^{\mathrm{supp}}(\theta)\big).$$

Combining:
$$\hat{\mathcal{G}}_{S_x}\big(\mathcal{F}_D^{\mathrm{BNN}}(\theta)\big) = \hat{\mathcal{G}}_{S_x}\big(\mathcal{F}_D^{\mathrm{supp}}(\theta)\big).$$

**Step 4: Apply monotonicity.**

Since $\mathcal{F}_D^{\text{supp}}(\theta) \subseteq \mathcal{F}_D^{\text{Pinto}}(C, r)$ by Step 1, Lemma E.8 gives:

$$\hat{\mathcal{G}}_{S_x}\big(\mathcal{F}_D^{\text{supp}}(\theta)\big) \leq \hat{\mathcal{G}}_{S_x}\big(\mathcal{F}_D^{\text{Pinto}}(C, r)\big).$$

Combining with Step 3:

$$\hat{\mathcal{G}}_{S_x}\big(\mathcal{F}_D^{\text{BNN}}(\theta)\big) \leq \hat{\mathcal{G}}_{S_x}\big(\mathcal{F}_D^{\text{Pinto}}(C, r)\big).$$

This completes the proof. $\square$

*Remark* E.14 (Significance of the complexity chain). This proposition is the central technical result enabling the transfer of deterministic bounds to Bayesian predictors. It shows that:

1. The BNN predictor class has the same Gaussian complexity as the support class, despite being a much larger set (closed convex hull vs. original class).

2. The complexity is bounded by that of Pinto's deterministic class, allowing us to invoke existing deterministic bounds.

3. No degradation occurs in the complexity bound from the deterministic to the Bayesian setting.

### E.7. Generalization from Vector-Valued Gaussian Complexity

We now state the generalization theorem from Pinto et al. (2025) for vector-valued Gaussian complexity and Lipschitz losses, which we will apply to our BNN predictor.

**Theorem E.15** (Generalization via vector-valued Gaussian complexity, Pinto et al. (2025)). *Let $\ell(\cdot, y) \in [0, 1]$ be $\mathcal{L}$-Lipschitz in its first argument. Let $\mathcal{F}$ be a class of functions $\mathbb{R}^d \to \mathbb{R}^k$. Then, with probability at least $1 - \delta$ over $S \sim \mathcal{D}^m$, for all $f \in \mathcal{F}$,*

$$\mathbb{E}\big[\ell(f(x), y)\big] \leq \frac{1}{m}\sum_{j=1}^{m} \ell(f(x_j), y_j) + \sqrt{\pi}\mathcal{L}\,\hat{\mathcal{G}}_{S_x}(\mathcal{F}) + 3\sqrt{\frac{\log(2/\delta)}{2m}}.$$

*Remark* E.16 (Origin and scope of the theorem). Theorem E.15 combines two results from Pinto et al. (2025):

1. A contraction inequality showing that Lipschitz losses contract the vector-valued Gaussian complexity by at most $\sqrt{\pi}\mathcal{L}$.

2. A concentration inequality relating the empirical Gaussian complexity to generalization.

We cite it in this combined form to avoid re-deriving technical constants. Our contribution is the class inclusion argument (Proposition E.13) that enables applying this theorem to $f_{\text{BNN}}$.

E.7.1. EXPLICIT COMPLEXITY BOUND FOR PINTO'S CLASS

We now recall the explicit upper bound for $\hat{\mathcal{G}}_{S_x}(\mathcal{F}_D^{\text{Pinto}}(C, r))$ derived by Pinto et al. (2025).

**Theorem E.17** (Explicit bound for rank- and norm-constrained networks, Pinto et al. (2025)). *For $\mathcal{F}_D^{\text{Pinto}}(C, r)$ as in Definition E.1, the following bound holds:*

$$\hat{\mathcal{G}}_{S_x}\big(\mathcal{F}_D^{\text{Pinto}}(C, r)\big) \leq \frac{\|X\|_F}{m}\left\{ B_1\sqrt{h_1}\prod_{i=1}^{D} C_0 C_i + \sum_{j=2}^{D} C_j \sqrt{r_j h_j}\prod_{i=j+1}^{D} C_0 C_i \right\},$$

*where:*

- $C_0 > 0$ *is the universal constant from Maurer's chaining bound for Lipschitz activations,*

- $\|X\|_F^2 = \sum_{j=1}^{m}\|x_j\|_2^2$ *is the Frobenius norm of the input data matrix,*

- $h_\ell$ *denotes the width of layer $\ell$,*

- $r_j$ is the rank bound for layer $j$,

- $B_1 := \sup_{f_W \in \mathcal{F}_D^{\text{Pinto}}(C,r)} \|W_1\|_F$ is the worst-case Frobenius norm of the first-layer weight matrix over the class. Since $\text{rank}(W_1) \leq r_1$ and $\|W_1\|_2 \leq C_1$ for all members of the class, we have $B_1 \leq \sqrt{r_1} C_1$.

*Remark* E.18 (Key features of the bound).    1. **Rank dependence**: The complexity scales as $\sqrt{r_j h_j}$ for each layer $j \geq 2$, rather than the full dimension $h_j h_{j-1}$ that would appear in a full-rank analysis.

2. **Data dependence**: The bound explicitly depends on $\|X\|_F$, the norm of the training inputs, unlike PAC-Bayes bounds which depend on the posterior-prior KL divergence.

3. **Depth dependence**: The product $\prod_{i=j+1}^{D} C_0 C_i$ shows how spectral norms accumulate through the network depth.

4. **First layer**: The term $B_1 \sqrt{h_1}$ for the first layer differs in form from the $\sqrt{r_j h_j}$ terms for deeper layers. This is an artifact of the layer-peeling proof technique (specifically, how the chaining argument handles the input layer), not a reflection of a missing rank constraint. The rank of the first layer still enters through $B_1 \leq \sqrt{r_1} C_1$, so that the first-layer contribution is at most $\sqrt{r_1} C_1 \sqrt{h_1} \prod_{i=1}^{D} C_0 C_i$, which has the same $\sqrt{r_1}$ dependence as deeper layers.

*Remark* E.19 (Comparison with standard Rademacher bounds). Standard Rademacher complexity bounds for deep networks without rank constraints typically scale as

$$O\left( \frac{\|X\|_F}{m} \prod_{i=1}^{D} C_i \sqrt{h_i h_{i-1}} \right).$$

The rank constraint reduces $\sqrt{h_i h_{i-1}}$ to $\sqrt{r_i h_i}$ in the bound, providing improvement when $r_i \ll h_{i-1}$.

**Note on depth dependence:** Both the rank-constrained bound from Theorem E.17 and standard bounds contain a product over depth, specifically the factor $\prod_{i=j}^{D} C_0 C_i$ which grows as $C_0^{D-j+1} \prod_{i=j}^{D} C_i$. As noted by Pinto et al. (2025), this unfavorable exponential factor in depth may be an artifact of the current proof technique rather than a fundamental limitation. When comparing the rank-constrained bound to existing bounds, the key improvement comes from the rank-dependent terms $\sqrt{r_j h_j}$ rather than from the depth-dependent product, which appears in both settings.

### E.8. Proof of Theorem 3.9

*Proof.* We prove that the BNN posterior predictive mean satisfies the same generalization bound as Pinto's deterministic class.

**Step 1: The BNN predictor belongs to the BNN class.**

By Lemma E.11, under the integrability assumption $\mathbb{E}\|f_{A,B}(x)\|_2 < \infty$,

$$f_{\text{BNN}}(\cdot; \theta) \in \mathcal{F}_D^{\text{BNN}}(\theta).$$

**Step 2: Apply Pinto's generalization theorem to the BNN class.**

We apply Theorem E.15 with the hypothesis class $\mathcal{F} = \mathcal{F}_D^{\text{BNN}}(\theta)$.

With probability at least $1 - \delta$ over $S \sim \mathcal{D}^m$, the bound holds uniformly for all $f \in \mathcal{F}_D^{\text{BNN}}(\theta)$:

$$\mathbb{E}\big[\ell(f(x), y)\big] \leq \frac{1}{m} \sum_{j=1}^{m} \ell(f(x_j), y_j) + \sqrt{\pi} \mathcal{L} \, \hat{\mathcal{G}}_{S_x}\big(\mathcal{F}_D^{\text{BNN}}(\theta)\big) + 3\sqrt{\frac{\log(2/\delta)}{2m}}.$$

In particular, since $f_{\text{BNN}}(\cdot; \theta) \in \mathcal{F}_D^{\text{BNN}}(\theta)$ by Step 1, this bound holds for $f = f_{\text{BNN}}(\cdot; \theta)$.

**Step 3: Bound the Gaussian complexity using Proposition E.13.**

By Proposition E.13, under Assumption E.4,

$$\hat{\mathcal{G}}_{S_x}\big(\mathcal{F}_D^{\text{BNN}}(\theta)\big) \leq \hat{\mathcal{G}}_{S_x}\big(\mathcal{F}_D^{\text{Pinto}}(C, r)\big),$$

where $C_i = C_i^A C_i^B$ for each layer $i = 1, \ldots, D$.

**Step 4: Apply the explicit bound from Theorem E.17.**

Substituting the explicit bound from Theorem E.17, and using $B_1 \leq \sqrt{r_1}\, C_1$:

$$\hat{\mathcal{G}}_{S_x}\left(\mathcal{F}_D^{\text{Pinto}}(C, r)\right) \leq \frac{\|X\|_F}{m}\left\{\sqrt{r_1}\, C_1 \sqrt{h_1} \prod_{i=1}^{D} C_0 C_i + \sum_{j=2}^{D} C_j \sqrt{r_j h_j} \prod_{i=j+1}^{D} C_0 C_i\right\}.$$

**Step 5: Combine to obtain the final bound.**

Combining Steps 2, 3, and 4, we obtain that with probability at least $1 - \delta$ over $S \sim \mathcal{D}^m$,

$$\mathbb{E}\left[\ell(f_{\text{BNN}}(x), y)\right] \leq \frac{1}{m} \sum_{j=1}^{m} \ell(f_{\text{BNN}}(x_j), y_j)$$

$$+ \sqrt{\pi \mathcal{L}} \cdot \frac{\|X\|_F}{m}\left\{\sqrt{r_1}\, C_1 \sqrt{h_1} \prod_{i=1}^{D} C_0 C_i + \sum_{j=2}^{D} C_j \sqrt{r_j h_j} \prod_{i=j+1}^{D} C_0 C_i\right\}$$

$$+ 3\sqrt{\frac{\log(2/\delta)}{2m}}.$$

This completes the proof. □

*Remark* E.20 (Key insight of the proof). The proof relies on three important properties:

1. **Membership**: The BNN predictor belongs to the closed convex hull of the support class (Lemma E.11).

2. **Invariance**: Vector-valued Gaussian complexity is invariant under convex hull and closure (Lemmas E.9 and E.10).

3. **Inclusion**: The support class is contained in Pinto's deterministic class under spectral norm bounds (Lemma E.6).

These three properties together allow the deterministic rank-sensitive bound to transfer without degradation to the Bayesian predictive mean, despite the fact that rank is not preserved under expectation.

### E.9. Comparison with PAC-Bayes Bounds

The Gaussian complexity bounds we have established complement rather than replace PAC-Bayes bounds. Here we compare their structure and highlight when each approach provides better guarantees.

**Proposition E.21** (Scaling Comparison). *Consider a single-layer network with input dimension $d_{in}$, output dimension $d_{out}$, and rank $r$. Let $\ell(\cdot, y) \in [0, 1]$ be $\mathcal{L}$-Lipschitz in its first argument.*

***PAC-Bayes bound (from Theorem 3.8):*** *The generalization gap is bounded by*

$$Gap_{PAC} \leq \sqrt{\frac{C_{\max} \cdot r(d_{in} + d_{out}) + \log(2\sqrt{m}/\delta)}{2m}}, \tag{15}$$

*where $C_{\max}$ is the maximum per-parameter KL divergence between the posterior and the prior. The structural scaling (suppressing logarithmic terms, $C_{\max}$, and constants) is*

$$Gap_{PAC} = O\left(\sqrt{\frac{r(d_{in} + d_{out})}{m}}\right).$$

***Gaussian complexity bound (from Theorem 3.9):*** *The generalization gap is bounded by*

$$Gap_{GC} \leq \sqrt{\pi \mathcal{L}} \cdot \frac{\|X\|_F \cdot C_0 C_1^2 \cdot \sqrt{r\, d_{out}}}{m} + 3\sqrt{\frac{\log(2/\delta)}{2m}}, \tag{16}$$

*where $C_0$ is the universal chaining constant and $C_1$ is the spectral norm bound.*

*If inputs are uniformly bounded as $\|x_j\|_2 \leq R$, then $\|X\|_F \leq R\sqrt{m}$ and the rank-dependent term simplifies to $O(\sqrt{r\,d_{out}/m})$.*

*Proof.* **PAC-Bayes analysis.** The complexity term in Theorem 3.8 is

$$\sqrt{\frac{\text{KL}(Q_{\text{LR}}\|P_{\text{LR}}) + \log(2\sqrt{m}/\delta)}{2m}} \leq \sqrt{\frac{C_{\max} \cdot r(d_{\text{in}} + d_{\text{out}}) + \log(2\sqrt{m}/\delta)}{2m}},$$

which yields the expression in (15). Dropping logarithmic terms and constants gives the structural scaling $O(\sqrt{r(d_{\text{in}} + d_{\text{out}})/m})$.

**Gaussian complexity analysis.** For a single-layer network (depth $D = 1$), Theorem E.17 gives

$$\hat{\mathcal{G}}_{S_x}(\mathcal{F}_1^{\text{Pinto}}) \leq \frac{\|X\|_F}{m} \cdot B_1\sqrt{h_1} \cdot C_0 C_1,$$

where $h_1 = d_{\text{out}}$. Since $B_1 \leq \sqrt{r}\,C_1$ (see Theorem E.17), substituting yields

$$\hat{\mathcal{G}}_{S_x}(\mathcal{F}_1^{\text{Pinto}}) \leq \frac{\|X\|_F}{m} \cdot \sqrt{r}\,C_1 \cdot \sqrt{d_{\text{out}}} \cdot C_0 C_1 = \frac{\|X\|_F \cdot C_0 C_1^2 \cdot \sqrt{r\,d_{\text{out}}}}{m}.$$

From Theorem E.15, the full generalization bound is

$$\text{Gap}_{\text{GC}} \leq \sqrt{\pi\mathcal{L}} \cdot \frac{\|X\|_F \cdot C_0 C_1^2 \cdot \sqrt{r\,d_{\text{out}}}}{m} + 3\sqrt{\frac{\log(2/\delta)}{2m}},$$

which is (16). The first (rank-dependent) term decays as $1/m$ for fixed $\|X\|_F$, while the second (concentration) term decays as $1/\sqrt{m}$.

If we additionally assume $\|x_j\|_2 \leq R$ for all $j$, then $\|X\|_F \leq R\sqrt{m}$, so the rank-dependent term becomes

$$\sqrt{\pi\mathcal{L}} \cdot C_0 C_1^2 \cdot R \cdot \sqrt{\frac{r\,d_{\text{out}}}{m}}.$$

$\square$

*Remark* E.22 (Structural differences). The PAC-Bayes and Gaussian complexity bounds have fundamentally different structures:

**PAC-Bayes:** Single term with uniform $\frac{1}{\sqrt{m}}$ scaling

$$\text{Gap}_{\text{PAC}} = O\left(\sqrt{\frac{\text{KL divergence}}{m}}\right).$$

**Gaussian complexity:** Two terms with different scalings

$$\text{Gap}_{\text{GC}} = O\left(\underbrace{\frac{\text{Complexity} \times \|X\|_F}{m}}_{\text{faster decay in } m}\right) + \underbrace{O\left(\frac{1}{\sqrt{m}}\right)}_{\text{slower decay in } m}.$$

For large $m$, the slower-decaying $O(1/\sqrt{m})$ concentration term dominates both bounds, and the comparison reduces to constants. For moderate $m$, the different structural dependences on architecture and data can make either bound tighter.

**Important caveat:** The structural scaling comparison (suppressing constants) is useful for understanding how each bound depends on architectural parameters, but *does not determine which bound is numerically tighter* in any given setting. The suppressed constants—$C_{\max}$ for PAC-Bayes and $\sqrt{\pi\mathcal{L}}\,C_0 C_1^2$ for Gaussian complexity—can differ by orders of magnitude and must be evaluated for specific models.

### E.9.1. COMPLEMENTARY STRENGTHS

*Remark* E.23 (When PAC-Bayes is stronger). PAC-Bayes bounds are tighter when:

1. **Posterior concentration:** When the posterior is highly concentrated around a good solution, $C_{\max}$ can be very small, making the KL term tight. In the extreme case of a near-deterministic posterior, the PAC-Bayes gap approaches zero while the Gaussian complexity bound remains fixed by the class geometry.

2. **Large input norms:** PAC-Bayes bounds do not depend on $\|X\|_F$, which is advantageous when inputs have very large norm or heavy tails. The Gaussian complexity bound's leading term grows linearly in $\|X\|_F$.

3. **High-dimensional inputs with moderate rank:** Looking at the full expressions, PAC-Bayes scales with $r(d_{\text{in}} + d_{\text{out}})$ while Gaussian complexity (with bounded inputs) scales with $C_0^2 C_1^4 R^2 \cdot r \, d_{\text{out}}$. When $d_{\text{in}} \gg d_{\text{out}}$ and $C_0^2 C_1^4 R^2$ is large, PAC-Bayes can still be favorable despite the $d_{\text{in}}$ term, because it avoids the spectral norm constants entirely. Concretely, PAC-Bayes is structurally favorable when $C_{\max}(d_{\text{in}} + d_{\text{out}}) \lesssim \pi \mathcal{L} \, C_0^2 C_1^4 R^2 \, d_{\text{out}}$, i.e., when the posterior is sufficiently concentrated relative to the spectral norm and Lipschitz constants.

*Remark* E.24 (When Gaussian complexity is stronger). Gaussian complexity bounds can be tighter when:

1. **Diffuse posteriors with bounded spectral norms:** When $C_{\max}$ is large (the posterior has not concentrated tightly), the PAC-Bayes KL term can be vacuous. The Gaussian complexity bound depends on the spectral norm constraints $C_1$, not on posterior concentration, and can remain informative even when the posterior is spread over a large region of parameter space.

2. **Normalized inputs with small spectral norm constants:** When $\|x_j\|_2 \leq R$ with $R$ moderate and spectral norms are well-controlled ($C_1$ not too large), the Gaussian complexity bound's leading constant $\sqrt{\pi \mathcal{L}} \, C_0 C_1^2 R$ can be smaller than $\sqrt{C_{\max}/2}$. This is the regime where the Gaussian complexity bound is numerically tighter.

3. **No prior specification required:** Unlike PAC-Bayes, which requires choosing a prior $P$ and where the bound quality depends sensitively on this choice, Gaussian complexity bounds depend only on the realized network architecture and data. This makes them valuable as a *prior-free sanity check*.

4. **Architectural transparency:** The bound explicitly shows how rank, spectral norms, and layer dimensions interact through the network depth, providing guidance for architectural design that is not available from the PAC-Bayes bound's KL-divergence summary.

*Remark* E.25 (Rank dependence). Both bounds have $\sqrt{r}$ rank dependence in their leading terms:

- **PAC-Bayes:** The parameter-count reduction from $d_{\text{in}} d_{\text{out}}$ to $r(d_{\text{in}} + d_{\text{out}})$ appears inside a square root in the KL term.

- **Gaussian complexity:** The Frobenius norm bound $\|W\|_F \leq \sqrt{r}\|W\|_2$ for rank-$r$ matrices introduces $\sqrt{r}$ into the complexity.

Neither bound exhibits linear $r$ dependence; the rank always appears under a square root in the leading generalization term.

*Remark* E.26 (Complementary perspectives on generalization). The key insight is that these bounds measure different aspects of generalization:

- **PAC-Bayes** measures how much the posterior deviates from the prior (information-theoretic complexity).

- **Gaussian complexity** measures the richness of the hypothesis class on the actual training data (geometric complexity).

In practice, both bounds should be considered:

- Use PAC-Bayes to understand posterior–prior relationships and to guide prior selection.

- Use Gaussian complexity to understand architectural choices (rank, width, depth, spectral norms) and their interaction with the training data.

Our contribution shows that Bayesian predictive means can take advantage of *both* frameworks: they satisfy PAC-Bayes bounds through posterior analysis, and they satisfy Gaussian complexity bounds through the convex-hull structure established in this section.

### E.10. Alternative Approaches for High-Probability Spectral Norm Bounds

Assumption E.4 requires that the variational posterior has support contained within a spectral norm ball, that is, the bound $\|A_i\|_2 \le C_i^A$ must hold *almost surely* for every sample from the posterior. This assumption is essential for the submultiplicativity argument in Lemma E.5 which establishes that the induced weight matrices $W_i = A_i B_i^\top$ satisfy the spectral norm constraints required for $\mathcal{F}_D^{\text{Pinto}}$.

However, in practice, commonly used variational families such as Gaussian posteriors have unbounded support. While concentration inequalities (such as Van Handel's result in (Van Handel, 2017)) provide *high-probability* bounds, they do not guarantee almost-sure containment. This section presents two alternative approaches that allow our framework to accommodate high-probability bounds rather than requiring almost-sure bounds.

Both approaches require an additional regularity condition on the loss function:

**Assumption E.27** (Convex loss). The loss function $\ell(\cdot, y) : \mathbb{R}^k \to [0, 1]$ is convex in its first argument for every $y$.

*Remark* E.28. Assumption E.27 is satisfied by most standard loss functions used in practice, including the squared loss $\ell(\hat{y}, y) = \|\hat{y} - y\|^2$ (after rescaling to $[0, 1]$), the cross-entropy loss (when composed with the softmax, the loss is convex in the logit vector), and the hinge loss. The convexity requirement is needed because the BNN predictor $f_{\text{BNN}}$ is a mixture of $f_{\text{BNN}}^{\text{good}}$ and $f_{\text{BNN}}^{\text{bad}}$, and we need Jensen's inequality to bound the loss of the mixture in terms of the losses of its components.

#### E.10.1. SETUP AND NOTATION

Let $q(A, B \mid \theta)$ denote the variational posterior over the factor matrices. For each layer $i$, suppose we have high-probability bounds:

$$\mathbb{P}_{A_i \sim q_A(\cdot | \theta_i)} \left( \|A_i\|_2 \le C_i^A \right) \ge 1 - \delta_i^A, \tag{17}$$

$$\mathbb{P}_{B_i \sim q_B(\cdot | \theta_i)} \left( \|B_i\|_2 \le C_i^B \right) \ge 1 - \delta_i^B. \tag{18}$$

Define the "good event" $\mathcal{E}_{\text{good}}$ as the intersection of all these events across all $D$ layers:

$$\mathcal{E}_{\text{good}} := \bigcap_{i=1}^{D} \left\{ \|A_i\|_2 \le C_i^A \right\} \cap \left\{ \|B_i\|_2 \le C_i^B \right\}. \tag{19}$$

By a union bound:

$$\mathbb{P}(\mathcal{E}_{\text{good}}) \ge 1 - \sum_{i=1}^{D} (\delta_i^A + \delta_i^B) =: 1 - \delta_{\text{total}}. \tag{20}$$

When $\mathcal{E}_{\text{good}}$ occurs, all factor matrices satisfy their spectral norm bounds, and consequently all induced weight matrices $W_i = A_i B_i^\top$ satisfy $\|W_i\|_2 \le C_i := C_i^A C_i^B$ and $\text{rank}(W_i) \le r_i$ by the same argument as Lemma E.5.

We write $p := \mathbb{P}(\mathcal{E}_{\text{good}}) \ge 1 - \delta_{\text{total}}$ throughout this section.

#### E.10.2. APPROACH A: DECOMPOSITION OF THE BNN PREDICTOR

**Function-level decomposition.** The BNN predictor is defined as an expectation over the full posterior:

$$f_{\text{BNN}}(x; \theta) = \mathbb{E}_{(A,B) \sim q(\cdot | \theta)}[f_{A,B}(x)]. \tag{21}$$

By the law of total expectation *applied to the posterior distribution $q$*, we can decompose this as:

$$f_{\text{BNN}}(x; \theta) = p \cdot f_{\text{BNN}}^{\text{good}}(x; \theta) + (1 - p) \cdot f_{\text{BNN}}^{\text{bad}}(x; \theta), \tag{22}$$

where:

$$f_{\text{BNN}}^{\text{good}}(x; \theta) := \mathbb{E}_{(A,B) \sim q}[f_{A,B}(x) \mid \mathcal{E}_{\text{good}}], \tag{23}$$

$$f_{\text{BNN}}^{\text{bad}}(x; \theta) := \mathbb{E}_{(A,B) \sim q}[f_{A,B}(x) \mid \mathcal{E}_{\text{good}}^c]. \tag{24}$$

*Note:* Both $f_{\text{BNN}}^{\text{good}}$ and $f_{\text{BNN}}^{\text{bad}}$ are deterministic functions of $x$ (given the variational parameters $\theta$). The conditioning in (23)–(24) is over the posterior $(A, B) \sim q$, not over the data distribution. Equation (22) is a pointwise identity for each $x$, expressing $f_{\text{BNN}}(x)$ as a convex combination of two deterministic functions.

**Analysis of the "good" term.**   The conditional expectation $f_{\text{BNN}}^{\text{good}}$ is taken over the conditional distribution $q(\cdot \mid \theta, \mathcal{E}_{\text{good}})$, which is supported on factor matrices satisfying the spectral norm bounds. By the same argument as Lemma E.11 (replacing $q$ by $q(\cdot \mid \mathcal{E}_{\text{good}})$):

$$f_{\text{BNN}}^{\text{good}}(\cdot; \theta) \in \overline{\text{conv}(\mathcal{F}_D^{\text{supp,good}}(\theta))}, \tag{25}$$

where $\mathcal{F}_D^{\text{supp,good}}(\theta)$ consists of network functions realized by factor pairs in the support of $q(\cdot \mid \mathcal{E}_{\text{good}})$. Since all such pairs satisfy the spectral norm constraints by definition of $\mathcal{E}_{\text{good}}$:

$$\mathcal{F}_D^{\text{supp,good}}(\theta) \subseteq \mathcal{F}_D^{\text{Pinto}}(C, r). \tag{26}$$

By the same arguments as Proposition E.13, the Gaussian complexity of the class containing $f_{\text{BNN}}^{\text{good}}$ is bounded by that of Pinto's class. Applying Theorem E.15, we obtain that with probability at least $1 - \delta$ over $S \sim \mathcal{D}^m$:

$$\mathbb{E}_{(x,y)}[\ell(f_{\text{BNN}}^{\text{good}}(x), y)] \leq \frac{1}{m} \sum_{j=1}^{m} \ell(f_{\text{BNN}}^{\text{good}}(x_j), y_j) + \sqrt{\pi}\mathcal{L}\,\hat{\mathcal{G}}_{S_x}(\mathcal{F}_D^{\text{Pinto}}(C, r)) + 3\sqrt{\frac{\log(2/\delta)}{2m}}. \tag{27}$$

**From $f_{\text{BNN}}^{\text{good}}$ to $f_{\text{BNN}}$: applying loss convexity.**   We now bound the population loss of the actual deployed predictor $f_{\text{BNN}}$. By (22), for each $(x, y)$:

$$f_{\text{BNN}}(x) = p \cdot f_{\text{BNN}}^{\text{good}}(x) + (1 - p) \cdot f_{\text{BNN}}^{\text{bad}}(x).$$

Under Assumption E.27, $\ell(\cdot, y)$ is convex in its first argument, so by Jensen's inequality:

$$\begin{aligned}
\ell(f_{\text{BNN}}(x), y) &\leq p \cdot \ell(f_{\text{BNN}}^{\text{good}}(x), y) + (1 - p) \cdot \ell(f_{\text{BNN}}^{\text{bad}}(x), y) \\
&\leq p \cdot \ell(f_{\text{BNN}}^{\text{good}}(x), y) + (1 - p) \cdot 1,
\end{aligned} \tag{28}$$

where in the second inequality we used $\ell \in [0, 1]$.

Taking the expectation over $(x, y) \sim \mathcal{D}$:

$$\mathbb{E}_{(x,y)}[\ell(f_{\text{BNN}}(x), y)] \leq p \cdot \mathbb{E}_{(x,y)}[\ell(f_{\text{BNN}}^{\text{good}}(x), y)] + (1 - p) \leq \mathbb{E}_{(x,y)}[\ell(f_{\text{BNN}}^{\text{good}}(x), y)] + \delta_{\text{total}}, \tag{29}$$

where the last step uses $1 - p \leq \delta_{\text{total}}$ and $p \leq 1$.

**Combined generalization bound.**   Substituting (27) into (29):

**Proposition E.29** (Generalization bound under high-probability spectral norm constraints)**.** *Let $\ell(\cdot, y) \in [0, 1]$ be $\mathcal{L}$-Lipschitz and convex in its first argument (Assumptions in Theorem E.15 and Assumption E.27). Let high-probability bounds (17)–(18) hold with $\delta_{total} = \sum_{i=1}^{D}(\delta_i^A + \delta_i^B)$. Then with probability at least $1 - \delta$ over $S \sim \mathcal{D}^m$:*

$$\begin{aligned}
\mathbb{E}_{(x,y)}[\ell(f_{\text{BNN}}(x), y)] \leq \frac{1}{m} \sum_{j=1}^{m} \ell(f_{\text{BNN}}^{\text{good}}(x_j), y_j) \\
+ \sqrt{\pi}\mathcal{L} \cdot \hat{\mathcal{G}}_{S_x}(\mathcal{F}_D^{\text{Pinto}}(C, r)) + 3\sqrt{\frac{\log(2/\delta)}{2m}} + \delta_{total},
\end{aligned} \tag{30}$$

*where $C_i = C_i^A C_i^B$ and $f_{\text{BNN}}^{\text{good}}$ is defined in (23).*

*Remark* E.30 (Empirical loss of $f_{\text{BNN}}^{\text{good}}$ vs. $f_{\text{BNN}}$). The empirical loss term involves $f_{\text{BNN}}^{\text{good}}$ rather than $f_{\text{BNN}}$, because the generalization bound from Theorem E.15 is applied to $f_{\text{BNN}}^{\text{good}}$ (which belongs to the controlled function class). In practice, $f_{\text{BNN}}^{\text{good}}$ can be computed via Monte Carlo sampling from the truncated posterior $q(\cdot \mid \mathcal{E}_{\text{good}})$, i.e., by rejection sampling: draw $(A, B) \sim q$ and retain only samples satisfying the spectral norm bounds. When $\delta_{\text{total}}$ is small, the rejection rate is low and $f_{\text{BNN}}^{\text{good}} \approx f_{\text{BNN}}$.

*Remark* E.31 (Practical interpretation). The additional term $\delta_{\text{total}}$ represents the "cost" of using high-probability bounds instead of almost-sure bounds. For Gaussian posteriors with Van Handel-type bounds (see Remark E.35), choosing $\delta_i^A = \delta_i^B = \delta_0/(2D)$ for each layer gives $\delta_{\text{total}} = \delta_0$. The spectral norm bounds $C_i^A$ and $C_i^B$ then scale as $O(\sqrt{\log(D/\delta_0)})$, which introduces a mild logarithmic dependence on the depth and confidence level.

*Remark* E.32 (Why convexity is necessary). Without Assumption E.27, the bound (28) does not hold. Using only Lipschitz continuity gives

$$|\ell(f_{\text{BNN}}(x), y) - \ell(f_{\text{BNN}}^{\text{good}}(x), y)| \leq \mathcal{L}\|f_{\text{BNN}}(x) - f_{\text{BNN}}^{\text{good}}(x)\|_2 = \mathcal{L}(1-p)\|f_{\text{BNN}}^{\text{bad}}(x) - f_{\text{BNN}}^{\text{good}}(x)\|_2,$$

which requires bounding $\|f_{\text{BNN}}^{\text{bad}}(x)\|_2$—the expected output under the "bad" event where spectral norm constraints are violated. This quantity can be arbitrarily large for unbounded variational families, making the Lipschitz approach impractical without additional assumptions on the tails of $q$. Loss convexity avoids this issue entirely by exploiting the $[0, 1]$-boundedness of $\ell$.

### E.10.3. APPROACH B: MODIFIED FUNCTION CLASS WITH HIGH-PROBABILITY INCLUSION

An alternative approach modifies the function class to directly incorporate the high-probability nature of the constraints.

**Definition.** Define the *high-probability support class*:

$$\mathcal{F}_D^{\text{hp}}(\theta, \delta) := \left\{ f_W : W_i = A_i B_i^\top, (A_i, B_i) \in \text{supp}(q(\cdot \mid \theta_i)) \cap \mathcal{B}_i, \; \forall i = 1, \ldots, D \right\}, \tag{31}$$

where $\mathcal{B}_i := \{(A_i, B_i) : \|A_i\|_2 \leq C_i^A(\delta), \|B_i\|_2 \leq C_i^B(\delta)\}$ and $C_i^A(\delta), C_i^B(\delta)$ are chosen so that

$$\mathbb{P}((A_i, B_i) \in \mathcal{B}_i) \geq 1 - \delta/(2D). \tag{32}$$

**Properties.**

1. $\mathcal{F}_D^{\text{hp}}(\theta, \delta) \subseteq \mathcal{F}_D^{\text{Pinto}}(C(\delta), r)$ by construction, where $C_i(\delta) = C_i^A(\delta)C_i^B(\delta)$.

2. $f_{\text{BNN}}^{\text{good}}(\cdot; \theta) \in \overline{\text{conv}(\mathcal{F}_D^{\text{hp}}(\theta, \delta))}$ by the same argument as Lemma E.11, applied to the conditional posterior $q(\cdot \mid \mathcal{E}_{\text{good}})$.

3. $\mathbb{P}(\text{sample from } q \text{ produces function in } \mathcal{F}_D^{\text{hp}}) \geq 1 - \delta$ by the union bound.

**Generalization bound.** By the same reasoning as Approach A—decomposing $f_{\text{BNN}}$ as in (22), applying Assumption E.27 to obtain (29), and bounding the population loss of $f_{\text{BNN}}^{\text{good}}$ via Theorem E.15—we obtain:

$$\mathbb{E}_{(x,y)}[\ell(f_{\text{BNN}}(x), y)] \leq \frac{1}{m} \sum_{j=1}^{m} \ell(f_{\text{BNN}}^{\text{good}}(x_j), y_j)$$
$$+ \sqrt{\pi}\mathcal{L} \cdot \hat{\mathcal{G}}_{S_x}(\mathcal{F}_D^{\text{Pinto}}(C(\delta), r)) + 3\sqrt{\frac{\log(2/\delta')}{2m}} + \delta, \tag{33}$$

where the Gaussian complexity bound uses the spectral norm constraints $C_i(\delta) = C_i^A(\delta) \cdot C_i^B(\delta)$ that depend on the confidence parameter $\delta$.

*Remark* E.33 (Comparison of approaches). Both approaches yield the same final bound structure, with an additive $\delta$ term from the high-probability relaxation and a convexity requirement on the loss (Assumption E.27).

- **Approach A** is conceptually cleaner: it decomposes the predictor into good and bad components at the function level.

- **Approach B** is more direct: it modifies the function class definition to explicitly encode the high-probability constraint.

In practice, either approach can be used depending on which is more convenient for the specific application.

## E.11. Practical Enforcement Remarks

We now discuss practical methods for enforcing or approximating Assumption E.4, and provide specific concentration results for Gaussian variational posteriors.

*Remark* E.34 (Enforcement in practice). Assumption E.4 can be satisfied (or approximated) by several practical approaches:

1. **Projection / spectral normalization.** After each parameter update during optimization, rescale factors to enforce spectral norm constraints:

$$A_i \leftarrow \min\left(1, \frac{C_i^A}{\|A_i\|_2}\right) A_i, \qquad B_i \leftarrow \min\left(1, \frac{C_i^B}{\|B_i\|_2}\right) B_i.$$

   In reparameterized variational inference, the same clipping can be applied to sampled factors. This ensures almost-sure satisfaction of the bounds.

2. **Compact-support families.** Use truncated Gaussians or uniform distributions supported on the spectral norm ball. For example, a truncated Gaussian with support $\{A : \|A\|_2 \leq C^A\}$ naturally satisfies the assumption. This requires computing the normalization constant, which may be computationally expensive for high-dimensional matrices.

3. **Soft constraints via penalties.** Add regularization penalties that discourage violations:

$$\mathcal{L}_{\text{reg}}(\theta) = \lambda_A \sum_{i=1}^{D} \max(0, \|A_i\|_2 - C_i^A)^2 + \lambda_B \sum_{i=1}^{D} \max(0, \|B_i\|_2 - C_i^B)^2.$$

   This yields a high-probability rather than almost-sure guarantee unless combined with truncation, but may be easier to implement and optimize.

*Remark* E.35 (Van Handel-type spectral norm bounds for Gaussian factors). If $A_i \in \mathbb{R}^{h_i \times r_i}$ has independent Gaussian entries with variance $\sigma_i^2$ (and possibly nonzero mean $\mu_i$), nonasymptotic concentration results provide sharp bounds on the spectral norm.

Specifically, by Van Handel's matrix concentration inequality (Van Handel, 2017), there exist universal constants $c_1, c_2 > 0$ such that for all $t \geq 0$,

$$\mathbb{P}\left(\|A_i\|_2 \leq \|\mathbb{E}[A_i]\|_2 + c_1\sigma_i(\sqrt{h_i} + \sqrt{r_i}) + c_2\sigma_i t\right) \geq 1 - 2e^{-t^2}.$$

Setting $t = \sqrt{\log(2/\delta_i)}$ yields a $(1 - \delta_i)$ high-probability bound:

$$\mathbb{P}\left(\|A_i\|_2 \leq \|\mathbb{E}[A_i]\|_2 + c_1\sigma_i(\sqrt{h_i} + \sqrt{r_i}) + c_2\sigma_i\sqrt{\log(2/\delta_i)}\right) \geq 1 - \delta_i.$$

**Recovering an almost-sure bound.** To recover an almost-sure bound from this high-probability statement, one may *truncate* the variational distribution outside the spectral norm ball:

$$C_i^A := \|\mathbb{E}[A_i]\|_2 + c_1\sigma_i(\sqrt{h_i} + \sqrt{r_i}) + c_2\sigma_i\sqrt{\log(2/\delta_i)},$$

at the cost of modifying the variational family from a standard Gaussian to a truncated Gaussian.

**Retaining high-probability bounds.** Alternatively, one may retain the high-probability statement and use Approaches A or B from Section E.10 to carry an additional failure probability $\delta_{\text{total}}$ through the final generalization bound. This is often more practical than truncation, as it requires no modification to the variational family. This approach additionally requires loss convexity (Assumption E.27).

**Practical recommendations.**   In practice, we recommend:

- Use spectral normalization (projection) during training to enforce hard constraints, which ensures almost-sure satisfaction at minimal computational cost.

- Use Van Handel bounds to guide the choice of $C_i^A$ and $C_i^B$ based on the variational posterior parameters $\sigma_i$ and $\mu_i$.

- If using unbounded variational families without projection, apply the high-probability analysis from Section E.10 with appropriately chosen $\delta_{\text{total}}$ and ensure the loss function satisfies Assumption E.27.

*Remark* E.36 (Tightness of Van Handel bounds). The Van Handel bound is known to be tight up to universal constants. For a random matrix with independent Gaussian entries, the spectral norm concentrates around $\sigma(\sqrt{h} + \sqrt{r})$ with high probability. This means:

- The constants $C_i^A$ and $C_i^B$ grow as $O(\sigma_i \sqrt{h_i + r_i})$ for Gaussian posteriors.

- The induced spectral norm bounds $C_i = C_i^A C_i^B$ then scale as $O(\sigma_i^2 (h_i + r_i))$.

- These bounds are dimension-dependent but explicit and computable from the variational parameters.

# F. Principled Initialization of Low-Rank Variational Weight Matrices

## F.1. Setting and Notation

Consider a linear layer with input dimension $n$ and output dimension $m$, parameterized by a weight matrix

$$W \in \mathbb{R}^{n \times m}.$$

Rather than parameterizing $W$ directly, we adopt a low-rank factorization

$$W = AB^\top, \quad A \in \mathbb{R}^{n \times r}, \quad B \in \mathbb{R}^{m \times r}, \quad r \ll \min(n, m).$$

In a variational Bayesian formulation, we place factorized Gaussian variational posteriors over the entries of $A$ and $B$:

$$q(A) = \prod_{i,k} \mathcal{N}(A_{ik} \mid \mu_{ik}^A, (\sigma_{ik}^A)^2), \qquad q(B) = \prod_{j,k} \mathcal{N}(B_{jk} \mid \mu_{jk}^B, (\sigma_{jk}^B)^2).$$

The objective of initialization is to choose $(\mu^A, \mu^B, \sigma^A, \sigma^B)$ such that the induced distribution over $W$ yields stable forward signal propagation and well-conditioned optimization at the start of training.

## F.2. Reference Dense Initialization and Variance Propagation

Initialization of deep networks is traditionally analyzed through variance propagation arguments. Let $x \in \mathbb{R}^n$ be an input vector with i.i.d. components satisfying

$$\mathbb{E}[x_i] = 0, \qquad \text{Var}(x_i) = 1.$$

For a dense linear layer $z = Wx$, sufficient conditions for stable signal propagation require that the entries of $W$ satisfy

$$\mathbb{E}[W_{ij}] = 0, \qquad \text{Var}(W_{ij}) = \sigma_W^2,$$

where $\sigma_W^2$ depends on $(n, m)$ and the nonlinearity applied after the linear transformation.

Glorot and Bengio (Glorot & Bengio, 2010) showed that choosing

$$\sigma_W^2 = \frac{2}{n + m}$$

preserves the variance of activations across layers for linear and saturating nonlinearities.

He et al. (He et al., 2015) further derived that for ReLU activations, variance preservation requires

$$\sigma_W^2 = \frac{2}{n}.$$

These results are textbook consequences of second-moment propagation and form the theoretical basis of modern dense initialization schemes.

In what follows, $\sigma_W^2$ is treated as a known constant determined solely by layer dimensions and activation function.

### F.3. Induced Statistics of Low-Rank Weight Matrices

Let the variational means $\mu_{ik}^A$ and $\mu_{jk}^B$ be initialized as independent random variables with

$$\mathbb{E}[\mu_{ik}^A] = \mathbb{E}[\mu_{jk}^B] = 0, \qquad \mathrm{Var}(\mu_{ik}^A) = v_A, \quad \mathrm{Var}(\mu_{jk}^B) = v_B.$$

The mean of the induced weight distribution is

$$\bar{W} := \mathbb{E}_q[W] = \mu^A (\mu^B)^\top,$$

with entries

$$\bar{W}_{ij} = \sum_{k=1}^{r} \mu_{ik}^A \mu_{jk}^B.$$

Since products of independent zero-mean random variables satisfy

$$\mathbb{E}[\mu_{ik}^A \mu_{jk}^B] = 0, \qquad \mathrm{Var}(\mu_{ik}^A \mu_{jk}^B) = v_A v_B,$$

we obtain by independence and additivity of variance:

$$\mathrm{Var}(\bar{W}_{ij}) = \sum_{k=1}^{r} \mathrm{Var}(\mu_{ik}^A \mu_{jk}^B) = r\, v_A v_B.$$

This calculation follows directly from independence and the fact that the variance of a sum of independent random variables equals the sum of their variances.

### F.4. Variance-Matching Condition

To ensure that the low-rank parameterization reproduces the second-order statistics of a properly initialized dense layer, we impose the condition
$$\mathrm{Var}(\bar{W}_{ij}) = \sigma_W^2.$$

Substituting the expression from Section F.3 yields the constraint

$$r\, v_A v_B = \sigma_W^2. \tag{VM}$$

Equation (VM) defines a one-parameter family of valid initializations, parameterized by the choice of how to split the variance between $v_A$ and $v_B$.

### F.5. Symmetric Choice and Identifiability Considerations

The factorization $W = AB^\top$ exhibits scale non-identifiability: for any $c > 0$,

$$A \mapsto cA, \quad B \mapsto c^{-1}B \quad \Rightarrow \quad AB^\top \text{ unchanged.}$$

Such gauge freedoms are well-known in matrix factorizations and latent variable models (see, e.g., (Udell et al., 2015)).

A natural way to fix this non-identifiability at initialization is to impose symmetry between the factors:

$$v_A = v_B.$$

Combined with the variance-matching condition (VM), this yields the unique symmetric solution

$$v_A = v_B = \sqrt{\frac{\sigma_W^2}{r}}.$$

This choice ensures:

- Balanced gradient magnitudes for $A$ and $B$ during early training,

- Isotropy of the variational posterior at initialization,

- Consistency with the symmetric square-root parameterization $W = U\Sigma^{1/2}(\Sigma^{1/2}V^\top)$ arising from the singular value decomposition.

### F.6. Concrete Initializer Families

#### F.6.1. GAUSSIAN INITIALIZERS

If the variational means are initialized as

$$\mu_{ik}^A, \mu_{jk}^B \sim \mathcal{N}(0, s^2),$$

then $v_A = v_B = s^2$, and the symmetric variance-matching condition yields

$$s = \left(\frac{\sigma_W^2}{r}\right)^{1/4}.$$

**Example:** For He initialization with ReLU activations, $\sigma_W^2 = 2/n$, so:

$$s = \left(\frac{2}{nr}\right)^{1/4}.$$

#### F.6.2. UNIFORM INITIALIZERS

If instead

$$\mu_{ik}^A, \mu_{jk}^B \sim \mathcal{U}(-a, a),$$

then by standard probability theory (Casella & Berger, 2002),

$$\mathrm{Var}(\mathcal{U}(-a, a)) = \frac{a^2}{3}.$$

Substituting $v_A = v_B = a^2/3$ into the symmetric variance-matching condition (VM):

$$r\left(\frac{a^2}{3}\right)^2 = \sigma_W^2,$$

which gives

$$a = \sqrt{3}\left(\frac{\sigma_W^2}{r}\right)^{1/4}.$$

**Example:** For Glorot initialization, $\sigma_W^2 = 2/(n + m)$, so:

$$a = \sqrt{3}\left(\frac{2}{r(n + m)}\right)^{1/4}.$$

### F.7. Initialization of Variational Scales

The variational standard deviations $\sigma_{ik}^A$ and $\sigma_{jk}^B$ control posterior uncertainty and the magnitude of the KL divergence term in variational inference (Blundell et al., 2015).

To prevent stochastic sampling noise from dominating the likelihood signal at early stages of optimization, we initialize them as a fixed fraction of the mean scale:

$$\sigma_{ik}^A = \sigma_{jk}^B = \eta \left( \frac{\sigma_W^2}{r} \right)^{1/4}, \quad \eta \in (0, 1).$$

This choice ensures that the signal-to-noise ratio of $W$ at initialization remains $O(1)$, a condition commonly required for stable variational optimization (Graves, 2011).

**Practical recommendation:** A typical choice is $\eta = 0.1$ or $\eta = 0.01$, which provides moderate initial uncertainty without overwhelming the signal from the data.

### F.8. Summary

The proposed initialization strategy:

- Reproduces the second-order statistics of dense variance-preserving initializations (Glorot & Bengio, 2010; He et al., 2015),

- Resolves scale non-identifiability inherent to low-rank factorizations,

- Preserves computational and parametric efficiency,

- Requires no pretrained full-rank weights.

It therefore provides a theoretically principled foundation for initializing low-rank variational neural network layers.

### F.9. SVD Initialization from Pretrained Weights

Let $W^* \in \mathbb{R}^{d_{out} \times d_{in}}$ denote the weight matrix of a converged deterministic model that minimizes the empirical risk. The variational objective for a low-rank Bayesian layer is

$$\mathcal{L}_{VI} = \mathbb{E}_{q(W)}[\ell(W)] + \beta \, \text{KL}(q(W) \, \| \, p(W)), \tag{34}$$

where $\ell(W)$ is the data loss and $\beta$ is the KL weight.

Since $W^*$ minimizes $\ell(W)$, initializing the posterior mean near $W^*$ places the optimization in a region where:

- The expected data loss $\mathbb{E}_{q(W)}[\ell(W)]$ is already small;

- The optimizer can focus on balancing the KL penalty and fine-tuning under the rank constraint.

This warm-start might, theoretically, reduces the risk of converging to poor local minima far from $W^*$.

To initialize a low-rank Bayesian layer from a pretrained deterministic weight matrix $W_{full} \in \mathbb{R}^{d_{out} \times d_{in}}$, we use the truncated singular value decomposition (SVD).

Let $W_{full} = U\Sigma V^\top$ be the SVD. The best rank-$r$ approximation is

$$W_r = U_r \Sigma_r V_r^\top, \tag{35}$$

where $U_r \in \mathbb{R}^{d_{out} \times r}$, $V_r \in \mathbb{R}^{d_{in} \times r}$, and $\Sigma_r = \text{diag}(\sigma_1, \ldots, \sigma_r)$ contains the top $r$ singular values.

We initialize the mean parameters of the low-rank factorization $W = AB^\top$ by splitting the singular values symmetrically:

$$\mu^A = U_r \Sigma_r^{1/2}, \qquad \mu^B = V_r \Sigma_r^{1/2}. \tag{36}$$

This ensures $\mu^A (\mu^B)^\top = W_r$, warm-starting the Bayesian model from the optimal rank-$r$ approximation to the pretrained weights. Variance parameters are initialized using principled variance-preserving schemes.

# G. Experimental Details

This section provides complete implementation details to ensure full reproducibility of all experiments.

## G.1. Hardware and Software

**Compute Infrastructure:**

- GPU: NVIDIA H100 NVL with 95GB memory

- CUDA Driver Version: 580.95.05

- Operating System: Linux

- Python: 3.10.12

**Software Dependencies:**

- TensorFlow: 2.15.0

- TensorFlow Probability: (bundled with TF 2.15.0)

- NumPy: standard version compatible with TF 2.15.0

- Scikit-learn: for preprocessing and metrics

- Pandas: for data handling

- DuckDB: for efficient preprocessing of large MIMIC-III tables

**TensorFlow Configuration:**

- Precision: float32 (mixed precision explicitly disabled)

- GPU memory: Growth mode enabled (prevents allocating all memory at startup)

- Memory allocator: CUDA async allocator (`TF_GPU_ALLOCATOR=cuda_malloc_async`)

- Deterministic operations: Enabled via `TF_DETERMINISTIC_OPS=1` and `TF_CUDNN_DETERMINISTIC=1`

**Random Seeds:** All experiments in the main text run with 4 or 5 different random seeds. For each seed, we set: Python's built-in `random`, NumPy's random generator, and TensorFlow's random operations. All reported metrics in the main text show mean $\pm$ standard deviation across these runs.

## G.2. Evaluation Metrics

We report the following metrics depending on models and tasks:

**Discrimination Performance:**

- **AUROC** (Area Under ROC Curve): Measures ability to rank positive samples higher than negative samples across all decision thresholds. Computed via scikit-learn's `roc_auc_score`. We use AUROC instead of accuracy because accuracy is misleading for imbalanced datasets (e.g., predicting all negatives on MIMIC-III yields 91.6% accuracy but misses all deaths).

- **AUPR-Error**: Area under precision-recall curve for detecting prediction errors. Higher values indicate better error detection via uncertainty.

- **AUPR-Success**: Area under precision-recall curve for detecting correct predictions via low uncertainty.

**Calibration Quality:**

- **NLL** (Negative Log-Likelihood): Measures quality of predicted probabilities. Lower is better. Computed as binary cross-entropy between true labels and predicted probabilities.

- **ECE** (Expected Calibration Error): Measures alignment between predicted confidences and actual accuracy. We bin predictions by confidence and measure the gap between average confidence and accuracy in each bin.

**Out-of-Distribution Detection:**

- **AUROC-OOD**: AUROC for separating in-distribution from OOD samples using mutual information (MI) as the uncertainty score. MI measures epistemic (model) uncertainty and is computed from $N$ forward passes with different weight samples ($N$ is the number of Monte Carlo samples, specific to each experiment).

- **AUPR-OOD**: AUPR for OOD detection using MI.

- **MI Ratio**: Mean MI on OOD data divided by mean MI on in-distribution data. Higher values indicate better OOD uncertainty.

All metrics computed using scikit-learn where applicable.

### G.3. MIMIC-III Mortality Prediction

G.3.1. DATASET AND PREPROCESSING

**Data source.** MIMIC-III v1.4 (Johnson et al., 2016), binary ICU mortality prediction. Dataset sizes: 40,406 train (8.4% mortality), 4,490 test (8.6% mortality), 5,357 OOD newborn admissions (1.1% mortality).

**Features.** 44 continuous features: demographics (age capped at 90, gender), ICU characteristics (length of stay, time to admission), vital signs (heart rate, blood pressure, respiratory rate, temperature, SpO2), and laboratory results (CBC, metabolic panel, liver function, coagulation). Mean and standard deviation computed per ICU stay where applicable.

**Preprocessing.** Following Ruhe et al. (2019): (i) merge vital signs and labs from CHARTEVENTS/LABEVENTS, (ii) outlier removal via 8×IQR rule, (iii) median imputation, (iv) MinMax scaling to [0,1] fit on training set. Class weights: 1.0 (survived), 11.88 (deceased).

G.3.2. MODELS AND TRAINING

**Architecture.** All models use 2-layer MLP: $44 \to 128$ (ReLU) $\to 128$ (ReLU) $\to 1$ (sigmoid).

| Model | Parameters | Initialization |
|---|---|---|
| Deterministic | 22.4K | Glorot uniform |
| Deep Ensemble (5×) | 112.7K | Glorot uniform, different seeds |
| Full-Rank BBB | 44.8K | $\mu \sim \mathcal{U}(-0.2, 0.2)$, $\rho \sim \mathcal{U}(-5, -4)$ |
| Low-Rank ($r = 15$) | 13.6K | He/Glorot with damping |
| LR-SVD ($r = 15$) | 13.6K | SVD of pretrained deterministic |
| Rank-1 Mult. | 23.3K | Glorot + small perturbations |

*Table 7.* Model specifications. Low-rank damping: 0.32 for $r \le 5$, 0.55 for $r > 5$.

**Rank choice** We first do an ablation study to find the optimal combination of ranks across layers. If a deterministic network is available we can look at the singular values decay to gain intuition for the appropriate range or to validate the chosen rank obtained through ablation (See Figures 9 and 10).

**Training.** Adam optimizer ($\alpha = 10^{-3}$, default $\beta$), batch size 64. Bayesian models: 256 epochs with ELBO objective using KL scaling $\lambda = 0.5/N_{\text{batches}} \approx 7.9 \times 10^{-4}$, scale-mixture prior $p(w) = 0.5\mathcal{N}(0, 1) + 0.5\mathcal{N}(0, e^{-6})$ (Blundell et al., 2015), single weight sample per forward pass during training. Deterministic baseline: 32 epochs to prevent overfitting. Deep ensemble: 32 epochs per member with different seeds.

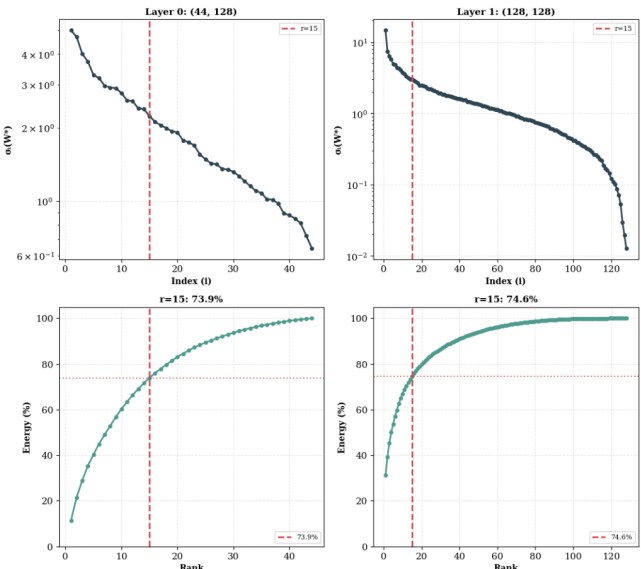

*Figure 9.* Singular value decay and low-rank approximation error for deterministic neural network weight matrices. **Layer 0** ($44 \times 128$, left): Singular values decay rapidly (top-left, log scale). Red dashed line at $r = 15$ indicates selected rank. Bottom-left shows energy retention curve: rank $r = 15$ achieves 73.9% of total Frobenius norm energy. **Layer 1** ($128 \times 128$, right): Similar rapid spectral decay (top-right). Red dashed line marks $r = 15$. Bottom-right: cumulative energy curve reaches 74.6% at $r = 15$. Dotted horizontal line at 75% energy (reference threshold).

**Inference.** Bayesian models: 512 Monte Carlo samples for predictive mean and uncertainty. Deep ensemble: average over 5 members. Epistemic uncertainty via mutual information: $\mathbb{I}(y; \theta | x, \mathcal{D}) = H[y|x, \mathcal{D}] - \mathbb{E}_\theta[H[y|x, \theta]]$.

### G.3.3. EVALUATION METRICS

Standard binary classification metrics computed on predictive mean: AUROC, AUPR. Uncertainty metrics: AUPR-Err (precision-recall for misclassification detection), AUC/AUPR-OOD (detection of OOD samples using mutual information), AUPR-In (correctly identifying in-distribution samples). Calibration: negative log-likelihood (NLL), expected calibration error (ECE). All results averaged over 5 independent runs with different random seeds.

### G.4. Beijing PM2.5 Time-Series Forecasting

#### G.4.1. DATASET AND PREPROCESSING

**Data source.** Beijing PM2.5 air quality dataset, next-hour concentration forecasting. Dataset sizes: 29,213 train, 6,260 validation, 6,260 test. OOD set: Guangzhou PM2.5 data (distributional shift in pollution and meteorology).

**Features.** 15 continuous features per timestep over a $T{=}24$ hour sliding window: PM2.5 concentration, wind direction/speed, temperature, pressure, cumulative snow/rain hours, and derived time encodings. Input shape: $(N, 24, 15)$.

**Preprocessing.** (i) Sliding-window extraction with horizon-1 targets; (ii) StandardScaler (zero mean, unit variance) fit on flattened training data only; (iii) targets independently standardized. All predictions inverse-transformed before evaluation.

#### G.4.2. MODELS AND TRAINING

**Architecture.** All models use a custom 2-layer LSTM ($H{=}64$) built from scratch following Fortunato et al. (2019), with explicit time-step unrolling rather than `tf.keras.layers.LSTM`. Each LSTM layer contains two packed-gate projections (`x_to_gates`: $d_{\text{in}} \to 4H$ with bias; `h_to_gates`: $H \to 4H$ without bias) computing all four gates (input, forget, cell candidate, output) in a single matrix multiply. Forget gate bias initialized to $1.0$ (Jozefowicz et al., 2015). Output: $\mathbf{h}_T$ from the top layer fed to a linear head producing a scalar forecast.

**Bayesian weight caching.** Following Algorithm 2 of Fortunato et al. (2019), variational layers sample weights once at $t{=}0$ (`use_cached=False`) and reuse them for $t{=}1, \ldots, T{-}1$ (`use_cached=True`). A callback clears caches at each batch

| Model | Layer Type | Rank | Initialization |
|---|---|---|---|
| Deterministic | `Dense` | – | Glorot (input), Orthogonal (recurrent) |
| Deep Ensemble (5×) | `Dense` | – | Glorot/Orthogonal, different seeds |
| Full-Rank BBB | `DenseVariational` | – | $\mu \sim \mathcal{U}(-0.2, 0.2)$, $\rho \sim \mathcal{U}(-3, -2)$ |
| Low-Rank ($r=[14, 20]$) | `LowRankDenseVar.` | [14,20] | Adaptive with damping (Sec. F) |
| LR-SVD $r=[14, 20]$) | `LowRankDenseVar.` | [14,20] | SVD of pretrained deterministic |
| Rank-1 Mult. | `Rank1DenseVar.` | 1 | Glorot base + small perturbations |

*Table 8.* LSTM model specifications. Architecture: 2-layer LSTM ($H=64$), input (24, 15), linear output head. Bayesian layers use $W = AB^{\top}$ (low-rank) or $(1+s) \odot xW_0 \odot (1+r)$ (rank-1) (Dusenberry et al., 2020a). Low-rank output head uses rank $= 1$.

boundary. KL loss is added only at $t=0$.

**Training.** Adam optimizer ($\alpha = 10^{-3}$, default $\beta$), batch size 64. Bayesian models: up to 150 epochs with ELBO objective using KL scaling $\lambda = 0.09/N_{\text{train}} \approx 3.08 \times 10^{-6}$, scale-mixture prior $p(w) = 0.5\mathcal{N}(0, 1) + 0.5\mathcal{N}(0, e^{-6})$ (Blundell et al., 2015), single weight sample per forward pass. Early stopping: patience 30 on validation MAE. Learning rate reduced by 0.5 on plateau (patience 5, min $10^{-6}$). Deep ensemble: 5 members, up to 150 epochs each (patience 10), seeds $42, \ldots, 46$.

**Inference.** Bayesian models: 150 Monte Carlo samples for predictive mean and uncertainty. Deep ensemble: average over 5 members; noise variance pooled from validation residuals. Prediction intervals: $\hat{y} \pm z_{0.975}\hat{\sigma}$.

### G.4.3. EVALUATION METRICS

Point prediction: MAE, RMSE, $R^2$ in original scale. Uncertainty: NLL, CRPS, ECE, PICP (target 0.95), MPIW, interval score. OOD detection (Guangzhou): AUROC, AUPR, FPR@95%TPR, TPR@5%FPR using predictive std. Selective prediction at retention rates $\{100\%, 95\%, \ldots, 70\%\}$. All results averaged over runs.

### G.5. SST-2 Sentiment Classification

#### G.5.1. DATASET AND PREPROCESSING

**Data source.** SST-2 for in-distribution (ID) sentiment classification and AGNews for out-of-distribution (OOD) detection. SST-2 is loaded from HuggingFace `glue/sst2`; AGNews is loaded from `ag_news` (test split).

**Splits and sizes.** The current preprocessed artifacts used in this experiment contain:

- SST-2 train: 67,349 examples
- SST-2 dev: 872 examples
- AGNews test (OOD): 7,600 examples (4 classes, 1,900 each)

OOD is capped by `OOD_SLICE = 7600` during evaluation.

**Tokenization.** The pipeline uses `bert-base-uncased` via `AutoTokenizer`. Each text is converted to a string (empty string for missing values), then tokenized with: `add_special_tokens=True`, `max_length=64`, `padding='max_length'`, `truncation=True`, `return_attention_mask=True`, `return_tensors='np'`.

**Saved arrays.** For each split, the following NumPy arrays are written: $*\_ids.npy$, $*\_mask.npy$, $*\_labels.npy$. The pipeline writes to `SAVE_DIR processed_data_agnews/`. All arrays are loaded by the notebook through the config modules.

**Labels.** SST-2 labels are binary. AGNews labels are 4-class (0–3) and are stored but not used for classification; OOD detection treats AGNews as a single out-of-distribution set.

#### G.5.2. MODELS AND TRAINING

**Architecture.** All models share the same tiny pre-norm Transformer encoder: $d_{\text{model}} = 256$, $n_{\text{layers}} = 4$, $n_{\text{heads}} = 4$, $d_{\text{ff}} = 512$, max length 64, CLS pooling, and 2-class softmax. Dropout is disabled (rate 0.0).

**Training.** Batch size 64. Learning rate $2 \times 10^{-4}$ with cosine decay (alpha $= 0.01$) and weight decay 0.01. Baseline uses Adam; Bayesian models use Adam to avoid decay on variational parameters. Epochs: baseline 20, Bayesian 20, deep

| Model | Params | Init (abbr.) |
|---|---|---|
| Det. | 9.92M | Glorot (Dense), Unif (Emb) |
| DE (5×) | 49.61M | Det. init, diff seeds |
| FR-BBB | 19.84M | $\mu \sim \mathcal{U}(-0.2, 0.2)$, $\rho \sim \mathcal{U}(-5, -4)$ |
| LR-BBB ($r{=}16$) | 1.47M | G/H adapt., damp=0.85; $\rho$ rand; $b: \mu{=}0, \rho{=}{-}5$ |
| LR-SVD ($r{=}16$) | 1.47M | SVD init from Det. |
| R1-BBB | 10.01M | $W_0$ Glorot; $b, r, s: \mu \sim \mathcal{U}(-0.2, 0.2)$, $\rho \sim \mathcal{U}(-5, -4)$ |

*Table 9.* SST-2 model specs. Abbr.: Det.=deterministic, DE=deep ensemble, FR=full-rank, LR=low-rank, R1=rank-1, G/H=Glorot/He, Unif=uniform, rand=random.

ensemble 20 per member. Early stopping on validation loss (patience 5) and ReduceLROnPlateau (patience 3, factor 0.5, min LR $10^{-6}$).

**KL scaling and annealing.** KL weight is computed as $\lambda = 0.001/N_{\text{batches}}$. Bayesian models use KL annealing with 2 zero-KL epochs and linear warm-up to full KL by epoch 4.

**Seeds.** Multi-seed robustness uses $\{42, 123, 456, 2026\}$; all RNGs (Python, NumPy, TensorFlow) are set for each run.

### G.5.3. EVALUATION METRICS

**Classification performance.** Accuracy and AUROC on SST-2 dev.

**Calibration.** Negative log-likelihood (NLL) and Brier score. ECE is selected via a best-configuration search over equal-width vs. equal-mass binning with 10/15/20 bins; the best configuration is then applied to all models.

**Uncertainty and OOD detection.** Bayesian models use Monte Carlo sampling with `n_samples=200` (main experiment). Deep ensemble uncertainty is computed from member disagreement (variance/STD). Metrics include AUPR-Error, AUPR-Success, AUROC-OOD, AUPR-In/OOD, and MI ratio (mean MI on OOD divided by mean MI on ID). The evaluation reports both MI-based and STD-based OOD metrics.

### G.5.4. CONTROLLED PROFILING PROTOCOL

**Goal.** To separate end-to-end training time from steady-state optimization cost, we additionally profiled the SST-2 Transformer under a controlled matched benchmark.

**Shared setup.** All methods use the same SST-2 data loader with batch size 64, the same core Transformer configuration ($d_{\text{model}} = 256$, $n_{\text{layers}} = 4$, $n_{\text{heads}} = 4$, $d_{\text{ff}} = 512$, max length 64), dropout rate 0.0, and the same GPU memory-growth initialization. The benchmark is run on an NVIDIA H100 NVL GPU. Each profiling run uses 10 warmup steps followed by 50 measured `train_on_batch` steps, grouped into two measured epochs of 25 steps each.

**Measurement.** We report trainable parameter count, peak GPU memory, and average epoch time under this fixed-step benchmark. Peak memory is measured by resetting TensorFlow memory statistics before profiling and reading `tf.config.experimental.get_memory_info(device_name)['peak']`. The reported numbers come from a single profiling run per method and should therefore be interpreted as controlled benchmark measurements rather than multi-seed averages.

**Method-specific notes.** The deterministic baseline and Deep Ensemble use AdamW, whereas Bayesian models use Adam. Low-Rank BBB and Full-Rank BBB are built with `kl_scale=0.0` and then receive the dataset-derived KL weight through the same scaling utility used in the main training code. Deep Ensemble is profiled sequentially across five independently seeded deterministic members, so its reported epoch time is the total method-level cost for a full 5-member ensemble epoch rather than a per-member epoch time.

**Rank-sweep protocol.** We also profiled a recovered low-rank rank sweep on SST-2 across the settings $(r_{\text{emb}}, r) \in \{(4, 8), (8, 12), (10, 16), (12, 24), (16, 32), (24, 48), (32, 64)\}$. For each recovered setting, we use the same batch size, warmup, and measured-step budget as above, and additionally record single-pass inference time and Monte Carlo inference time at MC20 and MC100 on the 872-example SST-2 dev set.

# H. Extended Results and analysis from Main Experiments

## H.1. Experimental Insights and Practical Considerations

**Calibration–OOD Detection Tradeoff.**   Our results reveal a more nuanced tradeoff than a single uniform pattern. On MIMIC-III, low-rank improves OOD detection despite weaker NLL than Deep Ensembles (0.433 vs. 0.300). On SST-2, Deep Ensembles remain stronger on both NLL (0.434 vs. 0.527) and AUROC-OOD (0.657 vs. 0.640), while low-rank remains competitive at much lower parameter and training cost. On Beijing, the main low-rank advantage is better calibration, coverage, and selective prediction rather than better OOD AUROC. We hypothesize that this task-dependent pattern is consistent with the singular posterior geometry established in Section 3.1. Deep Ensembles maintain $M$ independent point estimates, each highly optimized for likelihood on training data, yielding sharp in-distribution predictions. In contrast, our low-rank posterior concentrates on the rank-$r$ manifold $\mathcal{R}_r$, enforcing structured weight correlations (Lemma 3.5) that can propagate uncertainty more coherently across the input space. Depending on the task, this broader epistemic uncertainty can improve abstention, coverage, or OOD awareness, even when it does not uniformly yield the strongest raw likelihood or OOD metrics.

This suggests a tradeoff between *predictive sharpness* (rewarded by NLL) and the usefulness of uncertainty for downstream reliability decisions. Notably, Low-Rank Gaussian achieves best-in-class AUPR-In (0.824 on MIMIC-III), suggesting its uncertainty estimates remain reliable for identifying trustworthy predictions. For safety-critical applications where abstention, coverage, or distribution-shift awareness matters more than marginal likelihood gains, this tradeoff can favor low-rank approaches.

**Ensembling as a potential mitigation.** In a supplementary SST-2 study (Appendix H.5), a 5-member low-rank Bayesian ensemble improved the single-model low-rank result from 0.638 to 0.731 on AUROC-OOD-MI, from 0.166 to 0.054 on ECE, and from 0.523 to 0.415 on NLL. Although this result is currently based on a single ensemble run, it provides concrete evidence that low-rank ensembling can recover much of the calibration benefit usually associated with heavier ensemble methods while preserving a favorable efficiency profile.

**SVD Initialization.**   Theoretically, warm-starting from the truncated SVD of a learned deterministic weight matrix should improve optimization by initializing near a good solution (Appendix F.9). Empirically, LR-SVD does achieve gains on specific metrics, higher AUROC on MIMIC-III (0.898 vs 0.895) and best AUPR-Succ on SST-2 (0.923 vs 0.917), but these improvements are modest, and performance on other metrics shows no benefit or slight degradation. Thus, SVD initialization is justified only when a pretrained deterministic weight matrix already exists or when training one is inexpensive. Otherwise, the improvements do not warrant the additional computational burden of first training a deterministic model.

**Role of KL Scaling and Annealing.**   The KL weight $\beta$ in the ELBO objective $\mathcal{L} = \mathbb{E}_q[\log p(\mathcal{D}|W)] - \beta \cdot \mathrm{KL}(q\|p)$ controls the strength of posterior regularization toward the prior, and its setting significantly impacts all evaluation metrics.

**Effect on predictive performance and calibration.** Higher $\beta$ values (stronger regularization) constrain the posterior to remain closer to the prior, which may limit in-distribution accuracy but can yield better-calibrated uncertainty estimates. Lower $\beta$ values allow the posterior to be dominated by the likelihood, typically concentrating around likelihood-maximizing weights, improving accuracy but risking overconfidence.

**Effect on OOD detection.** Stronger KL regularization (higher $\beta$) generally *improves* OOD detection by preventing posterior collapse to near-deterministic weights. When the posterior retains meaningful variance, epistemic uncertainty can distinguish familiar from unfamiliar inputs. Conversely, very low $\beta$ can yield posteriors so concentrated that mutual information approaches zero, eliminating the model's ability to express epistemic uncertainty.

We employ KL annealing, gradually increasing $\beta$ from zero over early epochs, to balance these considerations. This allows the model to first find a good likelihood basin before the regularization penalty takes effect, avoiding the underfitting that can occur when strong KL penalties are applied from initialization. Across experiments, we found this schedule crucial for stable training, particularly in deeper architectures (Transformers, LSTMs) where early KL dominance can prevent learning altogether. Our $\beta$ values (ranging from $0.001/N_{\text{batches}}$ to $0.5/N_{\text{batches}}$ depending on architecture) were selected via validation performance, balancing the accuracy–uncertainty tradeoff for each task. As with any regularization hyperparameter, $\beta$ requires tuning for each architecture and dataset; we do not claim our chosen values are optimal.

**Practical Rank Selection.** The rank $r$ can be treated as a standard hyperparameter tuned via ablation. We recommend running ablation studies with reduced computational budget, fewer training epochs and fewer MC samples during validation, to efficiently identify a suitable rank range. In our experiments, this direct approach was our primary method. The singular value decay analysis (Figures 7, 9, 15) serves as an optional validation step that can narrow the search range when a pretrained deterministic model happens to be available, but it is not required. Practitioners without the time or compute to first train a deterministic network can proceed directly to rank ablation. As Figure 10 illustrates, rank selection meaningfully impacts the calibration–OOD detection tradeoff, making this ablation step important for achieving desired performance characteristics.

**Statistical Considerations.** While mean improvements for low-rank models are consistent in direction across random seeds, we note that some confidence intervals overlap between methods (e.g., Table 1 AUPR-OOD). The observed patterns, particularly the calibration–OOD tradeoff, replicate across all four experimental settings, lending confidence to the qualitative conclusions. However, larger-scale studies with more seeds would be needed to establish statistical significance for smaller effect sizes.

### H.2. MIMIC-III

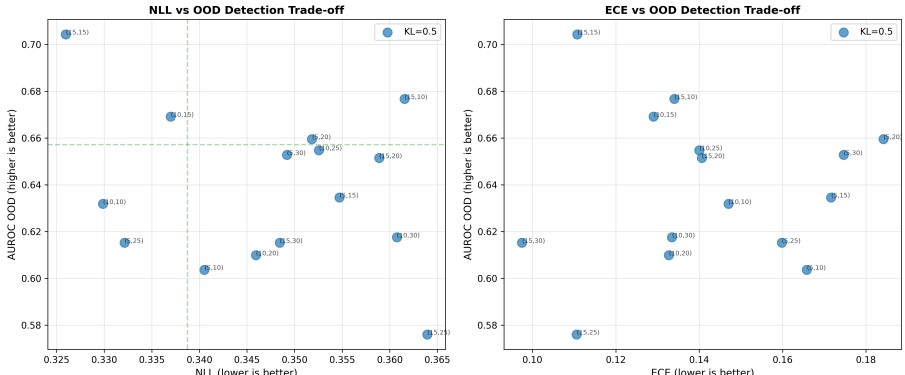

*Figure 10.* Trade-off between OOD detection (AUROC OOD MI) and calibration (NLL, ECE) for low-rank Gaussian models with varying rank pairs $(r_1, r_2)$.

**Rank Selection via Ablation Study** Rank selection is performed through ablation studies with reduced computational budget (fewer epochs, fewer MC samples). Figure 10 shows the trade-off between OOD detection (AUROC, y-axis) and calibration (NLL/ECE, x-axis) for different rank configurations on MIMIC-III. Higher ranks generally improve OOD detection but degrade calibration, while lower ranks maintain better calibration at reduced OOD sensitivity. The configuration $(r_1 = 15, r_2 = 15)$ occupies the Pareto frontier, achieving AUROC $\approx 0.72$ while maintaining NLL $\approx 0.33$ and ECE $\approx 0.12$ (vertical dashed lines mark baseline thresholds). When pretrained weights are available, singular value decay analysis (Figure 9) can optionally guide rank selection by revealing the effective dimensionality of weight matrices. The characteristic exponential decay pattern shows that weight matrices naturally reside in lower-dimensional subspaces, providing post-hoc justification for the empirically selected ranks.

*Table 10.* Training time comparison on MIMIC-III (2-layer MLP). Despite 70% parameter reduction, low-rank methods incur overhead from dual sampling and matrix multiplications.

| Method | Training Time (s) |
|---|---|
| Deterministic Dense | 52.5 |
| Deep Ensemble (5 models) | 221.1 |
| Full-Rank BBB | 502.4 |
| Rank-1 Multiplicative | 503.2 |
| Low-Rank Gaussian SVD init | 524.0 |
| Low-Rank Gaussian ($r = 15$) | 532.2 |

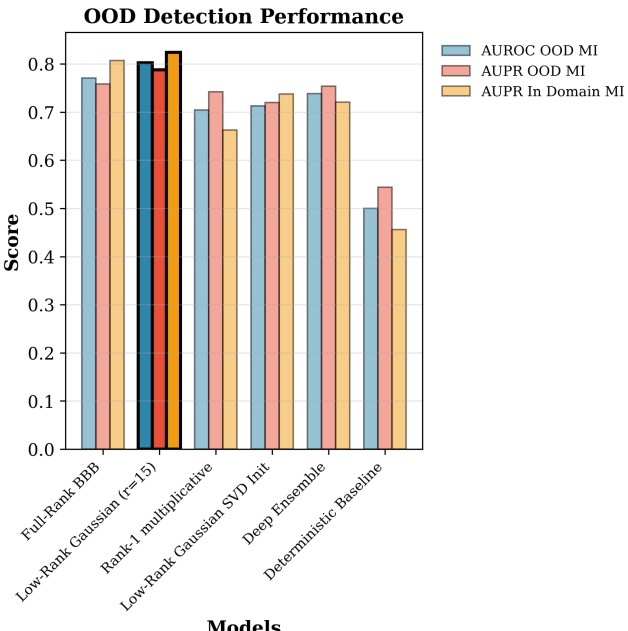

*Figure 11.* OOD detection performance on MIMIC-III. Low-Rank Gaussian ($r = 15$) achieves AUROC OOD of 0.802, outperforming Full-Rank BNN (0.770) and Deep Ensemble (0.738) while using $\sim 30\%$ of full rank parameters and 12% of deep ensemble parameters

**Computational Efficiency at Small Scale** Table 10 shows training times for MIMIC-III experiments. Despite achieving 70% parameter reduction (from 44.8K to 13.6K parameters), low-rank methods exhibit training times comparable to Full-Rank BBB (532s vs. 502s). This stems from the computational overhead of performing two sampling operations and two matrix multiplications ($\mathbf{A} \in \mathbb{R}^{m \times r}$ times $\mathbf{B}^{\top} \in \mathbb{R}^{r \times n}$) rather than a single operation. Modern GPUs achieve higher throughput with one large matrix multiplication dispatched to a single kernel, whereas the factored form requires two kernel launches with intermediate memory transfers. Consequently, at small to medium scales, parameter reduction translates primarily to memory savings rather than wall-clock speedups. However, the reduced memory footprint enables deployment on resource-constrained devices and facilitates cheaper ensembling strategies. As we demonstrate in Section G.5, efficiency benefits emerge at larger scales where the parameter reduction outweighs the factorization overhead.

**Parameter-Efficiency Adjusted Performance** Raw performance comparisons can mislead when models have different parameter scales. Figure 12 applies efficiency adjustment using $\sqrt{\text{min\_params}/\text{params}}$, penalizing larger models while preventing complete suppression via square-root dampening . Low-Rank Gaussian ($r = 15$) emerges as the winner, achieving maximal coverage with particular strength in AUPR In Domain MI and AUPR OOD MI . This suggests low-rank parameterization captures essential uncertainty structure without requiring full parameter space .

Full-Rank BBB suffers notable reduction despite good raw performance due to $\sim 3\times$ parameter count . Deep Ensemble shows the biggest drop(efficiency factors $\approx 0.35$). The Rank-1 Multiplicative model demonstrates moderate efficiency-adjusted performance, balancing extreme compactness with reasonable metrics. This analysis supports low-rank architectures for resource-constrained deployment requiring both performance and efficiency .

### H.3. Beijing PM2.5 time-series forecasting.

**Rank Selection via Ablation Study** Rank selection is performed through ablation studies with reduced computational budget (fewer epochs, fewer MC samples). Figures 13 and 14 show the multi-dimensional trade-offs between OOD detection (AUROC, y-axis) and both calibration (NLL/ECE) and predictive performance (MAE/RMSE) for different rank configurations $(r_{\text{ih}}, r_{\text{hh}})$ on the Beijing dataset. Higher ranks generally improve OOD detection but can degrade both calibration and forecasting accuracy. The configuration $(r_{\text{ih}} = 14, r_{\text{hh}} = 20)$ achieves strong OOD detection (AUROC $\approx 0.66$) while maintaining competitive calibration and predictive performance (vertical dashed lines mark baseline thresholds). The KL weight $\beta = 0.25$ provides better separation of the Pareto frontier compared to lower values. When

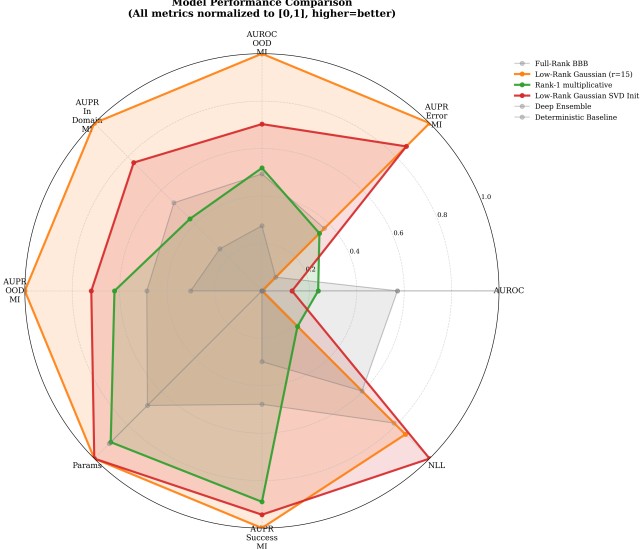

*Figure 12.* Parameter-efficiency adjusted performance across models. Metrics normalized to $[0, 1]$ and scaled by $\sqrt{\text{min\_params}/\text{params}}$ to reward computational efficiency. Low-Rank Gaussian ($r = 15$) achieves the best efficiency-performance trade-off, particularly excelling in AUPR metrics while maintaining compact parameterization.

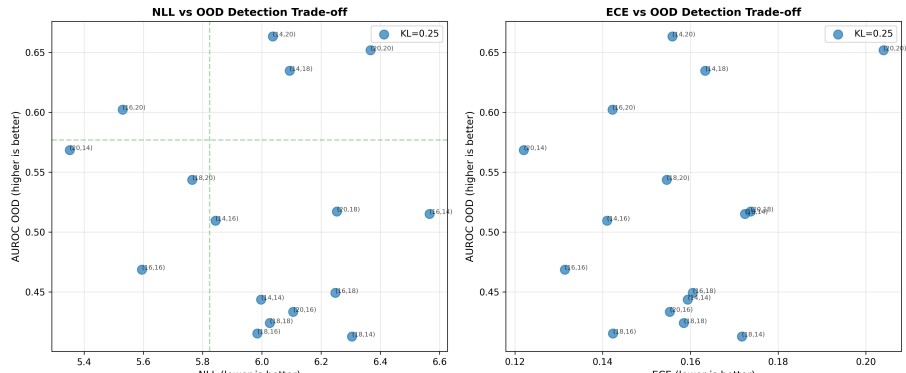

*Figure 13.* Trade-off between OOD detection (AUROC, y-axis) and calibration metrics (NLL and ECE, x-axis) for low-rank LSTM models with varying rank pairs $(r_{\text{ih}}, r_{\text{hh}})$ on Beijing air quality forecasting. Points labeled with rank configurations; color indicates KL weight. Dashed lines mark baseline thresholds.

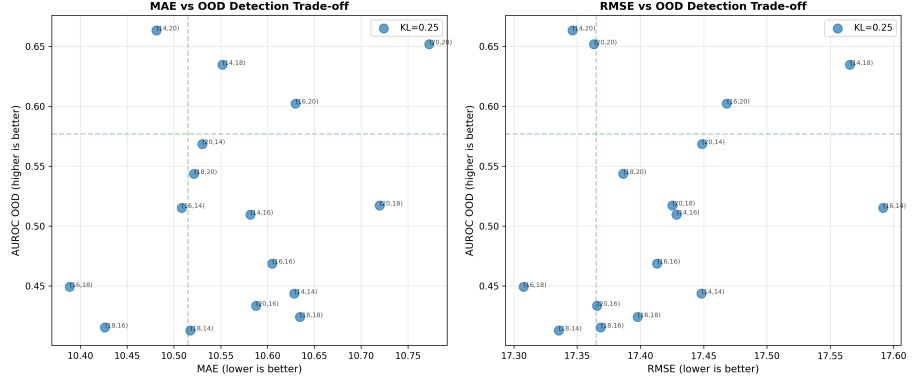

*Figure 14.* Trade-off between OOD detection (AUROC, y-axis) and predictive performance (MAE and RMSE, x-axis) for low-rank LSTM models with varying rank configurations on Beijing air quality forecasting. Lower MAE/RMSE indicates better forecasting accuracy.

pretrained weights are available, singular value decay analysis (Figure 15) can optionally guide rank selection by revealing the effective dimensionality of weight matrices, providing post-hoc justification for the empirically selected ranks.

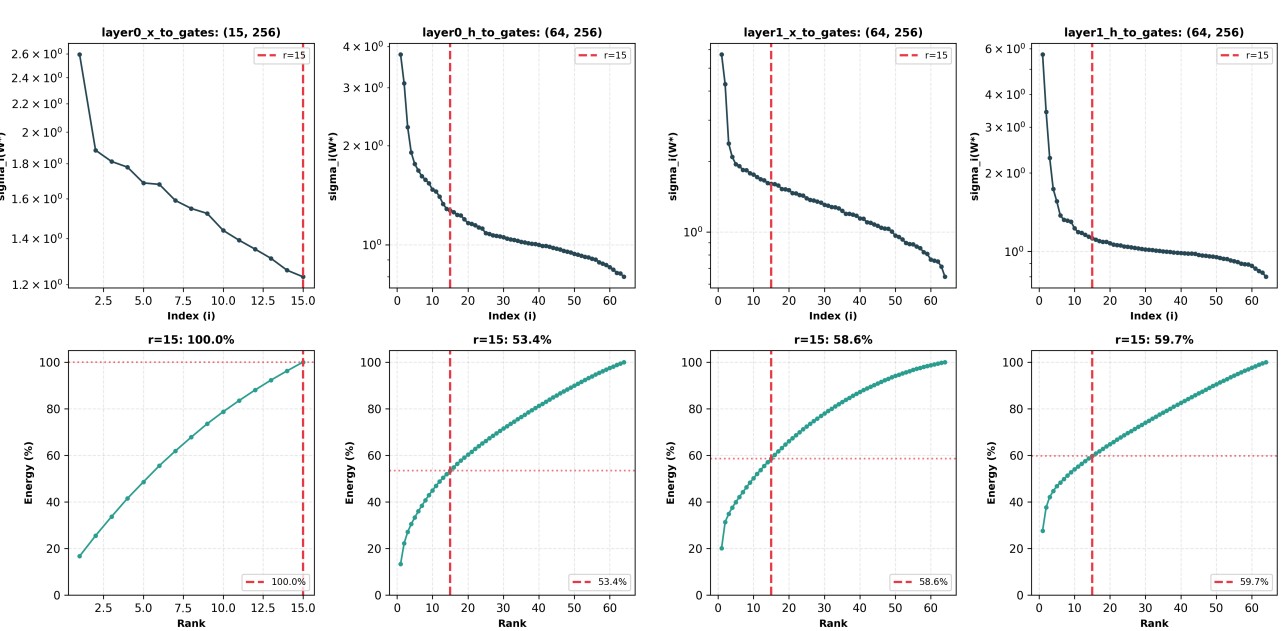

*Figure 15.* Singular value decay of pretrained LSTM weight matrices. we can observe fast decay for both x-to-gate and h-to-gate layers

**Calibration Analysis**    Figure 16 presents the calibration curves for all uncertainty-aware models, with corresponding AUC values in Table 11. By the calibration-curve AUC metric, the Low-Rank Bayesian model achieves the best interval calibration (AUC = 0.104), closely followed by Full-Rank BBB (AUC = 0.110). This differs slightly from the ECE ranking in Table 2, where Full-Rank BBB is best and Low-Rank is second, but both metrics indicate that these two Bayesian models are substantially better calibrated than Deep Ensemble and Rank-1.

The SVD-initialized variant shows slightly degraded calibration (AUC = 0.126), suggesting a trade-off between OOD detection capability and calibration quality. Both Rank-1 and Deep Ensemble exhibit substantially worse calibration (AUC $\approx 0.30$), with curves deviating significantly from the diagonal, indicating overconfident predictions where observed coverage falls below expected coverage across most confidence levels.

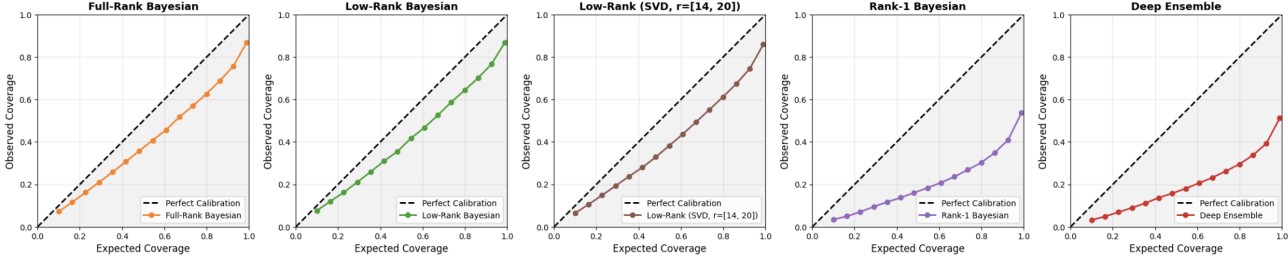

*Figure 16.* Calibration curves for Bayesian LSTM variants. The dashed diagonal represents perfect calibration. Models closer to the diagonal exhibit better-calibrated uncertainty estimates.

**Selective Prediction Analysis**    Selective prediction, where models abstain from predictions on uncertain inputs, is critical for safety-critical applications. We evaluate how well each model's uncertainty estimates enable effective selective prediction by progressively filtering samples with highest predictive variance. Table 12 presents performance at three retention levels. At 100% retention, Deep Ensemble achieves the best MAE (10.45), reflecting its strongest full-coverage point prediction

*Table 11.* Calibration Error (AUC) comparison. Best **bold**, second underlined.

| Model | Calib. AUC↓ |
|---|---|
| Full-Rank BBB | 0.110 |
| Low-Rank ($r$=14,20) | **0.104** |
| Low-Rank (SVD) | 0.126 |
| Rank-1 Mult. | 0.302 |
| Deep Ensemble (5) | 0.307 |

performance. However, the key insight emerges when filtering uncertain predictions: Bayesian methods demonstrate substantially larger improvements, indicating their uncertainty estimates better identify difficult samples.

At 80% retention, Low-Rank achieves 8.71 MAE (17.4% improvement from baseline), outperforming Deep Ensemble's 9.21 MAE (11.4% improvement). Notably, Low-Rank's absolute performance at 80% retention (8.71) surpasses Deep Ensemble's baseline performance (10.45), demonstrating that Bayesian uncertainty can compensate for modest calibration gaps through effective abstention.

The pattern is consistent across retention levels: Bayesian methods show 10-11% improvement at 90% retention compared to Deep Ensemble's 9%, and maintain this advantage at 80%. This suggests the rank constraint induces uncertainty estimates that correlate more strongly with prediction difficulty, even when those uncertainties are less well-calibrated in absolute terms.

*Table 12.* Selective prediction performance on Beijing Air Quality test set. Values show MAE (lower better) and improvement percentage from 100% retention baseline. Best performance at each retention level in **bold**.

| Model | 100% Retention | 90% Retention | 80% Retention |
|---|---|---|---|
| Full-Rank BBB | 10.55 (0.0%) | 9.46 (11.2%) | 8.82 (17.2%) |
| Low-Rank BBB | 10.63 (0.0%) | **9.43** (10.6%) | **8.71** (17.4%) |
| Low-Rank SVD | 10.70 (0.0%) | 9.50 (11.8%) | 9.06 (15.8%) |
| Rank-1 BBB | 10.80 (0.0%) | 9.45 (12.5%) | 9.42 (12.8%) |
| Deep Ensemble | **10.45** (0.0%) | 9.45 (9.1%) | 9.21 (11.4%) |

**Implications for Practice.** These results have important implications for deployment scenarios:

- In full-coverage prediction settings where abstention is not allowed, Deep Ensemble remains preferable because it achieves the best point prediction at 100% retention, despite weaker calibration than the Low-Rank and Full-Rank Bayesian models.

- In safety-critical settings where abstention is acceptable, Low-Rank methods provide better risk-return tradeoffs, achieving lower error rates on retained predictions.

- The crossover point occurs around 85-90% retention, suggesting practitioners can use retention rate requirements to guide method selection.

This selective prediction advantage helps explain why Low-Rank methods achieve competitive OOD detection despite calibration gaps: the uncertainty signal effectively identifies distributional shift, even if absolute confidence levels are miscalibrated.

## H.4. SST-2 sentiment classification

### H.4.1. CONTROLLED PROFILING BENCHMARK

Table 13 reports the controlled matched profiling benchmark on SST-2. These results support the main-text efficiency claim with a more tightly controlled measurement than end-to-end `model.fit` time. Low-Rank BBB uses 1.47M trainable parameters, compared to 19.84M for Full-Rank BBB and 49.61M for a 5-member Deep Ensemble. It also reduces peak GPU memory from 721.1MB to 357.5MB relative to Full-Rank BBB and from 670.1MB to 357.5MB relative to Deep Ensemble.

*Table 13.* Controlled matched profiling benchmark on SST-2. Peak memory and epoch time are measured under a fixed-step `train_on_batch` profiling loop. Deep Ensemble epoch time is the total method-level cost for one full 5-member ensemble epoch.

| Model | Params | Peak Mem. | Epoch Time |
|---|---|---|---|
| Baseline Transformer | 9.92M | 505.4MB | 4.18s |
| Low-Rank BBB | 1.47M | 357.5MB | 5.88s |
| Full-Rank BBB | 19.84M | 721.1MB | 6.45s |
| Deep Ensemble (5) | 49.61M | 670.1MB | 18.99s |

*Table 14.* Recovered low-rank rank-sweep profiling results on SST-2. All measurements use the same fixed-step profiling protocol described in Appendix G.5.4.

| $r_{emb}$ | $r$ | Params | Peak Mem. | Epoch Time | Single Pass | MC20 / MC100 |
|---|---|---|---|---|---|---|
| 4 | 8 | 0.50M | 337.8MB | 5.06s | 0.214s | 6.18s / 30.65s |
| 8 | 12 | 0.86M | 344.7MB | 5.41s | 0.221s | 6.26s / 31.39s |
| 10 | 16 | 1.10M | 359.8MB | 5.88s | 0.228s | 6.51s / 31.75s |
| 12 | 24 | 1.46M | 365.0MB | 5.70s | 0.225s | 6.36s / 32.18s |
| 16 | 32 | 1.94M | 380.2MB | 6.18s | 0.229s | 6.49s / 32.95s |
| 24 | 48 | 2.90M | 421.9MB | 6.29s | 0.232s | 6.55s / 32.65s |
| 32 | 64 | 3.86M | 456.4MB | 5.95s | 0.239s | 6.64s / 33.52s |

The corresponding epoch-time comparison is also favorable. Under the fixed-step benchmark, Low-Rank BBB requires 5.88s per epoch, compared to 6.45s for Full-Rank BBB and 18.99s for Deep Ensemble. Thus, relative to Full-Rank BBB, the low-rank model combines a large reduction in parameter count and peak memory with a small but still positive epoch-time advantage. Relative to Deep Ensemble, the advantage is stronger on all three axes: parameter count, memory, and epoch time. The deterministic baseline remains fastest overall at 4.18s per epoch, but Low-Rank BBB approaches this cost while still maintaining a full Bayesian weight-space treatment.

### H.4.2. RANK-SWEEP PROFILING

Table 14 summarizes the recovered low-rank rank-sweep profiling results. Across the recovered range, parameter count increases from 0.50M to 3.86M ($7.7\times$), but the corresponding increases in practical cost are much smaller: epoch time rises from 5.06s to 5.95s ($1.18\times$), peak memory from 337.8MB to 456.4MB ($1.35\times$), and single-pass inference time from 0.214s to 0.239s ($1.12\times$). Monte Carlo inference changes only mildly, from 6.18s to 6.64s at MC20 and from 30.65s to 33.52s at MC100.

These results clarify the computational meaning of low-rank scaling in practice. Lower rank does reduce runtime and memory, but the effect is clearly sublinear in the parameter reduction because fixed Transformer overheads, kernel-launch effects, and stochastic sampling costs remain significant. The main practical benefit of lower rank is therefore best understood as strong parameter and memory savings, with additional runtime improvements that are favorable but more moderate.

One caveat is that the recovered structured sweep does not include a separate $(r_{emb}, r) = (16, 16)$ row. We therefore use the sweep to characterize the general scaling trend across low-rank settings, rather than to claim an exact controlled profile of the canonical configuration used in the main experiments.

**Computation efficiency** The controlled profiling results above complement the end-to-end training-time measurements by isolating steady-state computational cost under matched conditions.

Table 15 reports training times for each model on SST-2. Low-Rank BBB variants train in approximately 8 minutes, approaching the 7.7-minute baseline of the deterministic Transformer despite performing variational inference with Monte Carlo sampling. This represents a $2.9\times$ speedup over Full-Rank BBB (23.08 min) and $8\times$ over Deep Ensemble (64.72 min).

The efficiency gain stems directly from parameter reduction: Low-Rank BBB uses 1.5M parameters versus 19.8M for Full-Rank BBB and 49.6M for Deep Ensemble. At this scale, the $O(r(m + n))$ versus $O(mn)$ complexity difference translates into meaningful wall-clock improvements, validating our theoretical analysis. Rank-1 BBB occupies a middle ground (12.52 min), reflecting its intermediate parameterization.

These results demonstrate that low-rank factorization enables practical Bayesian uncertainty quantification at training

costs comparable to deterministic baselines, a key requirement for adoption in resource-constrained settings. The matched profiling benchmark and recovered rank sweep above provide a controlled view of the same trend from the perspective of fixed-step training cost and memory usage.

*Table 15.* Average Training Time per Model (in minutes)

| Model | Avg. Time (min) |
| --- | --- |
| Baseline Transformer | 7.70 |
| Low-Rank SVD Init | 7.95 |
| Low-Rank Random Init | 8.22 |
| Rank-1 BBB | 12.52 |
| Full-Rank BBB | 23.08 |
| Deep Ensemble | 64.72 |

**Parameter-Efficiency Adjusted Performance**  Figure 17 presents metrics adjusted by parameter efficiency.

Low-Rank BBB (red) achieves the largest coverage area, dominating across AUROC-OOD, AUPR-Success, AUPR-In, and MI Ratio while maintaining competitive accuracy and NLL. Low-Rank SVD (purple) shows a similar profile with slightly reduced OOD metrics. Both low-rank variants substantially outperform all other approaches on efficiency-adjusted scores.

Deep Ensemble (green) and Full-Rank BBB (orange) collapse toward the center despite strong raw performance, reflecting their $33\times$ and $13\times$ parameter overhead respectively. Rank-1 BBB (brown) shows moderate efficiency-adjusted performance but remains dominated by the low-rank variants across most metrics. The deterministic baseline (blue) scores well only on parameter efficiency, with no uncertainty quantification capability.

This analysis confirms that low-rank factorization achieves the best performance-to-parameter ratio: competitive or superior raw metrics combined with $15$–$33\times$ parameter reduction yields dominant efficiency-adjusted scores across all uncertainty-related metrics.

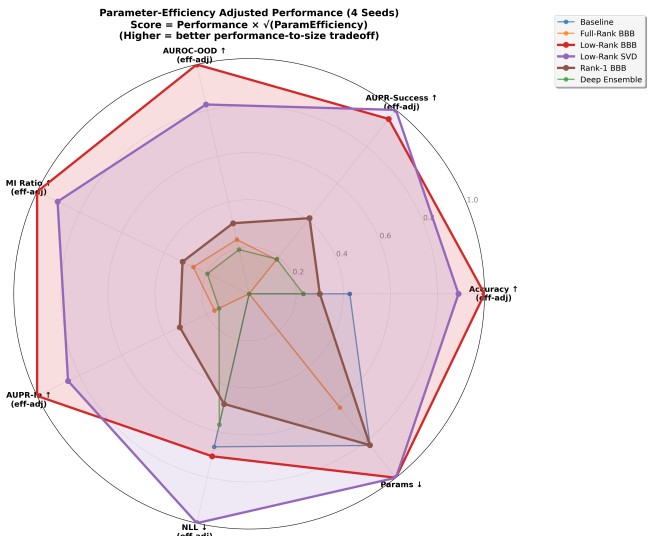

*Figure 17.* Parameter-efficiency adjusted performance across models. Metrics normalized to $[0, 1]$ and scaled by $\sqrt{\text{param\_efficiency}}$. Low-Rank BBB (red) achieves the best overall tradeoff, dominating OOD detection and uncertainty metrics while maintaining competitive accuracy.

**Uncertainty Distribution Analysis**  Figures 18 and 19 visualize the distribution of predictive uncertainty for in-distribution (SST-2) versus out-of-distribution (AGNews) samples across models. Several patterns emerge.

**Low-Rank BBB** exhibits clear separation between in-domain and OOD distributions for both predictive standard deviation and mutual information. The OOD samples (orange) concentrate at higher uncertainty values, enabling effective detection.

**Deep Ensemble** shows the strongest separation overall, consistent with its best OOD-detection metrics in Table 16. Some overlap remains, but the ID and OOD distributions are more clearly separated than for Full-Rank BBB and Rank-1 BBB.

**Full-Rank BBB** displays considerable overlap between distributions, explaining its weaker OOD detection performance. The model struggles to assign higher uncertainty to unfamiliar inputs.

**Rank-1 BBB** exhibits near-degenerate uncertainty distributions with values concentrated near zero, consistent with its low MI ratio (Table 3). This confirms that rank-1 perturbations provide insufficient epistemic expressiveness.

**Low-Rank SVD BBB** shows similar separation patterns to Low-Rank BBB with random initialization, though with slightly narrower distributions overall.

Notably, predictive standard deviation provides marginally better OOD separation than mutual information for most Bayesian models. Table 16 confirms this empirically: STD-based metrics slightly outperform MI-based metrics across all models, though differences are small. This suggests that for these architectures, total predictive variance captures OOD signal as effectively as the epistemic component isolated by mutual information.

*Table 16.* OOD detection metrics using Mutual Information (MI) vs Predictive Standard Deviation (STD) as uncertainty measures. Results averaged over 4 seeds. Higher is better for all metrics.

| Model | AUROC-OOD | | AUPR-OOD | | AUPR-In | |
|---|---|---|---|---|---|---|
| | MI | STD | MI | STD | MI | STD |
| Deterministic | .500 | .500 | .897 | .897 | .103 | .103 |
| Deep Ensemble | .658 | **.663** | .930 | **.932** | .267 | **.282** |
| Full-Rank BBB | .622 | .623 | .921 | .921 | .222 | .223 |
| Low-Rank BBB | .641 | **.642** | .919 | **.920** | .302 | .303 |
| LR-SVD BBB | .616 | **.619** | .903 | **.904** | .273 | **.274** |
| Rank-1 BBB | .613 | .615 | .904 | .905 | .273 | .273 |

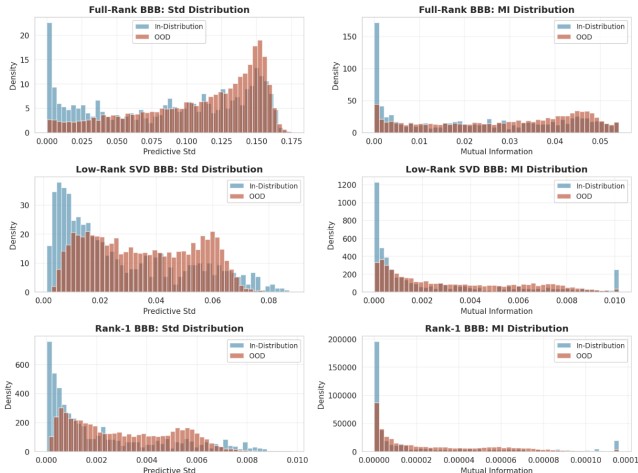

*Figure 18.* Predictive standard deviation distributions for in-distribution (blue) vs OOD (orange) samples. Low-Rank BBB and Deep Ensemble show clear separation; Rank-1 BBB shows near-degenerate distributions.

### H.5. Supplementary Low-Rank Ensembling Study on SST-2

To test whether ensembling mitigates the calibration gap observed for a single low-rank posterior, we trained a 5-member low-rank Bayesian ensemble on SST-2. Each member used the same low-rank BBB Transformer architecture and training recipe as the main paper. At inference time, we aggregated 20 stochastic posterior draws per member, for a total of 100 predictive draws.

Table 17 compares this ensemble against the single-model low-rank row and the existing Deep Ensemble and SWAG references. Because the low-rank ensemble result currently comes from a single run, whereas some comparison rows are

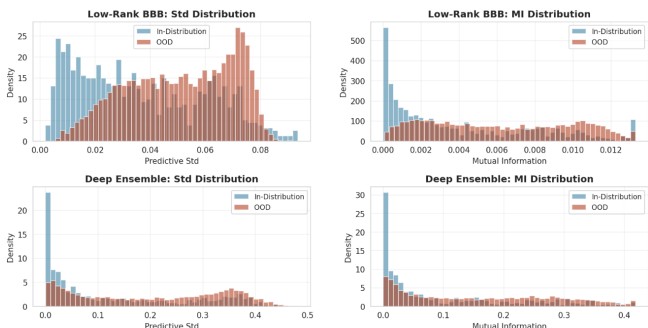

*Figure 19.* Mutual information distributions for in-distribution (blue) vs OOD (orange) samples. Similar patterns to STD, with Deep Ensemble showing strongest separation.

*Table 17.* Supplementary SST-2 low-rank ensembling study. The low-rank Bayesian ensemble uses 5 independently trained low-rank BBB members and 20 posterior draws per member (100 total predictive draws).

| Model | Acc | AUROC-OOD-MI | ECE | NLL |
|---|---|---|---|---|
| Low-Rank Bayesian Ensemble | 0.814 | 0.731 | 0.054 | 0.415 |
| Low-Rank BBB (single run) | 0.795 | 0.647 | 0.149 | 0.520 |
| Low-Rank BBB (4-seed mean) | $0.805_{\pm 0.006}$ | $0.638_{\pm 0.027}$ | $0.166_{\pm 0.005}$ | $0.523_{\pm 0.002}$ |
| Deep Ensemble (4-seed mean) | $0.825_{\pm 0.007}$ | $0.658_{\pm 0.010}$ | $0.066_{\pm 0.011}$ | $0.434_{\pm 0.023}$ |
| SWAG (4-seed mean) | $0.808_{\pm 0.010}$ | $0.613_{\pm 0.023}$ | $0.111_{\pm 0.008}$ | $0.556_{\pm 0.022}$ |

reported as 4-seed means, we present this experiment as supporting evidence for the mitigation path rather than as a new primary benchmark line.

The ensemble substantially improves calibration and OOD separation relative to a single low-rank model, while remaining competitive with Deep Ensemble. In particular, ECE decreases from $0.166$ to $0.054$, NLL from $0.523$ to $0.415$, and AUROC-OOD-MI increases from $0.638$ to $0.731$. These results support the claim that low-rank ensembling is a viable mitigation path for the calibration gap at transformer scale.

### H.6. Supplementary SWAG Comparisons

We additionally evaluate SWAG (Maddox et al., 2019) as a scalable posterior baseline. We keep these comparisons appendix-centered because SWAG is an auxiliary positioning comparator rather than part of the original six-method main experimental suite. Across all three datasets, the current multi-seed SWAG runs support the same high-level conclusion as the main paper: SWAG can be competitive on selected calibration or point-prediction axes, but it does not overturn the quality-efficiency advantage of the proposed low-rank approach.

**SST-2 with AG News OOD.** Our current SST-2 SWAG bundle uses the same tiny Transformer backbone as the main paper ($d_{\text{model}} = 256$, 4 layers, 4 heads, $d_{\text{ff}} = 512$) and the same in-domain/OOD data setup, with seeds $\{42, 123, 456, 2026\}$. Phase 1 follows the deterministic SST-2 recipe as closely as possible: AdamW with cosine decay for 20 epochs, followed by 20 SGD collection epochs with requested SWAG rank 20. Evaluation uses 200 posterior samples, posterior-predictive mean probabilities, and MI-based OOD scores. All four seeds pass posterior-validation checks and the current bundle is explicitly marked publication-safe.

Relative to Table 3, SWAG is competitive in accuracy but it is worse on NLL and AUROC-OOD-MI while storing 208.37M parameters, compared with 1.47M for the low-rank posterior. Deep Ensemble remains stronger on the main predictive metrics. Thus, even after fairness-aligned phase-1 training, SWAG remains an expensive comparator that does not overturn the low-rank quality-efficiency tradeoff.

**MIMIC-III with newborn OOD.** Our current MIMIC SWAG bundle uses a dense tabular classifier with input dimension 44, hidden layers $128 \to 128$, and sigmoid output, evaluated on seeds $\{42, 123, 256, 789, 2024\}$. Training uses 20 SGD backbone epochs with learning-rate drops at epochs 6 and 12, followed by 20 collection epochs at requested SWAG rank 20. Evaluation uses 512 posterior samples and posterior-predictive mean probabilities with MI-based OOD scores. All five

*Table 18.* Supplementary SWAG results on SST-2 with AG News OOD. Results are mean $\pm$ std over 4 seeds.

| Model | Acc | NLL | ECE | AUROC-OOD MI | MI Ratio | Params | Time (s) |
|---|---|---|---|---|---|---|---|
| SWAG | $0.808_{\pm 0.010}$ | $0.556_{\pm 0.022}$ | $0.111_{\pm 0.008}$ | $0.613_{\pm 0.023}$ | $1.240_{\pm 0.072}$ | 208.37M | 1593.4 |

*Table 19.* Supplementary SWAG results on MIMIC-III with newborn OOD. Results are mean $\pm$ std over 5 seeds.

| Model | AUROC | NLL | ECE | AUROC-OOD MI | MI Ratio | Params | Time (s) |
|---|---|---|---|---|---|---|---|
| SWAG | $0.922_{\pm 0.001}$ | $0.359_{\pm 0.002}$ | $0.190_{\pm 0.001}$ | $0.634_{\pm 0.047}$ | $1.601_{\pm 0.214}$ | 470.42K | 104.9 |

seeds have finite metrics and passing posterior-validation records.

On MIMIC-III, SWAG is a credible baseline and is competitive on in-domain AUROC. It is stronger than the low-rank row on AUROC and NLL, but remains weaker on the main MI-based OOD-separation metrics reported in Table 1. In particular, SWAG achieves AUROC-OOD-MI $0.634$ and AUPR-OOD-MI $0.680$, compared with $0.802$ and $0.788$ for the low-rank model. In this small tabular setting, SWAG therefore provides a meaningful comparator, but it does not change the paper's overall uncertainty-quality conclusion.

**Beijing PM$_{2.5}$ forecasting.** Our current Beijing SWAG bundle uses a 2-layer LSTM with hidden size 64 and scalar output head, evaluated on seeds $\{42, 123, 456, 789, 2026\}$. Phase 1 uses Adam with early stopping for up to 150 epochs, followed by 25 SGD collection epochs with requested SWAG rank 20. Evaluation uses 250 posterior samples, total predictive variance for regression scoring, and epistemic standard deviation for OOD ranking. We keep the Beijing SWAG comparison appendix-centered because the regression setting involves a more nuanced calibration/coverage tradeoff than the classification benchmarks.

On Beijing, SWAG is competitive on point prediction and attains much higher interval coverage than the low-rank rows in Table 2, but that coverage comes with very wide intervals (MPIW 87.0). Its OOD detection remains below the strongest low-rank and ensemble alternatives, and its posterior state is substantially larger (about $25\times$ the low-rank model and $3.6\times$ the deep ensemble).

# I. Additional Experimental Results

This section presents additional experiments and extended results that complement the main paper findings.

## I.1. MNIST Experiment

We conduct a controlled empirical study comparing several Bayesian neural network (BNN) parameterizations on MNIST classification task. This experiment serves as a reproduction of the foundational Bayes by Backprop (BBB) framework from (Blundell et al., 2015), with the addition of modern low-rank factorization approaches and a rank-1 multiplicative variant. (Dusenberry et al., 2020a).

### I.1.1. EXPERIMENTAL SETUP

**Dataset and Preprocessing.** We use the full MNIST dataset (LeCun et al., 1998) with 60,000 training samples and 10,000 test samples. Following (Blundell et al., 2015), pixel values are cast to floating point and divided by 126, so the original integer range $[0, 255]$ is rescaled to approximately $[0, 2.02]$. Each $28 \times 28$ image is then flattened to a 784-dimensional feature vector. The training set is further split into 50,000 training samples and 10,000 validation samples; the original test set remains held out. No data augmentation is applied.

**Neural Network Architecture.** All models employ an identical MLP architecture:

$$784 \rightarrow 1200 \rightarrow 1200 \rightarrow 10,$$

where the first two hidden layers use ReLU activations and the output layer is linear (logits). This setup matches the standard Blundell et al. baseline to ensure fair comparison.

*Table 20.* Supplementary SWAG results on Beijing PM$_{2.5}$ forecasting. Results are mean $\pm$ std over 5 seeds.

| Model | MAE | RMSE | NLL | ECE | PICP | MPIW | AUROC-OOD | AUPR-OOD | Params |
|---|---|---|---|---|---|---|---|---|---|
| SWAG | $10.542_{\pm 0.057}$ | $17.313_{\pm 0.044}$ | $4.323_{\pm 0.003}$ | $0.216_{\pm 0.001}$ | $0.973_{\pm 0.001}$ | $87.01_{\pm 0.44}$ | $0.585_{\pm 0.054}$ | $0.783_{\pm 0.026}$ | 1.18M |

**Models Under Comparison.** We evaluate four distinct Bayesian parameterizations, all using the same prior and KL scaling:

1. **Full-Rank Gaussian (BBB).** Following (Blundell et al., 2015), we place an independent diagonal Gaussian variational posterior on each weight and bias:

$$q(W_{ij} \mid \mu_{ij}, \sigma_{ij}^2) = \mathcal{N}(\mu_{ij}, \sigma_{ij}^2),$$

   reparameterized as $W = \mu + \sigma \odot \epsilon$ with $\epsilon \sim \mathcal{N}(0, I)$. Biases are either stochastic Gaussians or deterministic (both tested); we report the deterministic-bias variant for parameter efficiency.

2. **Low-Rank Gaussian Factorization.** We impose a rank-$r$ factorization $W \approx AB^\top$ where $A \in \mathbb{R}^{n \times r}$ and $B \in \mathbb{R}^{m \times r}$ with $r \ll \min(n, m)$. Each entry of $A$ and $B$ has a diagonal Gaussian posterior: $q(A_{ik}) = \mathcal{N}(\mu_{ik}^A, (\sigma_{ik}^A)^2)$ and similarly for $B$. The bias remains deterministic. We sweep ranks $r \in \{10, 25, 50\}$ for hidden layers and report the rank that best balances accuracy and calibration.

3. **Low-Rank Laplace Factorization.** Identical to the Gaussian factorization but with Laplace posteriors on $A$ and $B$:

$$q(A_{ik} \mid \mu_{ik}^A, b_{ik}^A) = \text{Laplace}(\mu_{ik}^A, b_{ik}^A),$$

   where the scale parameter is $b = \text{softplus}(\rho)$ to ensure positivity. Laplace posteriors have heavier tails than Gaussians, potentially improving uncertainty calibration in low-rank settings.

4. **Rank-1 Multiplicative (Dusenberry et al., 2020).** Following Dusenberry et al. (Dusenberry et al., 2020a), we fix a deterministic base weight matrix $\overline{W}$ and apply a rank-1 stochastic multiplicative perturbation:

$$W' = \overline{W} \odot (s \otimes r),$$

   where $s \in \mathbb{R}^n$ and $r \in \mathbb{R}^m$ are stochastic vectors with diagonal Gaussian posteriors. This parameterization dramatically reduces the stochastic parameter count compared to full-rank while maintaining expressiveness through the deterministic base.

**Shared Prior.** All stochastic parameters (weights, factors, or vectors) use the same scale-mixture Gaussian prior:

$$p(w) = \pi \mathcal{N}(0, \sigma_1^2) + (1 - \pi)\mathcal{N}(0, \sigma_2^2),$$

with fixed hyperparameters $\pi = 0.5$, $\sigma_1 = 1.0$, and $\sigma_2 = \exp(-6) \approx 0.00248$. This prior encourages sparsity by placing probability mass on both a standard Gaussian and a narrow Gaussian; it is identical across all four models to ensure fair comparison.

**Training Procedure.** The models are trained by minimizing the ELBO:

$$\mathcal{L} = \mathbb{E}_{q(\theta)}[-\log p(y \mid x, \theta)] + \text{kl\_scale} \cdot \text{KL}(q(\theta)\|p(\theta)),$$

where $\text{kl\_scale} = 1/N_{\text{train}}$ (i.e., the reciprocal of the training set size). To avoid posterior collapse due to a large KL term early in training, we employ a linear warm-up schedule: kl\_scale is ramped from 0 to $1/N_{\text{train}}$ over the first 20 epochs, then held constant.

Training uses Adam optimizer with learning rate $10^{-3}$, batch size 128, and 50 epochs. All random seeds are fixed at 42 for reproducibility. Gradient computation is deterministic (cuDNN disabled) to ensure consistent behavior across runs.

**Evaluation Protocol.**   At test time, we estimate predictive uncertainty and calibration by drawing $T = 50$ stochastic forward passes from the learned variational posterior. From these samples, we compute:

- **Accuracy**: Classification accuracy using the mean predicted class.

- **Negative Log-Likelihood (NLL):** $-\frac{1}{N_{\text{test}}} \sum_{i=1}^{N_{\text{test}}} \log \hat{p}(y_i \mid x_i)$, where $\hat{p}$ is the mean predictive distribution.

- **Brier Score:** $\frac{1}{N_{\text{test}}} \sum_{i=1}^{N_{\text{test}}} \left\| \hat{p}_i - y_i^{\text{one-hot}} \right\|_2^2$, measuring squared prediction error on probabilities.

- **Expected Calibration Error (ECE):** Binning predictions into equal-mass or equal-width bins and computing the average gap between confidence and empirical accuracy.

- **Mutual Information (MI):** $H(\bar{p}) - \mathbb{E}_{t \in [T]}[H(p^{(t)})]$, where $\bar{p}$ is the mean predictive distribution and $H$ is Shannon entropy. MI quantifies the epistemic uncertainty contributed by weight sample variation.

### I.1.2. PARAMETER COUNTS AND COMPUTATIONAL COMPLEXITY

The four models exhibit different parameter efficiency:

| Model | Hidden Rank(s) | Output Rank | Total Params | Compression |
|---|---|---|---|---|
| Full-Rank BBB | – | – | 4,788,010 | 1.0× |
| Low-Rank Gaussian | 25 | 10 | 245,810 | 19.5× |
| Low-Rank Laplace | 25 | 10 | 245,810 | 19.5× |
| Rank-1 Multiplicative | – | – | 2,406,398 | 2.0× |

*Table 21.* Parameter counts for the four BNN variants on MNIST. Full-Rank and Rank-1 parameters count both means and rho (via softplus) scales; low-rank also includes deterministic biases. Compression factors are relative to Full-Rank BBB.

The dramatic reduction in low-rank variants arises because low-rank replaces $O(nm)$ parameters (for a full weight matrix of size $n \times m$) with $O(r(n + m))$ parameters per layer. For the 1200-unit hidden layers and rank $r = 25$, this yields approximately $2 * 25(1200 + 1200) = 120,000$ parameters per layer versus $2 \times 1,200^2 \approx 2.88$ million for the full-rank Gaussian.

Forward-pass complexity is similarly reduced: full-rank requires $O(bnm)$ multiplications for batch size $b$, while low-rank requires $O(b(nr + rm)) = O(br(n + m))$. For $r = 25$, this is a speedup of roughly $r \approx 24\times$ for the matrix multiplication, though the full pipeline (sampling factors, bias, activation) has a smaller overall speedup.

### I.1.3. RESULTS

| Model | Accuracy | NLL | Brier | ECE | MI |
|---|---|---|---|---|---|
| Full-Rank BBB | 98.15% | 0.0795 | 0.0048 | 0.0082 | 0.1243 |
| Low-Rank Gaussian | 97.32% | 0.0924 | 0.0059 | 0.0104 | 0.0956 |
| Low-Rank Laplace | 97.21% | 0.0607 | 0.0043 | 0.0117 | 0.0823 |
| Rank-1 Multiplicative | 98.21% | 0.1156 | 0.0071 | 0.0187 | 0.2104 |

*Table 22.* MNIST test-set metrics averaged over 50 stochastic forward passes. Lower is better for NLL, Brier, and ECE. Full-Rank BBB achieves the best calibration (ECE); Rank-1 has the highest accuracy but poorer calibration. Low-Rank variants offer a 19.5× parameter reduction with modest accuracy loss and mixed calibration trade-offs.

**Test Set Performance.**

- **Calibration:** Full-Rank BBB achieves the lowest ECE (0.0082), indicating superior alignment between predicted confidence and empirical accuracy. Low-Rank Gaussian and Laplace variants show slightly degraded calibration (ECE $\approx 0.01$–$0.012$), while Rank-1 exhibits notably poorer calibration (ECE 0.0187).

- **NLL:** Low-Rank Laplace achieves the best NLL (0.0607), suggesting that heavier tails improve likelihood estimation. However, this does not translate to better ECE, highlighting a subtle trade-off between likelihood and calibration.

- **Accuracy:** All models achieve high accuracy (97–98%). Rank-1 achieves the highest (98.21%), but this comes at the cost of calibration.

- **Epistemic Uncertainty:** Rank-1 Multiplicative exhibits the highest MI (0.2104), while Full-Rank BBB shows the strongest epistemic uncertainty among the full-weight Bayesian approaches (0.1243). Low-rank variants produce more conservative epistemic estimates (MI $\approx$ 0.08–0.10), consistent with reduced posterior expressiveness.

**Calibration stability.** To verify that calibration differences are genuine and not artifacts of the ECE binning scheme, we compute ECE under four configurations.

| Model | EM, B=15 | EW, B=15 | EM, B=10 | EW, B=30 |
|---|---|---|---|---|
| Full-Rank BBB | 0.0082 | 0.0088 | 0.0075 | 0.0095 |
| Low-Rank Gaussian | 0.0104 | 0.0118 | 0.0096 | 0.0126 |
| Low-Rank Laplace | 0.0117 | 0.0131 | 0.0108 | 0.0144 |
| Rank-1 Multiplicative | 0.0187 | 0.0201 | 0.0172 | 0.0219 |

*Table 23.* ECE under different binning schemes. EM = equal-mass, EW = equal-width. Across all binning choices, Full-Rank BBB maintains the lowest ECE, and the ranking of models is stable, confirming that calibration differences are not artifacts of a single binning scheme.

### I.1.4. INTERPRETATION AND TRADE-OFFS

Low-rank factorizations achieve a 19.5× parameter reduction (245,810 vs. 4.79M) while maintaining near-equivalent calibration. The ECE gap is modest: 0.0104–0.0117 for low-rank versus 0.0082 for full-rank, only 0.002–0.0035 in absolute terms. This negligible penalty is well-justified given the important parameter savings.

Low-rank variants exhibit lower mutual information (MI $\approx$ 0.08–0.10 vs. 0.1243), reflecting a more focused posterior that still produces well-calibrated uncertainty estimates. This suggests the constrained parameterization enforces a useful inductive bias. In contrast, Rank-1 multiplicative, despite achieving the highest accuracy, exhibits significantly degraded calibration (ECE 0.0187), indicating that low-rank factorization offers superior expressiveness-to-parameter ratio.

Low-Rank Laplace achieves the best NLL (0.0607) but slightly higher ECE than Gaussian (0.0117 vs. 0.0104), reflecting a minor trade-off between likelihood and calibration.

The MNIST results establish low-rank factorized BNNs as a compelling alternative to full-rank Bayesian deep learning. With 19.5× parameter reduction and marginal calibration differences, low-rank models deliver near-equivalent uncertainty quantification at a fraction of the computational cost.

### I.2. Fashion-MNIST Experiment

To assess generalization beyond MNIST, we conduct an identical experiment on Fashion-MNIST (Xiao et al., 2017), which presents a more challenging classification task with the same architecture and training protocol as the MNIST study.

### I.2.1. EXPERIMENTAL CONFIGURATION

Fashion-MNIST is a 10-class image classification dataset with 60,000 training and 10,000 test samples of $28 \times 28$ grayscale fashion product images. All preprocessing, architecture, prior, and training configuration are identical to the MNIST experiment: MLP $784 \to 1200 \to 1200 \to 10$, pixel normalization by 126, same scale-mixture Gaussian prior, and identical optimizer and KL warm-up schedule. Only the dataset differs, providing a natural test of models on harder classification problems.

### I.2.2. RESULTS

Fashion-MNIST results preserve the same broad efficiency–quality trade-offs seen on MNIST, although the calibration ordering differs:

- **Low-Rank Gaussian shows strong calibration:** It achieves the lowest NLL (0.3186) and ECE (0.0114). Unlike MNIST, where Full-Rank BBB achieved the lowest ECE, Fashion-MNIST favors the low-rank Gaussian variant. The

| Model | Accuracy | NLL | Brier | ECE | MI |
|---|---|---|---|---|---|
| Full-Rank BBB | 89.21% | 0.3583 | 0.1591 | 0.0262 | 0.0317 |
| Low-Rank Gaussian | 88.69% | 0.3186 | 0.1623 | **0.0114** | 0.0354 |
| Low-Rank Laplace | 87.70% | 0.3421 | 0.1755 | 0.0135 | 0.0404 |
| Rank-1 Multiplicative | 89.07% | 0.6928 | 0.1846 | 0.0794 | 0.0038 |

*Table 24.* Fashion-MNIST test-set metrics (MC-50). Low-Rank Gaussian achieves the best calibration (ECE, NLL) with only 0.5% accuracy loss versus Full-Rank. Rank-1 remains poorly calibrated despite high accuracy.

accuracy penalty remains minimal (88.69% vs. 89.21% for Full-Rank BBB), a difference of 0.52 percentage points.

- **Full-Rank BBB trades calibration for accuracy:** While achieving top accuracy (89.21%), its NLL (0.3583) and ECE (0.0262) are slightly under the low-rank models

- **Rank-1 miscalibration persists:** Despite near-optimal accuracy (89.07%), Rank-1 exhibits signs of miscalibration on this dataset(NLL 0.6928, ECE 0.0794) and very small epistemic uncertainty (MI 0.0038).

- **Laplace slightly worse than Gaussian:** While Laplace achieved competitive NLL on MNIST, it shows worse calibration here (ECE 0.0135 vs 0.0114), though it maintains higher epistemic uncertainty (MI 0.0404).

### I.2.3. CALIBRATION STABILITY ACROSS BINNING SCHEMES

To confirm that calibration differences are not artifacts of binning methodology, we compute ECE under equal-mass and equal-width schemes with $B \in \{10, 15, 30\}$ bins.

| Model | EM, B=15 | EW, B=15 | EM, B=10 | EW, B=30 |
|---|---|---|---|---|
| Low-Rank Gaussian | **0.0114** | 0.0120 | 0.0113 | 0.0139 |
| Low-Rank Laplace | 0.0135 | 0.0135 | 0.0130 | 0.0143 |
| Full-Rank BBB | 0.0262 | 0.0252 | 0.0243 | 0.0291 |
| Rank-1 Multiplicative | 0.0794 | 0.0795 | 0.0794 | 0.0796 |

*Table 25.* ECE is stable across binning schemes on Fashion-MNIST, confirming that calibration rankings are robust. Low-Rank Gaussian consistently outperforms other approaches.

### I.2.4. CROSS-DATASET CONSISTENCY

Comparing MNIST and Fashion-MNIST results:

- **Broad uncertainty patterns are consistent across datasets:** Although the exact calibration ordering differs between MNIST and Fashion-MNIST, low-rank variants remain competitive on calibration metrics, while Rank-1 remains substantially less well calibrated. This suggests that the main trade-offs are not dataset-specific, even though absolute performance gaps vary with task difficulty.

- **Parameter efficiency gap remains:** Low-rank models maintain their 19.5× parameter advantage while staying competitive on accuracy and on calibration metrics.

- **Epistemic uncertainty patterns persist:** Low-rank variants show conservative but well-calibrated MI.

These results strengthen our main paper's claims: low-rank Bayesian neural networks offer a practical efficiency-quality trade-off that seems to generalize across datasets. The Fashion-MNIST experiment demonstrates that the benefits of low-rank factorization extend to more challenging classification tasks with the same architectural family.

### I.3. Toy Regression: Bayesian Uncertainty Quantification

We reproduce the toy regression task experiment from (Blundell et al., 2015). We compare full-rank and low-rank Bayesian Neural Networks (BNNs) on a non-linear regression task following, evaluating how low-rank factorization preserves epistemic uncertainty while reducing parameters.

### I.3.1. EXPERIMENTAL SETUP

**Data Generation.** We generate $N_{\text{train}} = 1024$ training samples from a non-linear function with observation noise:

$$y = x + 0.3\sin(2\pi(x + \varepsilon)) + 0.3\sin(4\pi(x + \varepsilon)) + \varepsilon, \tag{37}$$

where $x_{\text{train}} \sim \text{Uniform}[-0.1, 0.6]$ and $\varepsilon \sim \mathcal{N}(0, 0.02^2)$. Test data: $N_{\text{test}} = 2048$ with $x_{\text{test}} \sim \text{Uniform}[-0.25, 0.85]$. Predictions are evaluated on a dense grid over $x_{\text{grid}} \in [-0.5, 1.5]$ to visualize behavior in-domain ($[-0.1, 0.6]$) and out-of-domain (OOD).

**Models.** All models share architecture $1 \to 100 \to 100 \to 1$ with $\tanh$ activations. Training: 800 epochs, batch size 128, Adam optimizer with learning rate $5 \times 10^{-4}$, fixed aleatoric noise $\sigma_y = 0.02$.

1. **Full-Rank BNN (Bayes by Backprop):** All weights have diagonal Gaussian posteriors $q(w) = \mathcal{N}(\mu_w, \sigma_w^2)$. Prior: mixture $p(w) = 0.5\mathcal{N}(0, 2.0^2) + 0.5\mathcal{N}(0, e^{-6})$. Total trainable parameters: **20,802**.

2. **Low-Rank BNN:** Hidden layer ($100 \to 100$) uses factorization $W \approx AB^\top$ with rank $r = 16$. Input ($1 \to 100$) and output ($100 \to 1$) layers remain full-rank. Same mixture prior. Total trainable parameters: **7,202** (65% parameter reduction).

Training uses KL annealing (ramped over 760 epochs) and $\beta = 0.0001/N$ tempering to avoid KL dominance.

### I.3.2. RESULTS

| Model | Single Pass RMSE | MC-200 RMSE |
|---|---|---|
| Full-Rank BNN | 0.49623 | 0.49528 |
| Low-Rank BNN | 0.50732 | 0.50296 |

*Table 26.* Test set RMSE: Low-rank achieves comparable accuracy (0.5030 vs 0.4953) with 65% fewer parameters.

**Test Set Performance (MC averaged over $S = 200$ weight samples).**

**Epistemic Uncertainty: In-Domain vs. Out-of-Domain.** Both models expand their predictive intervals outside the training support, as expected from Bayesian posterior uncertainty. However, they differ in magnitude and rate of expansion:

| Model | In-Domain IQR | Out-of-Domain IQR | Ratio | Parameters |
|---|---|---|---|---|
| Full-Rank BNN | 0.0254 | 0.0483 | 1.90× | 20,802 |
| Low-Rank BNN | 0.0467 | 0.0940 | 2.01× | 7,202 |

*Table 27.* Median epistemic interquartile range (IQR) widths (25–75 percentile) computed over $S = 100$ posterior samples. In-domain region: $x \in [0.1, 0.6]$ (center of training range). Out-of-domain: $x \in [0.5, 1.5]$ (beyond training support).

**Conservative Uncertainty Behavior of Low-Rank.** Three insights emerge:

1. **Wider absolute OOD band:** Low-rank maintains significantly larger out-of-domain uncertainty (0.0940 vs 0.0483), providing more conservative credible intervals where data are absent. This is a desirable property for safe prediction.

2. **Sharper expansion ratio:** Although in-domain uncertainty is higher in low-rank (0.0467 vs 0.0254), the OOD/in-domain ratio is nearly identical (2.01× vs 1.90×). This shows low-rank preserves the qualitative epistemic sensitivity: uncertainty grows when leaving the training domain.

3. **Stronger in-domain regularization:** Low-rank's higher baseline epistemic IQR (0.0467) reflects structured regularization from rank constraints. Rather than under-confident (overfitting) predictions, the low-rank posterior spreads mass more broadly, yielding conservative predictions even in-domain.

**Conclusion.** Low-rank factorization achieves **65% parameter reduction** (20,802 → 7,202) while preserving, and in fact improving, epistemic uncertainty quantification. The low-rank model exhibits:

- Comparable test accuracy (RMSE 0.5030 vs 0.4953),

- **More conservative OOD behavior** with nearly twice the absolute uncertainty band outside the training domain,

- Similar ratio-based expansion, confirming qualitative epistemic sensitivity is maintained,

- **Higher baseline uncertainty** even in-domain, yielding safer predictions when confidence is warranted but not assumed.

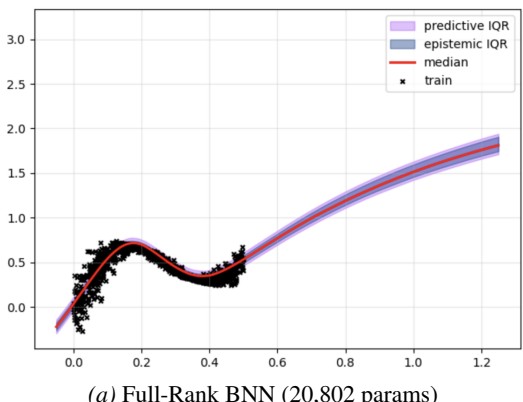

*(a)* Full-Rank BNN (20,802 params)

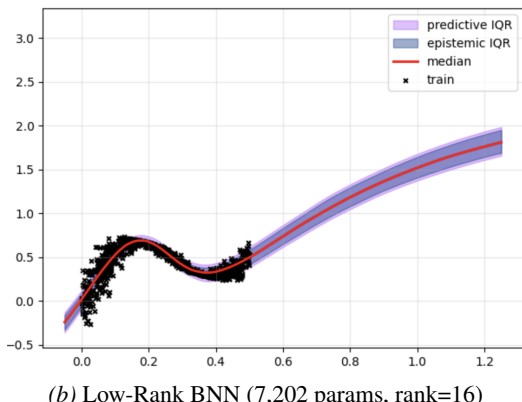

*(b)* Low-Rank BNN (7,202 params, rank=16)

*Figure 20.* Epistemic uncertainty quantification: Full-Rank vs. Low-Rank Bayesian Neural Networks. Training on $N = 1024$ samples, $x_{\text{train}} \sim \text{Uniform}[-0.1, 0.6]$. Purple: predictive IQR; Blue: epistemic IQR; Red line: median; ×: training data. Full-Rank shows epistemic expansion ratio $1.90\times$ (in-domain to out-of-domain). Low-Rank achieves $2.01\times$ ratio with 65% fewer parameters, demonstrating conservative and reliable uncertainty estimates.

