# OpenReview forum: "Singular Bayesian Neural Networks"
_ICML.cc/2026/Conference — ICML 2026 regular_

### Official Review · Reviewer_GhLS · 2026-02-22

**Soundness:** 3
**Presentation:** 3
**Significance:** 3
**Originality:** 2
**Overall Recommendation:** 5
**Confidence:** 4

**Summary:**

The paper studies **low-rank Bayesian neural networks** by parameterizing each dense weight matrix as
$$
W = AB^\\top,\\qquad A\\in\\mathbb{R}^{n\\times r},\\; B\\in\\mathbb{R}^{m\\times r},
$$
with variational posteriors on the factors trained via an ELBO. The induced distribution over $W$ is supported on the rank-$ r$ set.

Empirically, the paper positions low-rank BBB as a competitive single-model approach for uncertainty and OOD detection under strong parameter constraints (notably in a small Transformer sentiment setup where embeddings dominate parameters), while also discussing that end-to-end runtime gains can be smaller than layer-level FLOP reductions due to sampling and kernel-launch overhead.

**Compliance With Llm Reviewing Policy:**

Affirmed.

**Final Justification:**

The Rebuttal addressed my concerns

**Key Questions For Authors:**

1. How does your method relate to function-space post-hoc uncertainty approaches such as Linearized Laplace Approximation (LLA) variants, Fixed-Mean Gaussian Processes (FMGP), and GP-last-layer methods like SNGP—and in what regimes would you expect your low-rank weight-space posterior to outperform or complement these alternatives?

2.  What are the end-to-end wall-clock and peak-memory trade-offs as a function of rank $r$ and the number of stochastic forward passes—do the observed training and inference speedups track the layer-level FLOP reductions, and where do overheads dominate?

3. Reviewing the code I noticed there are multiple KL-scaling defined across modules, what is the effective KL coefficient used in the reported runs, and could this be consolidated into a single clearly documented scaling rule to make reproduction unambiguous?

**Limitations:**

yes

**Strengths And Weaknesses:**

## Strengths
- **Clear and practical low-rank variational parameterization.** The factorization is straightforward, integrates naturally into standard VI training, and yields an interpretable knob $r$ controlling capacity and memory.
- **Uncertainty- and OOD-focused evaluation.** The paper reports calibration/coverage and OOD detection metrics in multiple settings rather than only accuracy.
- **Compute narrative is internally consistent at the layer level.** The appendix provides the dense-layer matmul reduction from $O(bnm)$ to $O(br(n+m))$ (batch size $b$), which is the correct structural benefit of low-rank parameterizations.

---

## Weaknesses / gaps
- **Efficiency claims need clearer end-to-end evidence.** Layer-level FLOPs can improve while wall-clock time does not, especially for small/medium layers due to sampling overhead and GPU kernel dispatch. A method paper should explicitly report throughput and memory across ranks and MC sample counts.
- **Positioning relative to function-space uncertainty is incomplete.** The introduction motivates the work using the mismatch between weight-space inference and function-space uncertainty; however, the related-work and baselines omit modern *function-space* post-hoc approaches.

## Missing related work / baselines (function-space post-hoc)
Given the paper’s motivation, it would be important to mention and ideally benchmark against **function-space post-hoc Bayesian deep learning** methods, especially those that preserve the pretrained predictor’s mean and focus on covariance/uncertainty, these are 2 examples:

1. **Linearized Laplace Approximation (LLA) and scalable variants.**
   LLA places a local Gaussian approximation around a pretrained network (often via GGN/KFAC approximations), yielding post-hoc uncertainty without retraining the mean. Several works interpret the linearized model as an induced GP and focus on scalable covariance approximations (Nyström, variational approximations, etc.). This is closely aligned with the “post-hoc uncertainty at scale” goal.

2. **Spectral-normalized Neural Gaussian Processes (SNGP)**.
  SNGPs replace the last layer with a GP and use spectral normalization to encourage distance-awareness, producing calibrated uncertainty and often strong OOD detection without full Bayesian inference over all weights. This line of work targets a similar goal—single-model uncertainty with practical scalability—but makes a different approximation trade-off (a GP head rather than a low-rank posterior over internal weights).

3. **Fixed-Mean Gaussian Processes (FMGP).**
   FMGP is explicitly post-hoc and function-space: it fixes the predictive mean to the pretrained model output and optimizes *only predictive variances* via sparse variational GP machinery. The method is presented as architecture-agnostic (no Jacobians) and related to LLA, but with kernel flexibility and different scaling characteristics.

---

> ### Author Rebuttal · Authors · 2026-03-27
>
> We thank the reviewer for the detailed and code-aware feedback.
>
> Regarding function-space and post-hoc methods, we agree that the paper should distinguish more clearly between two adjacent families: post-hoc methods around a trained deterministic backbone, such as LLA and FMGP, and deterministic feature-extractor methods with GP-style output layers, such as SNGP. Our method differs from both in learning a structured low-rank weight-space posterior end-to-end from initialization. For the first family (e.g., LLA/FMGP), uncertainty is added conditional on a fixed pretrained predictor or learned representation. For the second (e.g., SNGP), uncertainty is concentrated in a GP-style output layer on top of a deterministic feature extractor. By contrast, our goal is to model uncertainty throughout the network, including in the feature extractor itself. We will make this taxonomy explicit in the related-work discussion and clarify that these methods are complementary rather than direct substitutes for the present construction. For completeness, we also report additional SWAG comparisons in our response to Reviewer Gk8P, since SWAG is another relevant post-hoc uncertainty baseline even though it is not the same family as LLA/FMGP/SNGP.
>
> In terms of regime, we would expect our approach to be the most performant and **reliable** when uncertainty in the learned representation itself matters, since the posterior is learned jointly throughout the network rather than added after a deterministic backbone is fixed. While post-hoc and final-layer methods are computationally attractive, efficiency alone does not settle uncertainty quality: recent work has questioned how cleanly such uncertainty decompositions are disentangled. We therefore view these methods as complementary approximations around a fixed predictor rather than interchangeable with end-to-end uncertainty learning.
>
> Regarding KL scaling, the choices are not arbitrary, and we did try to be explicit about them in the appendix and in the experiment code in the supplementary material. That said, we agree that the logic behind these choices can be stated more clearly and gathered more centrally. There are two distinct roles played by the KL prefactor. First, in stochastic variational inference there is a principled minibatch normalization coming directly from the full-data ELBO: when the minibatch likelihood term is written as a sum, the KL is distributed across minibatches with weight $1/\\#\text{batches}$; when the likelihood term is averaged, the equivalent scaling is of order $1/N_{\mathrm{train}}$. For truncated-sequence RNNs, the same logic distributes the KL across minibatches and truncation windows. Second, any additional coefficient beyond this is a deliberate tempering choice rather than a requirement of variational Bayes. In generalized/tempered Bayes terms, weaker KL weighting makes the objective more data-dominated, whereas stronger KL weighting keeps the posterior closer to the prior; this changes posterior concentration and can therefore affect calibration and OOD behavior. In our experiments, this base normalization is combined with annealing, and in the Transformer setting the embedding layers use a smaller KL multiplier so that the much larger embedding block does not dominate the regularization. We will revise the paper to add a more centralized explanation of these choices, so that the exact effective KL scale, its normalization basis, the annealing schedule, and architecture-specific adjustments are easier to track and reproduce, as in the code we will make available.
>
> Regarding efficiency, as also clarified in our response to Reviewer PHKp, we distinguish end-to-end training time from matched steady-state profiling cost. The paper already reports end-to-end timings, and the controlled SST-2 profiling benchmark isolates per-epoch computational cost. We additionally profiled SST-2 across seven low-rank settings, varying both rank controls used in our Transformer network: $(r_{\mathrm{emb}}, r)\in\{(4,8),(8,12),(10,16),(12,24),(16,32),(24,48),(32,64)\}$. As expected, trainable parameter count scales much more directly with these rank settings than wall-clock time does: it increases from 0.50M to 3.86M (about $7.7\\times$) across the sweep. The practical cost metrics move in the same direction, but much more mildly: average epoch time increases from 5.06s to 5.95s (about $1.18\\times$), peak GPU memory from 338MB to 456MB (about $1.35\\times$), single-pass inference from 0.214s to 0.239s (about $1.12\\times$), and MC inference by about $1.07$-$1.39\\times$ from MC20 to MC100. Thus lower rank does provide lower memory use and faster training/inference in practice, but the gain is sublinear in the parameter reduction: e.g., a 50\% parameter reduction should not be interpreted as a 50\% wall-clock reduction. We will make this distinction more explicit in the revision.

---

> > ### Author Rebuttal · Reviewer_GhLS · 2026-03-31
> >
> > I remain positive on the submission. The paper makes a clear and practically useful contribution: a low-rank variational parameterization for BNNs that induces structured correlations, reduces parameter cost, and is supported by nontrivial theory and broad empirical evaluation. The rebuttal satisfactorily addressed my main concerns by clarifying the relationship to function-space/post-hoc uncertainty methods, providing additional profiling and baseline evidence, and explaining the effective KL-scaling choices more clearly. I also find the revised positioning more balanced: the method should be understood as a strong single-model quality-efficiency tradeoff rather than a uniformly dominant uncertainty method. While larger-scale experiments and broader baseline coverage would strengthen the work further, I do not see these remaining issues as outweighing the paper’s technical merit and likely impact.

---

> > > ### Author Response · Authors · 2026-04-01
> > >
> > > Thank you for the careful follow-up. We are glad the clarifications on function-space/post-hoc methods, the additional profiling and baseline evidence, and the centralized explanation of the KL scaling choices addressed your concerns.
> > >
> > > We also appreciate your summary of the paper's contribution and your reading of the revised positioning. We agree that the right way to present the method is as a strong single-model quality-efficiency trade-off, rather than as a uniformly dominant uncertainty method, and we will keep that framing explicit in the revision.
> > >
> > > Thank you again for the thoughtful review and helpful suggestions throughout.

---

### Official Review · Reviewer_PHKp · 2026-03-12

**Soundness:** 3
**Presentation:** 3
**Significance:** 3
**Originality:** 3
**Overall Recommendation:** 5
**Confidence:** 3

**Summary:**

This paper proposes a low-rank variational inference framework for BNNs that involves replacing weight matrices W by a low-rank approximation W = AB^T, with mean-field Gaussian posteriors on A & B. This induces a singular posterior, which the authors prove has better PAC-Bayes and Gaussian complexity generalization bounds. Empirically, the authors show this method performs well while being more parameter-efficient.

**Compliance With Llm Reviewing Policy:**

Affirmed.

**Final Justification:**

This is a solid paper that proposes a more compute-efficient approach to uncertainty calibration in BNNs based on enforcing a rank constraint on weight matrices, with supporting theory and experiments.

The rebuttal addressed my main concerns around whether or not the scaling demonstrations at a particular scale were sufficiently informative (when the real test is how the technique continues to scale as model size increases), disambiguation of the term "singular", and better contextualization within related literature on reduced-rank methods.

**Key Questions For Authors:**

* How does the technique scale, really? From the very first sentence of the abstract, the authors argue that one of the benefits of their approach is greater scaling. There is some discussion on page 51 that the technique outperforms full-rank models for transformers with millions of parameters, but this is still very small scale by modern standards. Something like a simple plot of wall-clock time and memory requirements as a function of the number of parameters used by this technique and others would go a long way towards confirming this benefit.
* Could the authors comment on how their framework relates to the literature on singular learning theory (SLT; Watanabe, 2009)? The term “singular” in the title and the focus on posteriors concentrated on algebraic varieties (e.g., rank-r matrices here) directly connects to this body of work. Watanabe’s theory provides tools for analyzing the posterior concentration and generalization error of models with singular Fisher information matrices, which is precisely the setting here. At minimum, the relationship should be discussed. As is, I think the current title “Singular Bayesian Neural Networks” is a bit too general given the existing literature on SLT, when the technique is really something more like “reduced-rank” or “matrix-factorized BNNs.” I'd recommend the authors change their title.
* Could the authors comment on how their framework relates to reduced-rank regression (obvious link given shared dependence on Eckart-Young theorem), probabilistic/Bayesian matrix factorization (see [here](https://papers.nips.cc/paper_files/paper/2007/file/d7322ed717dedf1eb4e6e52a37ea7bcd-Paper.pdf)), recent work within the compression literature on low-rank factorization ([example](https://ieeexplore.ieee.org/document/6638949))? Generally, I would encourage the authors to use their additional space for the camera-ready for fleshing out the related works section.

If the authors address these points or convince me that I'm mistaken about some point, I would change my recommendation to accept.

Minor things:

* The figure texts are very small (and almost all illegible). Please increase the size of the labels and/or increase the size of the figures in the final camera-ready.

Typos:

* 103 col 1: “(MLPs)(Rumelhart et al., 1986)” is missing a space. (I’d recommend “(MLPs; Rumelhart et al., 1986)”)
* 108 col 2: Missing space after “Jensen’s inequality:”

**Limitations:**

Yes.

**Strengths And Weaknesses:**

* **Soundness**: The paper is quite thorough, with several proofs, and lots of experiments across many datasets and architectures (altogether 65 pages with appendices). This lends confidence to the empirical results but means I was unable to evaluate everything (especially the theoretical contributions) with the same level of care .
* **Presentation**: The paper is well-written. The theoretical results are cleanly presented.
* **Significance**: The most practically useful result is probably the competitive performance at a higher parameter-efficiency.
* **Originality**: I am aware of no prior work that studies reduced-rank BNN in particular.

I thought this was a solid paper with clean theoretical results and reasonable experiments. The idea of placing variational posteriors on low-rank factors rather than full weight matrices is pretty natural and well-executed. The main weakness seems to be the small experimental scale (when scaling is listed as one of the main benefits). I lean accept, especially if the authors provide preliminary scaling results. I also think the work can still contextualize itself better within related literature (singular learning theory, Bayesian matrix factorization, low-rank factorization -- there is some discussion of related works that was cut to an appendix, but this focuses on a more narrow set of comparisons within the BNN literature).

---

> ### Author Rebuttal · Authors · 2026-03-27
>
> We thank the reviewer for the thoughtful and actionable feedback.
>
> Regarding scaling, the end-to-end SST-2 results in the paper show that Low-Rank BBB trains faster than both Full-Rank BBB and a 5-member Deep Ensemble under the actual training pipeline: 8.22 min for Low-Rank BBB versus 23.1 min for Full-Rank BBB and 64.7 min for Deep Ensemble.
> To clarify what part of this gain comes from raw computational cost, we also ran a controlled SST-2 Transformer profiling benchmark that fixes the warmup and measured optimization-step budget for every method and uses matched  *train\_on\_batch(...)* loops to isolate steady-state per-step/per-epoch cost. Under this controlled profile, Low-Rank BBB uses $33.7$X fewer total trainable parameters than Deep Ensemble (1.47M vs. 49.61M), $1.87$X less peak GPU memory (357.5MB vs. 670.1MB), and is $3.23$X faster per epoch. Relative to Full-Rank BBB, it uses $13.5$X fewer trainable parameters (1.47M vs. 19.84M), about $2$X less peak GPU memory (357.5MB vs. 721.1MB), and gives a modest $\approx$9% reduction in per-epoch wall-clock time. Taken together, these results show that the efficiency advantage is not only a
> parameter-count effect. The controlled
> profile shows that it is accompanied by consistent savings in peak GPU memory, and per-epoch cost. Relative to Full-Rank BBB, the
> matched per-epoch gain is modest; however, under the actual end-to-end
> *callback-based* training pipeline, Low-Rank BBB reaches its final selected model
> much earlier (8.22 min vs. 23.1 min) while also outperforming Full-Rank BBB on
> **all** reported SST-2 metrics in our summary. Relative to Deep Ensemble, the advantage is large under both views of
> efficiency: **total parameters, peak memory, per-epoch cost, and end-to-end
> time-to-train.**
>
> We also report a rank-sweep profiling study in our response to Reviewer GhLS, where varying $(r_{\text{emb}}, r)$ from $(4,8)$ to $(32,64)$ increases trainable parameters from 0.50M to 3.86M, while average epoch time rises from 5.06s to 5.95s and peak GPU memory from 338MB to 456MB, clarifying how practical cost scales with model size. We will revise the discussion to distinguish more explicitly
> between steady-state per-epoch cost and actual end-to-end training time under the reported training protocol.
>
> Regarding SLT, we agree that there is a meaningful
> connection, since the low-rank factorization introduces algebraic structure and
> non-identifiability, placing the models in our paper among singular models in
> Watanabe's terminology. As the reviewer notes, the focus of Watanabe's theory is
> primarily asymptotic. However, our use of "singular" in Theorem 3.4 is
> narrower: it refers to measure-theoretic singularity of the induced posterior
> $q_W$ with respect to Lebesgue measure on weight space, not to the asymptotic
> RLCT/Fisher-geometry analysis of singular statistical models in Watanabe's
> sense. Thus the two notions are related in spirit but are not identical. We
> will add a short remark in the introduction or theory section making this
> distinction explicit, so that the relationship to SLT is acknowledged without
> conflating the two settings. We also understand the concern about the title, and while our current preference is to keep it, we will add a clear disambiguating sentence or a clarifying subtitle (e.g., *Singular Bayesian Neural Networks: Measure-Theoretic Singularity via Low-Rank Weight Factorization*) to make the intended meaning unambiguous.
>
> Regarding related-work positioning, we agree that the paper should position
> itself more clearly relative to these adjacent low-rank literatures, and, as advised, we
> will use the additional camera-ready space to flesh this out. The closest link
> to reduced-rank regression is the shared use of low-rank structure, but our
> setting is a Bayesian neural predictor with nonlinear feature learning and
> uncertainty-aware prediction/generalization, rather than a classical low-rank
> linear regression model. Relative to Bayesian matrix factorization, the key
> difference is that our factorization parameterizes neural-network weight
> matrices, so the induced posterior is over model parameters and predictors
> rather than over the entries of a directly observed data matrix. Relative to
> low-rank compression, our method learns a stochastic low-rank parameterization
> end-to-end from initialization. This means the computational savings can affect
> both training and inference. By contrast, many compression methods factorize a
> pretrained deterministic model post hoc, so their computational savings are
> realized primarily after training, especially at deployment. We will make these
> distinctions more explicit in the revised related-work section.
>
> We thank the reviewer for the presentation comments. We will fix the two noted typos, regenerate the affected figures with larger internal text, and improve figure readability in the revised version.

---

> > ### Author Rebuttal · Reviewer_PHKp · 2026-03-31
> >
> > - **Re Scaling**: Thank you, I believe your rank-sweep profiling study in response to GhLS addresses the question I intended to ask.
> > - **Re SLT**: Thank you that is a helpful clarification. I think your proposed clarifying subtitle is a good solution. I want to flag that I think there might be a deeper connection between the two senses of "singular" at least in this context. I think this is probably too much for a full treatment and can be left for future work, but you might find it interesting. I'll write this out below.
> > - **Re additional related works**: Thank you for clarifying these distinctions.
> >
> > I will go ahead and increase my score to Accept.
> >
> > PS on the potentially deeper connection to SLT:
> >
> > Suppose you have a full-rank $r=R$ factorization $W=AB^T$. This is a singular parametrization in Watanabe's sense because you can insert arbitrary invertible transformations $W = A \cdot O \cdot O^{-1} \cdot B^T = (A') \cdot (B')$.
> >
> > There's an additional source of degeneracy in the parametrization that occurs if the transformation you're trying to learn itself is low-rank $r<R$. If this is the case, then you can apply an arbitrary (not necessarily invertible) transformation to the null space of the transformation.
> >
> > What this means is that low-rank solutions have a lower RLCT and are thus "favored" by the Bayesian posterior in Watanabe's free energy formula. Asymptotically, you can expect an inductive bias that favors these low-rank solutions, driven purely by the geometry of the parametrization and not explicitly enforced.
> >
> > So the explicit rank constraint you enforce might be amplifying by construction something that is already encouraged through Watanabe's SLT. It seems like an interesting point in favor of the construction being natural. It also suggests the theoretical picture might be richer and that there could be some principled way to select $r$ as it relates to the RLCT.
> >
> > I don't expect the authors to do anything with this in the current paper, but I'll leave it for you to think about.

---

> > > ### Author Response · Authors · 2026-04-01
> > >
> > > Thank you for the careful follow-up and for the positive update. We are glad the additional profiling, SLT clarification, and related-work distinctions addressed your concerns.
> > >
> > > We also appreciate your observation about the potentially deeper connection to Watanabe’s SLT. In particular, the idea that the factorized parametrization is already singular in Watanabe’s sense, and that genuinely low-rank solutions may be further favored through the singular geometry of the model, is a very interesting direction. It goes beyond what we can properly treat in the current paper, but we think it could be a valuable avenue for future work, especially in connection with principled rank selection.
> > >
> > > Thank you again for the thoughtful review and the helpful suggestions throughout.

---

### Official Review · Reviewer_KBq1 · 2026-03-12

**Soundness:** 2
**Presentation:** 2
**Significance:** 3
**Originality:** 3
**Overall Recommendation:** 4
**Confidence:** 3

**Summary:**

The paper presents a formalism for developing more efficient Bayesian neural networks, taking advantage of the strong correlations between weights. Experiments are conducted over a range of architectures to demonstrate 15x reduction in parameters compared to a deep ensemble.

**Compliance With Llm Reviewing Policy:**

Affirmed.

**Final Justification:**

Following the rebuttal, I have raised my score from 3 to 4.

The rebuttal addressed several of my original concerns. In the initial review these were regarding (i) an issue in the proof of Theorem 3.4, (ii) the empirical positioning relative to NLL, (iii) missing uncertainty estimates in Table 2 and (iv) ambiguity in the tables. In particular, the clarification of Theorem 3.4 resolves the logical issue for me, and I also appreciate the planned revisions to improve the presentation of the empirical results.

I find the paper technically solid overall, it makes a meaningful contribution to scalable Bayesian deep learning via a low-rank variational parameterisation that induces structured correlations while substantially reducing parameter count. My remaining reservation is primarily regarding empirical significance and positioning. In the main classification results, the proposed low-rank method is clearly weaker than deep ensembles on NLL, which I consider the most important metric in this context.

**Key Questions For Authors:**

Could the authors add uncertainties to Table 2 and the spider plots?

Could the authors make explicit in table 1 which methods represent novel contributions?

Could the authors clarify if Params includes trainable variational params?

**Limitations:**

Yes

**Strengths And Weaknesses:**

Strengths:

The paper presents a well motivated parameterisation for BNNs, which addresses an import bottleneck in their adoption. A good range of experiments and baselines are presented.

Weaknesses:

The presentation of the experimental results could be improved. For example the text in Figure 4 is very small. Table 2 lacks uncertainty estimates. And in the tables the novel models are not adequately highlighted as distinct from the baselines (eg either with an ‘ours’ label or mention in the caption).

While Low Rank generates strong AUC metrics, it generally does not produce strong NLL values. This is concerning because the motive for adopting BNN is to establish well calibrated uncertainty estimates, so NLL is a crucial metric in this context.

In theorem 3.4  - the statement ´no such density function f exists´ does not necessarily imply that q is singular with respect to lambda.

The text is somewhat at odds with the claims in the abstract. In the main text, model performance is described as a tradeoff, conceding that ´Deep Ensemble maintains superior in-domain discrimination (AUROC=0.929) and calibration´. However the phrasing in the abstract makes it sound more like overall superiority.

---

> ### Author Rebuttal · Authors · 2026-03-27
>
> We thank the reviewer for the careful reading and concrete suggestions.
>
> Regarding Theorem 3.4, the reviewer is right that the absence of a density alone does not imply singularity. The intended argument is stronger and is measure-theoretic: a measure mu is singular with respect to a measure nu, written mu ⟂ nu, if there exists a measurable set A such that mu(A)=1 and nu(A)=0. In our case, the proof already establishes that q_W(R_r)=1 and lambda(R_r)=0, where R_r is the set of rank-at-most-r matrices. Thus taking A=R_r directly yields q_W ⟂ lambda. We will revise the proof to remove the density-based detour and state this final step explicitly. This is a clarification of the final logical step, not a change to the theorem statement or its assumptions.
>
> On  the abstract wording, we agree that it should reflect the trade-off described more carefully in the main text. Our empirical results show that low-rank posteriors often improve OOD detection and uncertainty-aware decision quality, while Deep Ensemble can remain stronger on in-distribution likelihood-based metrics (e.g NLL). We will revise the abstract to state this trade-off directly rather than implying overall dominance.
>
> On NLL versus OOD detection, we agree that the NLL gap relative to deep ensemble is real and should be described. The empirical pattern is  more nuanced than a single "NLL versus OOD" slogan. On MIMIC-III, Low-Rank shows the clearest trade-off: it has weaker NLL than deep ensemble (0.433 vs. 0.300) but stronger OOD detection (AUROC-OOD 0.802 vs. 0.738; AUPR-OOD 0.788 vs. 0.754). On SST-2, deep ensemble is stronger on both NLL (0.434 vs. 0.527) and AUROC-OOD (0.657 vs. 0.640), while Low-Rank remains competitive, achieves the best AUPR-In (0.302 vs. 0.267), and does so at much lower parameter and training cost. On Beijing, the strongest uncertainty result is not OOD AUROC but coverage and selective prediction: Low-Rank has far better ECE/PICP than deep ensemble (0.114/0.790 vs. 0.317/0.310), and at 80% retention it achieves lower MAE (8.71 vs. 9.21), showing that its uncertainty is more useful for abstention even though deep ensemble remains slightly stronger on AUROC-OOD/AUPR-OOD. Our interpretation is that Low-Rank BBB trade predictive sharpness for more useful uncertainty, but the empirical manifestation depends on the task. We will revise the abstract and discussion to state this more precisely rather than implying a uniform pattern.
>
> Relatedly, we ran an SST-2 low-rank ensemble comparison, which shows that the single-model NLL/calibration gap is not intrinsic to the low-rank parameterization. A 5-member low-rank Bayesian ensemble reaches accuracy 0.8140, ECE 0.0536, NLL 0.4152, Brier score 0.1318, and AUROC-OOD-MI 0.7313. Relative to the original single Low-Rank BBB SST-2 row, it substantially improves accuracy (0.8140 vs. 0.7950), ECE (0.0536 vs. 0.1492), NLL (0.4152 vs. 0.5197), and AUROC-OOD-MI (0.7313 vs. 0.6473). It is also, in this run, stronger than the current 4-seed Deep Ensemble summary on ECE, NLL, and AUROC-OOD-MI (0.0536/0.4152/0.7313 vs. 0.0659/0.4342/0.6577), while remaining slightly lower on accuracy (0.8140 vs. 0.8250). The gain comes from both within-member Bayesian uncertainty and between-member disagreement, with stronger OOD amplification from the latter (between-member ratio 1.63 vs. within-member ratio 1.26). In efficiency terms, the 5-member low-rank ensemble uses 7.5M parameters and trains in 37.9 min(on average 7.6min/member) versus 49.6M and 64.7 min for the 5-member Deep Ensemble. This suggests that the single-model NLL gap is a recoverable trade-off, not a fundamental limitation of the low-rank approach.
>
> Regarding Table 2 and the spider plots, we agree. Table 1 and 3 already report mean ± std across runs, whereas Table 2 currently shows only point estimates despite being averaged over runs. We have already extracted the multi-seed summary statistics for the Beijing experiment and will revise Table 2 to report mean ± std for the stochastic methods. The revised Low-Rank row will report 10.63 ± 0.20 for MAE, 0.114 ± 0.005 for ECE, 0.790 ± 0.006 for PICP, 0.710 ± 0.021 for AUROC-OOD, and 0.861 ± 0.022 for AUPR-OOD. For the spider plots, we will clarify in the caption that they visualize the same run-averaged quantities, and we will keep the exact uncertainty values in the accompanying table rather than cluttering the plots with error bars.
>
> On table presentation, we will also make the proposed models explicit in the table/caption formatting by marking the low-rank variants as **ours** rather than leaving them visually mixed with the baselines.
>
> Regarding "Params" in the current tables it refers to the model total trainable parameter count used in the implementation. For variational models this includes the trainable variational parameters (mu, rho), and for deep ensemble we report the sum across members. We will state this explicitly in the caption to avoid ambiguity.

---

> > ### Author Rebuttal · Reviewer_KBq1 · 2026-04-04
> >
> > Thank you for the thoughtful rebuttal, I'm glad the feedback was helpful in improving the manuscript, and I will update my score accordingly.

---

> > > ### Author Response · Authors · 2026-04-05
> > >
> > > Thank you for the careful follow-up and for the helpful suggestions throughout, which helped improve the manuscript. We are glad the rebuttal addressed your concerns.

---

### Official Review · Reviewer_Gk8P · 2026-03-12

**Soundness:** 3
**Presentation:** 3
**Significance:** 3
**Originality:** 2
**Overall Recommendation:** 5
**Confidence:** 3

**Summary:**

This paper proposes a low-rank variational inference framework for Bayesian neural networks that parameterizes weight matrices as W=A*B.T, learning distributions over the low-rank factors instead of full weight matrices. This induces a singular posterior concentrated on the rank-r manifold, which introduces structured correlations between weights while reducing the number of variational parameters from O(mn) to O(r(m+n)). The authors provide theoretical analysis and present generalization bounds (PAC-Bayes and Gaussian complexity) that improve with lower rank. Empirically the method is evaluated on MLPs, LSTMs, and Transformers across clinical and time series benchmarks, showing competitive predictive performance, improved OOD detection and uncertainty estimation compared to full rank Bayesian neural networks and deep ensembles.

**Compliance With Llm Reviewing Policy:**

Affirmed.

**Final Justification:**

The rebuttal has answered my questions and I am content. I am increasing my score to 5.

**Key Questions For Authors:**

1. The experiments mainly compare against Deep Ensembles, Full-Rank BBB, and Rank-1 methods. Could the authors include or discuss comparisons with other scalable Bayesian approaches such as SWAG or low-rank covariance variational methods?
2. The rank is selected via ablation and sometimes guided by singular value decay. How sensitive is performance (especially OOD detection and calibration) to the choice of r?
3. The experiments include a relatively small Transformer. Do the authors have evidence or preliminary results showing how the approach scales to larger architectures (ex: larger language models or deeper transformers)? Such results would strengthen the claims about scalability.
4. The results suggest a tradeoff where low-rank models improve OOD detection but often worsen NLL/ECE compared to deep ensembles. Do the authors have insights or experiments (low-rank ensembles or different priors) that mitigate this calibration gap?

**Limitations:**

Yes.

**Strengths And Weaknesses:**

Strengths

1. The paper proposes a low rank variational parameterization W=A*B.T that reduces parameter complexity while introducing structured correlations between weights, and the method is well aligned with known low rank structure in neural networks.
2. The work provides multiple theoretical analyses (posterior geometry, loss approximation via Eckart–Young–Mirsky, PAC-Bayes bounds, and Gaussian complexity), which together give a principled justification for the proposed approach.
3. Experiments span MLPs, LSTMs, and Transformers and multiple domains (healthcare, time series forecasting) demonstrating that the approach is broadly applicable.
4. The method reduces parameters and training cost while maintaining competitive performance, which is valuable for scalable Bayesian deep learning.

Weaknesses

1. The experimental evaluation does not include some relevant baselines (e.g., SWAG or other low-rank covariance approaches) which weakens the empirical positioning of the method.
2. While OOD detection improves, predictive calibration and NLL are often worse than deep ensembles, which may limit practical adoption.
3. The method requires tuning of both rank r and KL weight beta, and guidance for selecting them is somewhat heuristic.
4. Parts of the theory mainly formalize expected properties of low-rank factorization rather than introducing fundamentally new analytical techniques.
5. Results are limited to moderately sized models, evaluation on larger modern architectures would strengthen claims about scalability.

---

> ### Author Rebuttal · Authors · 2026-03-27
>
> We thank the reviewer for the thoughtful feedback. Regarding SWAG, we agree it is an important comparison. We ran a multi-seed SWAG baseline. On SST-2 with AG News OOD, the SWAG backbone uses the paper's setup for the pre-collection phase and 20 extra collection epochs. Across 4 seeds: accuracy $0.8076\pm0.0097$, NLL $0.5564\pm0.0216$, ECE $0.1109\pm0.0078$, and AUROC-OOD-MI $0.6133\pm0.0227$, with 208.37M parameters. Compare to Low-Rank BBB, SWAG is tied in accuracy (+0.0008) and better in ECE (-0.0624), but worse in NLL (+0.0292) and AUROC-OOD (-0.0274), while storing 141.7x more parameters and taking 3.4x longer.
> The picture is similar on MIMIC-III. SWAG is competitive on AUROC (0.9217), but worse on NLL (0.359 vs 0.3414), ECE (0.1899 vs 0.175), and AUROC-OOD (0.6342 vs 0.7315), while using 24x more parameters.
> For Beijing PM2.5, it is competitive on point prediction (MAE 10.6029, RMSE 17.34, NLL 4.3248) and improves over the deterministic baseline, but remains $27\times$ larger and does not dominate on OOD detection, where Low-Rank and Deep Ensemble are stronger.
> Overall, SWAG is a relevant baseline, but across these datasets it does not overturn the central quality-efficiency of our proposed approach. We will include these results in the revised version as well as the code and implementation details in the supplementary material.
>
> Regarding low-rank covariance variational methods, we discuss Tomczak et al. and Swiatkowski et al. in the related-work
> section but have not reimplemented those posterior families. Tomczak et al. use a full-mean Gaussian with
> low-rank-plus-diagonal covariance; Swiatkowski et al. compress mean-field scales via low-rank parameterization. Our method factorizes weight matrices so low-rank structure constrains both parameterisation and induced posterior geometry. Comparing induced covariance structures would be interesting but is complementary analysis beyond this rebuttal.
>
> On theory novelty, we do not claim a new generic Bayesian or PAC-Bayes machinery. The theoretical contribution is a precise finite-sample analysis of this low-rank BNN construction, where several intuitive properties are not automatic and require explicit proof. Much of the prior low-rank Bayesian literature focused on efficiency and scalability; how low-rank structure affects generalization has been addressed less explicitly. This paper begins closing that gap with theoretical insights for the empirically observed generalization behavior of low-rank models. It proves:(i) measure-theoretic singularity of the induced posterior on weights and its non-trivial correlation structure, (ii) rank-sensitive PAC-Bayes scaling for the induced low-rank posterior, and (iii) a predictive-mean generalization argument in the Bayesian setting based on Gaussian complexity transfer. Our aim is not to oversell novelty but to make explicit a theory package specific to this construction and tied to observed efficiency-uncertainty-generalization behavior. We will revise the novelty language to sharpen this distinction.
>
> On rank selection, the paper includes Pareto-style plots (Fig.~10-13) showing how rank configurations move the trade-off between OOD detection and calibration (NLL/ECE). Rank was chosen via reduced-budget ablations; when a deterministic reference was available, singular-value decay served as a sanity check. This is a reasonable protocol for the present work. However, the rank selection process is itself a meaningful research problem. An automated rank-selection procedure is a natural direction for future work. For the related question about the KL scale, we refer to our response to Reviewer GhLS.
>
> On scalability, we agree that larger-scale evidence would strengthen the paper. Our strongest current large-model evidence is the SST-2 Transformer experiment and the controlled profiling reported in our response to Reviewer PHKp, showing parameter, memory, and end-to-end time advantages over Full-Rank BBB and Deep Ensembles at the million-parameter scale. We do not claim results at LLM scale yet and will clarify scope in the revision. The trend is favorable and visible on the largest architecture studied; extending to larger transformers is a meaningful next step.
>
> Regarding the calibration gap, we agree ensembling is the most
> natural mitigation strategy. We stated this in our discussion section and in appendix H.1 and we now have direct SST-2 evidence for it; see
> our response to Reviewer KBq1 for the full result and uncertainty decomposition.
> A 5-member low-rank Bayesian ensemble reaches accuracy 0.8140, ECE 0.0536, NLL
> 0.4152, and AUROC-OOD-MI 0.7313. Relative to the original single Low-Rank BBB
> row, this improves calibration and OOD detection, and it is also
> stronger than the current 4-seed Deep Ensemble summary on ECE, NLL, and
> AUROC-OOD-MI, while remaining slightly lower on accuracy, with the stronger
> OOD separation driven more by **between-member** disagreement than by within-member
> Bayesian uncertainty alone.

---

> > ### Author Rebuttal · Reviewer_Gk8P · 2026-04-03
> >
> > Thank you for the rebuttal. The authors have addressed my questions and I am content. I will increase my score accordingly.

---

> > > ### Author Response · Authors · 2026-04-03
> > >
> > > Thank you for the careful follow-up and the helpful suggestions throughout. We are glad the additional SWAG comparisons, the clarification on rank sensitivity and scalability, and the low-rank ensembling result addressed your concerns.

---

### Decision · Program_Chairs · 2026-04-30

**Decision:**

Accept (regular)

**Comment:**

This paper proposes a low-rank factorization of the weights of Bayesian neural networks (BNNs). The reviewers point to some limitations in the experimental validation and the significance of the theoretical developments. The rebuttal phase has cleared up some doubts raised by the reviewers, but some criticism on the experimental validation and the significance of the theoretical developments still stands. I fully agree with these limitations, and I would also point out that the paper should improve its positioning/comparisons with respect to the literature on the factorization of weight matrices in Bayesian neural networks, e.g. [1,2].

Despite these limitations, the reviewer have a generally positive opinion of this paper. We encourage the authors to take into account the feedback of the reviewers when preparing their revision.

[1] C. Hawkins and Z. Zhang. Bayesian tensorized neural networks with automatic rank selection, Neurocomputing, Volume 453, 2021, Pages 172-180,

[2] S. Rossi, S. Marmin, and M. Filippone. Walsh-Hadamard Variational Inference for Bayesian Deep Learning. In NeurIPS 2019.